

**The Positive Effect of Formaldehyde on the Photocatalytic**
**Renoxification of Nitrate on TiO$_2$ Particles**
Yuhan Liu, Xuejiao Wang, Mengshuang Sheng, Chunxiang Ye, Jing Shang*
*State Key Joint Laboratory of Environmental Simulation and Pollution Control,*
*College of Environmental Sciences and Engineering, Peking University, 5 Yiheyuan*
*Road, Beijing 100871, P. R. China*
Corresponding author: Jing Shang
Email: shangjing@pku.edu.cn
**Abstract**
Renoxification is the recycling of NO$_3^-$/HNO$_3$ into NO$_x$ under illumination; it is
promoted by the photocatalysis of TiO$_2$. Formaldehyde (HCHO), the most abundant
carbonyl compound in the atmosphere, may participate in the renoxification of
nitrate-doped TiO$_2$ (NO$_3^-$-TiO$_2$) aerosols. In this study, we established an
environmental chamber reaction system under different light sources, excluding direct
photolysis of nitrate by adjusting the illumination wavelength, to explore the
photocatalytic renoxification process. It is suggested that HCHO and TiO$_2$ have a
significant synergistic effect on photocatalytic renoxification via the
NO$_3^-$-NO$_3\cdot$-HCHO-HNO$_3$-NO$_x$ pathway. Adsorbed HCHO may react with nitrate
radicals through hydrogen abstraction to form HNO$_3$ on the surface, resulting in the



mass generation of $NO_x$. We found that for 4 wt% $NO_3^-$-$TiO_2$ aerosols (e.g.,
$KNO_3$-$TiO_2$), the $NO_x$ concentration reached up to 110 ppb, and was 2 orders of
magnitude higher than in the absence of HCHO. Nitrate type and contents, relative
humidity, and HCHO concentration were found to influence $NO_x$ release. The
significant synergistic enhancement effect of renoxification affects photochemical
processes such as atmospheric oxidation and nitrogen cycling on the surfaces of
particles containing semiconductor oxides, with the participation of hydrogen donor
organics.
**1 Introduction**
The levels of ozone ($O_3$) and hydroxyl radicals ($\cdot OH$) in the troposphere can be
promoted by nitrogen oxides ($NO_x = NO + NO_2$), such that $NO_x$ plays an important
role in the formation of secondary aerosols and atmospheric oxidants (Platt et al.,
1980; Stemmler et al., 2006; Harris et al., 1982; Finlayson-Pitts and Pitts, 1999). $NO_x$
can be converted into nitric acid ($HNO_3$) and nitrate ($NO_3^-$) through a series of
oxidation and hydrolysis reactions and is eventually removed from the atmosphere
through subsequent wet or dry deposition (Dentener and Crutzen, 1993; Goodman et
al., 2001; Monge et al., 2010; Bedjanian and El Zein, 2012). However, comparisons
of observations and modeling results for the marine boundary layer, land, and free
troposphere (Read et al., 2008; Lee et al., 2009; Seltzer et al., 2015) have shown
underestimation of $HNO_3$ or $NO_3^-$ content, $NO_x$ abundance, and $NO_x$/$HNO_3$ ratios,
indicating the presence of a new, rapid $NO_x$ circulation pathway (Ye et al., 2016b;



Reed et al., 2017). Some researchers have suggested that deposited $NO_3^-$ and $HNO_3$
can be recycled back to gas phase $NO_x$ under illumination, via the renoxification
process (Schuttlefield et al., 2008; Romer et al., 2018; Bao et al., 2020; Shi et al.,
2021b). Photolytic renoxification occurs under light with a wavelength of < 350 nm,
through the photolysis of $NO_3^-/HNO_3$ adsorbed on the solid surface to generate $NO_x$.
Notably, the photolysis of $NO_3^-HNO_3$ is reported to occur at least 2 orders of
magnitude faster on different solid surfaces (natural or artificial) or aerosols than in
the gas phase (Ye et al., 2016a; Zhou et al., 2003; Baergen and Donaldson, 2013).
Several recent studies have shown that renoxification has important atmospheric
significance (Deng et al., 2010; Kasibhatla et al., 2018; Romer et al., 2018; Alexander
et al., 2020), providing the atmosphere with a new source of photochemically reactive
nitrogen species, i.e., HONO or $NO_x$, resulting in the production of more
photooxidants such as $O_3$ or ·OH (Ye et al., 2017), which further oxidize volatile
organic compounds (VOCs), leading to the formation of more chromophores, thereby
affecting the photochemical process (Bao et al., 2020).

60        Renoxification processes have recently been observed on different types of

atmospheric particles, such as urban grime and mineral dust (Ninneman et al., 2020;
Bao et al., 2018; Baergen and Donaldson, 2013; Ndour et al., 2009). Atmospheric
titanium dioxide ($TiO_2$) is mainly derived from windblown mineral dust, with mass
mixing ratios ranging from 0.1 to 10% (Chen et al., 2012). $TiO_2$ is widely used in
industrial processes and building exteriors for its favorable physical and chemical
properties. Titanium and nitrate ions have been found to coexist in atmospheric

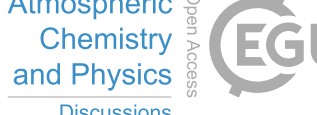

particulates in different regions worldwide (Sun et al., 2005; Schwartz-Narbonne et al.,
2019). The relative content of $TiO_2$ and $NO_3^-$ in atmospheric particles varies greatly,
and nitrate-coated $TiO_2$ ($NO_3^-$-$TiO_2$) aerosols containing $TiO_2$ as the main body can
be used to effectively represent particles for sandstorm modeling (Sun et al., 2005;
Kim et al., 2012). $TiO_2$ is a semiconductor metal oxide that can facilitate the
photolysis of nitrate and the release of $NO_x$ due to its photocatalytic activity (Ndour et
al., 2009; Chen et al., 2012; Verbruggen, 2015; Schwartz-Narbonne et al., 2019).
Under ultraviolet (UV) light, $TiO_2$ generates electron-hole pairs in the conduction and
valence bands, respectively (Linsebigler et al., 1995). Nitrate ions adsorbed at the
oxide surface react with the photogenerated holes ($h^+$) to form nitrate radicals ($NO_3\cdot$),
which are subsequently photolyzed to $NO_x$, mainly under visible light illumination
(Schuttlefield et al., 2008; George et al., 2015; Schwartz-Narbonne et al., 2019). Thus,
the renoxification of $NO_3^-$ is faster on $TiO_2$ than on other oxides in mineral dust
aerosols such as $SiO_2$ or $Al_2O_3$ (Lesko et al., 2015; Ma et al., 2021). In this study, we
refer to renoxification involving $h^+$ and $NO_3^-$ in the reaction as photocatalytic
renoxification based on the photocatalytic properties of $TiO_2$.
Many previous studies have focused mainly on particulate nitrate-$NO_x$
photochemical cycling reactions, despite the potential impact of other reactant gases
in the atmosphere. Formaldehyde (HCHO), the most abundant carbonyl compound in
the atmosphere, which can react at night with $NO_3\cdot$ via hydrogen abstraction reactions
to form $HNO_3$ (Atkinson, 1991). Our previous study showed that the degradation rate
of HCHO was faster on $NO_3^-$-$TiO_2$ aerosols than on $TiO_2$ particles, perhaps as a result



of HCHO oxidation by $NO_3 \cdot$ (Shang et al., 2017). To date, no studies have reported the
effect of HCHO on photocatalytic renoxification. Adsorbed HCHO would react with
$NO_3 \cdot$ generated on the $NO_3^-$-$TiO_2$ aerosol surface, thus alter the surface nitrogenous
species and renoxification process. The present study is the first to explore the
combined effect of HCHO and photocatalytic $TiO_2$ particles on the renoxification of
nitrate. The wavelengths of the light sources were adjusted to exclude photolytic
renoxification while making photocatalytic renoxification available for better
elucidate the reaction mechanism. We investigated the effects of various influential
factors including nitrate type, nitrate content, RH, and initial HCHO concentration, to
understand the atmospheric renoxification of nitrate in greater detail.
**2 Methods**
**2.1 Environmental chamber setup**
Details of the experimental apparatus and protocol used in the current study have been
previously described (Shang et al., 2017). Briefly, the main body of the environmental
chamber is a 400 L polyvinyl fluoride (PVF) bag filled with synthetic air (high purity
$N_2$ (99.999%) mixed with high purity $O_2$ (99.999%) in the ratio of 79:21 by volume,
Beijing Huatong Jingke Gas Chemical Co.). The chamber is capable of temperature
(~293 K) and relative humidity (0.8–70%) control using a water bubbler and air
conditioners, respectively. The chamber is equipped with two light sources both with
the central wavelength of 365 nm. One is a set of tube lamps with a main spectrum of
320–400 nm and a small amount of 480-600 nm visible light (Figure S1a). The other
is a set of Light-emitting diode (LED) lamps with a narrow main spectrum of 350-390



111 nm (Figure S1b). The light intensities for the tube and LED lamp at 365 nm were 300

112 $\mu W \cdot cm^{-2}$ and 200 $\mu W \cdot cm^{-2}$, respectively, measured in the middle of the chamber.

113 Aerosol samples were introduced into the chamber by a transient high-pressure

114 airflow. $NO_x$ concentrations at the outlet of the chamber were monitored by a

115 chemiluminescence $NO_x$ analyzer (ECOTECH, EC9841B). HCHO was generated by

116 thermolysis of paraformaldehyde at 70 °C and detected via acetyl acetone

117 spectrophotometric method using a UV-Vis spectrophotometer (PERSEE, T6) or a

118 fluorescence spectrophotometer (THERMO, Lumina), depending on different initial

119 HCHO concentrations. The particle size distribution was measured by a Scanning

120 Nano Particle Spectrometer (HCT, SNPS-20). Electron Spin Resonance

121 (Nuohai Life Science, MiniScope MS 5000) was used to measure ·OH on the surface

122 of particles. 5,5-dimethl-1-pyrroline-N-oxide (DPMO, Enzo) was used as the capture

123 agent. 50 μL particle-containing suspension mixed with 50 μL DMPO (concentration

124 of 200 μM) was loaded in a 1 mm capillary. Four 365 nm LED lamps were placed

125 side by side vertically at a distance of about 1 cm from the capillary, and the

126 measurement was carried out after 1 min of irradiation. The modulation frequency

127 was 100 kHz, the modulation amplitude was 0.2 mT, the microwave power was 10

128 mW and the sweep time was 60 s.

129 **2.2 Nitrate-TiO₂ composite samples**

130 In our experiments, two nitrate salts, potassium nitrate (AR, Beijing Chemical Works

131 Co., Ltd) or ammonium nitrate (AR, Beijing Chemical Works Co., Ltd), were

132 complexed with pure $TiO_2$ (≥ 99.5%, Degussa AG) powder or $TiO_2$ (1 wt.%)/$SiO_2$



mixed powder to prepare $NO_3^--TiO_2$ or $NO_3^--TiO_2$ (1 wt.%)/$SiO_2$ samples. $TiO_2$ was
simply mixed in nitrate solutions at the desired mass mixing ratio (with nitrate content
of 1 wt.%, 4 wt.%, 20 wt.%, 80 wt.% and 95 wt.%) to obtain a mash. The mash was
dried at 90 °C and then ground carefully to ensure a uniform composite of particles.
$SiO_2$ (AR, Xilong Scientific Co., Ltd.) with no optical activity was also chosen for
comparison, and samples of $KNO_3-SiO_2$ and $KNO_3-TiO_2$(1 wt.%)/$SiO_2$ samples with
a potassium nitrate content of 4 wt.% were prepared. The blank $TiO_2$ sample was
solved in pure water with the same procedure as mentioned above. 4 wt.%
$HNO_3-TiO_2$ composite particles were prepared for comparison. Concentrated nitric
acid (AR, Beijing Chemical Works Co., Ltd) was diluted to 1 M and $TiO_2$ was added
to the nitric acid solution and stirred evenly. A layer of aluminum foil was covered on
the surface of the $HNO_3-TiO_2$ homogenate and dried naturally in the room. After
air-drying, follow the same steps above to grind for use. We also selected Arizona Test
Dust (ATD, Powder Technology Inc.), whose chemical composition and weight
percentage were shown in Table S1, as a substitute of $NO_3^-/TiO_2$ to investigate the
"photocatalytic renoxification" process of nitrate and the positive effect of HCHO.
**2.3 Environmental chamber experiments**
The experiments carried out in the environmental chamber can be divided into two
categories according to whether HCHO was involved or not. (1) No HCHO
involvement in the reaction. The PVF bag was inflated by 260 L synthetic air, and
then 75 mg $TiO_2$ particles were sprayed into PVF bag. As shown in Figure S2, the
concentration of the particles decreased rapidly due to the sedimentation of the larger



particles and the electrostatic adsorption of the particles by the environmental
chamber. The size distribution of $TiO_2$ reached stable after about 60 min with the peak
particle size was about 120 nm, similar to that of atmospheric particles in some urban
areas in China (Wang et al., 2015; Li et al., 2019). The size distribution could
maintain for more than 4 hours, with the number concentration in the chamber
decreased by no more than 5% per hour. (2) With the participation of HCHO. The
PVF bag was inflated by 125 L synthetic air, followed by the introduction of HCHO,
and then the chamber was filled up with zero air to about 250 L. It can be seen from
Figure S3 that it took about 60 min for the HCHO concentration to reach stable. Then,
75 mg $TiO_2$ or $NO_3^-/TiO_2$ powders were introduced and the concentration of HCHO
decreased upon the introduction. It took about another 60 min for HCHO
concentration to get stable. After the concentrations of both HCHO and aerosol
became stable, the lamps were turned on and the concentrations of $NO_x$ were
monitored.
To determine the background value of $NO_x$ in the reaction system, four blank
experiments were carried out under illumination without nitrate: "synthetic air",
"synthetic air + $TiO_2$", "synthetic air + HCHO" and "synthetic air + HCHO + $TiO_2$".
In the blank experiments of "synthetic air" and "synthetic air + $TiO_2$", the $NO_x$
concentration remained stable during 180 min illumination, and the concentration
change was no more than 0.5 ppb (Figure S4a). Therefore, the environmental chamber,
synthetic air and the surface of $TiO_2$ particles were thought to be relatively clean, and
there was no generation and accumulation of $NO_x$ under illumination. When HCHO



was introduced into the environmental chamber, $NO_x$ accumulated ~2 ppb in 120 min
with or without $TiO_2$ particles (Figure S4b). Compared with the blank experiment
results when there was no HCHO, $NO_x$ might come from the generation process of
HCHO (impurities in paraformaldehyde). However, considering the high
concentration level of $NO_x$ produced in the $NO_3^-$-$TiO_2$ system containing HCHO
under the same conditions in this study (see later in Figure 2), the $NO_x$ generated in
this blank experiment can be negligible.
**3 Results and discussion**
**3.1 The positive effect of $TiO_2$ on the renoxification process**
We investigated the photocatalytic role of $TiO_2$ on renoxification. The light source
was two 365 nm tube lamps containing small amounts of 400–600 nm visible light;
this setup was suitable for exciting $TiO_2$ and the photolysis of available nitrate
radicals. Raw $NO_x$ data measured in the chamber under dark and illuminated
conditions for 4 wt.% $KNO_3$-$SiO_2$ and 4 wt.% $KNO_3$-$TiO_2$ (1 wt.%)/$SiO_2$ are shown
in Figure 1. The ratio of 1 wt. % $TiO_2$ to $SiO_2$ corresponds to their ratio in sand and
dust particles. We observed no $NO_x$ in the $KNO_3$-$SiO_2$ sample under dark or
illumination, indicating very weak direct photolysis of nitrate under our 365 nm
tube-lamp illumination conditions. However, when the sample containing $TiO_2$/$SiO_2$
was illuminated, $NO_x$ continually accumulated in the chamber. This finding confirms
that $NO_x$ production arising from photodissociation of $NO_3^-$ on $TiO_2$/$SiO_2$ was caused
by the photocatalytic property of $TiO_2$ (i.e., photocatalytic renoxification) and was not
due to the direct photolysis of $NO_3^-$ (photolytic renoxification).



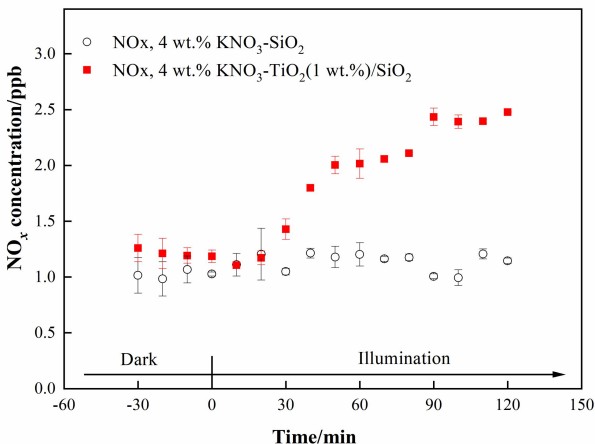


**Figure 1.** Effect of illumination on the release of $NO_x$ from 4 wt.% $KNO_3$-$SiO_2$ and 4

wt.% $KNO_3$-$TiO_2$(1 wt.%)/$SiO_2$ at 293 K and 0.8% of relative humidity. 365 nm tube

lamps were used during the illumination experiments.

$TiO_2$ can be excited by UV illumination to generate electron-hole pairs, and the

$h^+$ can react with adsorbed $NO_3^-$ to produce $NO_3 \cdot$ (Ndour et al., 2009). Thus, in the

present study, $NO_3 \cdot$ mainly absorbed visible light emitted from the tube lamps, which

was subsequently photolyzed to $NO_x$ through Eqs. (3) and (4) (Wayne et al., 1991),

which explains why $NO_x$ was observed in this study. Thus, we demonstrated that $TiO_2$

can be excited at illumination wavelengths of ~365 nm, even when then content was

very low, and that $NO_x$ accumulated due to the production and further phytolysis of

$NO_3 \cdot$. However, the production rate of $NO_x$ was very slow, reaching only 1.3 ppb

during 90 min of illumination. This result may have been caused by the blocking

effect of $K^+$ on $NO_3^-$. $K^+$ forms ion pairs with $NO_3^-$, and electrostatic repulsion

between $K^+$ and $h^+$ prevents $NO_3^-$ from combining with $h^+$ to generate $NO_3 \cdot$ to a

certain extent, thereby weakening the positive effect of $TiO_2$ on the renoxification of





KNO$_3$ (Rosseler et al., 2013).

$$TiO_2 + h\upsilon\ (\lambda < 390\ nm) \rightarrow e^- + h^+ \qquad (1)$$


$$NO_3^- + h^+ \rightarrow NO_3\cdot \qquad (2)$$

$$NO_3\cdot\ +\ h\upsilon\ (\lambda < 640\ nm) \rightarrow NO_2 + O\cdot \qquad (3)$$

$$NO_3\cdot\ +\ h\upsilon\ (585\ nm < \lambda < 640\ nm) \rightarrow NO + O_2 \qquad (4)$$

**3.2 The synergistic positive effect of TiO$_2$ and HCHO on the renoxification**
**process**
LED lamps with a wavelength range of 350–390 nm and no visible light were used to
irradiate 4 wt.% KNO$_3$-TiO$_2$ without generating NO$_x$ (NO$_2$ and NO concentrations
fluctuate within the error range of the instrument) (Figure S5). TiO$_2$ can be excited
under this range of irradiation, producing NO$_3$ radicals as discussed above. The lack
of NO$_x$ generation indicates that neither nitrate photolysis nor NO$_3\cdot$ photolysis
occurred under 365 nm LED lamp illumination conditions. In addition, it has been
shown that NO$_3\cdot$ photolysis only occurs in visible light (Aldener et al., 2006).
Therefore, the LED lamp setup was used in subsequent experiments to exclude the
direct photolysis of both KNO$_3$ and NO$_3\cdot$, but allow the excitation of TiO$_2$. This
approach allowed us to investigate the process of photocatalytic renoxification caused
by HCHO in the presence of photogenerated NO$_3\cdot$.

230        Atmospheric trace gases can undergo photocatalytic reactions on the surface of

TiO$_2$ (Chen et al., 2012). As the illumination time increased, the concentration of
HCHO showed a linear downward trend, which was consistent with zero-order



reaction kinetics (Figure S6). The zero-order reaction rate constants of HCHO on
$TiO_2$ and 4 wt.% $KNO_3$-$TiO_2$ particles were $9.1 \times 10^{-3}$ and $1.4 \times 10^{-2}$ ppm min$^{-1}$,
respectively, which were much higher than that for gaseous HCHO photolysis (Shang
et al., 2017). We suggested that the produced $NO_3·$ contributed to the enhanced uptake
of HCHO. Therefore, we suggest that $NO_3·$ production contributed to enhanced
HCHO uptake. Future studies should explore whether HCHO affects the
photocatalytic renoxification of $NO_3^-$-$TiO_2$.
Variation in $NO_x$ concentration within the chamber containing nitrate-$TiO_2$
particles with or without HCHO is shown in Figure 2. For $KNO_3$-$TiO_2$ particles, the
$NO_x$ concentration began to increase upon irradiation in the presence of HCHO,
reaching ~110 ppb within 120 min. This result indicates that HCHO greatly promoted
photocatalytic renoxification of $KNO_3$ on the surfaces of $TiO_2$ particles. This reaction
process can be divided into two stages: a rapid increase within the first 60 min and a
slower increase within the following 60 min, each consistent with zero-order reaction
kinetics. The slow stage is due to the photodegradation of HCHO on $KNO_3$-$TiO_2$
aerosols, which led to a decrease in its concentration, gradually weakening the
positive effect. $NO_x$ is the sum of $NO_2$ and $NO$, both of which showed a two-stage
concentration increase. The $NO_2$ generation rate was nearly 6 times that of NO, as
compared to using the zero-order rate constant within 60 min (1.18 ppb min$^{-1}$ $NO_2$, $R^2$
= 0.96; 0.19 ppb min$^{-1}$ NO, $R^2$ = 0.91). This burst-like generation of $NO_x$ can be
ascribed to the reaction between generated $NO_3·$ and HCHO via hydrogen abstraction
to form adsorbed nitric acid ($HNO_3$(ads)) on $TiO_2$ particles. Based on the analysis of





the absorption cross section of HNO$_3$ adsorbed on fused silica surface,the HNO$_3$(ads)
absorption spectrum has been reported to be red-shifted compared to HNO$_3$(g),
extending from 350 to 365 nm, with a simultaneous cross-sectional increase (Du and
Zhu, 2011). Therefore, HNO$_3$(ads) was subjected to photolysis to produce NO$_2$ and
HONO (Eqs. (6)-(8)) under the LED lamp used in this study. A previous study of
HNO$_3$ photolysis on the surface of Pyrex glass showed that the ratio of the formation
rates of photolysis products (J$_{NOx}$/J$_{(NOx+HONO)}$) was > 97% at RH = 0% (Zhou et al.,
2003), suggesting that NO$_x$ is the main gaseous product under dry conditions. Thus,
the effect of HONO on product distribution and NO$_x$ concentration was negligible in
this study. Together, these results suggest that NO$_3$· and HCHO generate HNO$_3$(ads)
on particle surfaces through hydrogen abstraction, which contributes to the substantial
release of NO$_x$ via photolysis. This photocatalytic renoxification via the
NO$_3^-$-NO$_3$·-HCHO-HNO$_3$-NO$_x$ pathway is important considering the high abundance
of hydrogen donor organics in the atmosphere.

$$NO_3\cdot + HCHO \rightarrow CHO\cdot + HNO_3(ads) \qquad (5)$$

$$HNO_3(ads) + h\upsilon \rightarrow [HNO_3]^*(ads) \qquad (6)$$

$$[HNO_3]^*(ads) \rightarrow HNO_2(ads) + O(^3P)(ads) \qquad (7)$$

$$[HNO_3]^*(ads) \rightarrow NO_2(ads) + \cdot OH(ads) \qquad (8)$$


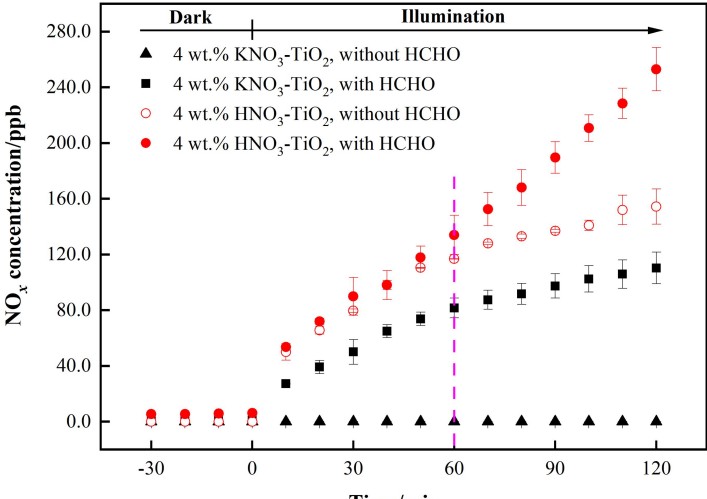


**Figure 2.** Effect of formaldehyde on the renoxification processes of different nitrate-

doped particles at 293 K and 0.8% of relative humidity. 365 nm LED lamps were used

during the illumination experiment. The initial concentration of HCHO was about 9

ppm.

To demonstrate the proposed HCHO mechanism and the photolysis contribution

of $HNO_3$ to $NO_x$, we prepared an $HNO_3$-$TiO_2$ sample by directly dissolving $TiO_2$ into

dilute nitric acid. The formation of $NO_x$ on $HNO_3$-$TiO_2$ without HCHO under

illumination was obvious (Figure 2), and occurred even more rapidly than that on

$KNO_3$-$TiO_2$ with HCHO. The renoxification of $HNO_3$-$TiO_2$ particles was further

enhanced following the introduction of HCHO. The $NO_x$ concentration increased by

~250 ppb after 2 h of illumination, which was 2.2 times faster than the increase in

$KNO_3$-$TiO_2$ concentration under the same conditions. This difference is due to the fact

that $HNO_3$ dissociates on particle surfaces to generate $NO_3^-$, such that $HNO_3$ exists on

$TiO_2$ as both $HNO_3$(ads) and $NO_3^-$(ads). Similarly, $NO_3^-$(ads) completed the



$NO_3^-$-$NO_3\cdot$-HCHO-$HNO_3$-$NO_x$ pathway as described above through the reaction
process shown in Eqs. (2) to (8). The rates of $NO_x$ production from $HNO_3$-$TiO_2$
particles with and without HCHO were similar for the first 60 min (Figure 2), mainly
due to the direct photolysis of partial $HNO_3$(ads). However, after 60 min, $NO_x$ was
generated rapidly in the presence of HCHO, perhaps due to the dominant
photocatalytic renoxification of $NO_3^-$(ads). These findings indicate that HCHO
converts $NO_3^-$ on particle surfaces into $HNO_3$(ads) by reacting with $NO_3\cdot$, and then
$HNO_3$(ads) photolyzes at a faster rate to generate $NO_x$, allowing HCHO to enhance
the formation of $NO_x$. Overall, the photocatalytic renoxification of $NO_3^-$-$TiO_2$
particles affects atmospheric oxidation and the nitrogen cycle, and the presence of
HCHO further enhances this impact.
Photocatalytic renoxification reaction occurs on the surfaces of mineral dust due
to the presence of semiconductor oxides with photocatalytic activity such as $TiO_2$
(Ndour et al., 2009). In this study, we selected the commercial mineral dust ATD to
study the effects of HCHO on this process. We detected $\cdot$OH in irradiated pure $TiO_2$
and ATD samples using electron spin resonance (ESR) technique, and found that for
ATD samples, the peak intensity of $\cdot$OH generation was 40% that of $TiO_2$ samples
(Figure S8). $\cdot$OH originates in the reaction of $h^+$ with surface adsorbed water (Ahmed
et al., 2014). ATD contains semiconductor oxides such as $TiO_2$ and $Fe_2O_3$, and is
thought to exhibit photocatalytic properties affecting the renoxification of nitrate. The
$NO_3^-$ content of ATD is $4 \times 10^{17}$ molecules $m^{-2}$, which is ~0.25 wt.% of the total mass
(Huang et al., 2015; Jiyeon et al., 2017). The $NO_x$ concentration changes observed in


307 the environmental chamber demonstrated that HCHO promoted the renoxification of

308 ATD particles (Figure S9). This result suggests that mineral dust containing

309 photocatalytic semiconductor oxides such as $TiO_2$, $Fe_2O_3$, and ZnO can greatly

310 promote the conversion of granular nitrate to $NO_x$ in the presence of HCHO.

311 **3.3 Influential factors on the photocatalytic renoxification process**

312 **3.3.1 The influence of nitrate type**

313 As discussed above, $HNO_3$ and $KNO_3$ undergo different renoxification processes on

314 the surface of $TiO_2$ under the same illumination conditions, suggesting that cations

315 bound to $NO_3^-$ significantly affect $NO_x$ production. Different types of cations coexist

316 with nitrate ions in atmospheric particulate matter, among which ammonium ions

317 ($NH_4^+$) are important water-soluble ions that can be higher in content than $K^+$ in urban

318 fine particulate matter (Zhou et al., 2016; Tang et al., 2021; Wang et al., 2021),

319 especially in heavily polluted cities.(Tian et al., 2020) Equal amounts of 4 wt.%

320 $NH_4NO_3$-$TiO_2$ particles were introduced into the chamber and illuminated under the

321 same conditions. HCHO had a much stronger positive effect on the release of $NO_x$

322 over $NH_4NO_3$-$TiO_2$ particles (Figure 3), which may be ascribed to $NH_4^+$. Combined

323 with the results of $NH_4NO_3$-$TiO_2$ particles and $KNO_3$-$TiO_2$ particles, it seems that the

324 affinity rather than electrostatic repulsion should be the primary effect of cations on

325 the production of $NO_x$. On substrates without photocatalytic activity such as $SiO_2$ and

326 $Al_2O_3$, $NH_4NO_3$ cannot generate $NO_x$,(Ma et al., 2021) such that $NO_x$ production

327 depends on the effect of $TiO_2$. The $h^+$ generated by $TiO_2$ excitation reacts with

328 adsorbed $H_2O$ to produce $\cdot OH$ (Eq. (9)), which gradually oxidizes $NH_4^+$ to $NO_3^-$ (Eq.





(10)). In our previous study, we demonstrated that irradiated $(NH_4)_2SO_4$-$TiO_2$ samples
had lower $NH_4^+$ and $NO_3^-$ peaks (Shang et al., 2017). Therefore, more $NO_3^-$
participated in the photocatalytic renoxification process via the
$NO_3^-$-$NO_3\cdot$-HCHO-$HNO_3$-$NO_x$ pathway to generate $NO_x$. Moreover,the results
without HCHO are shown in Figure 4a, both $NH_4NO_3$-$TiO_2$ particles and $KNO_3$-$TiO_2$
particles produced almost no $NO_x$, indicating the importance of HCHO for
renoxification to occur. Due to the high content of $NH_4NO_3$ in atmospheric particulate
matter, the positive effect of HCHO on the photocatalytic renoxification process may
have some impact on the concentrations of $NO_x$ and other atmospheric oxidants.

$$h^+ + H_2O \rightarrow \cdot OH \qquad (9)$$


$$\cdot OH + NH_4^+ / NH_3 \rightarrow NO_2^- \rightarrow NO_3^- \qquad (10)$$

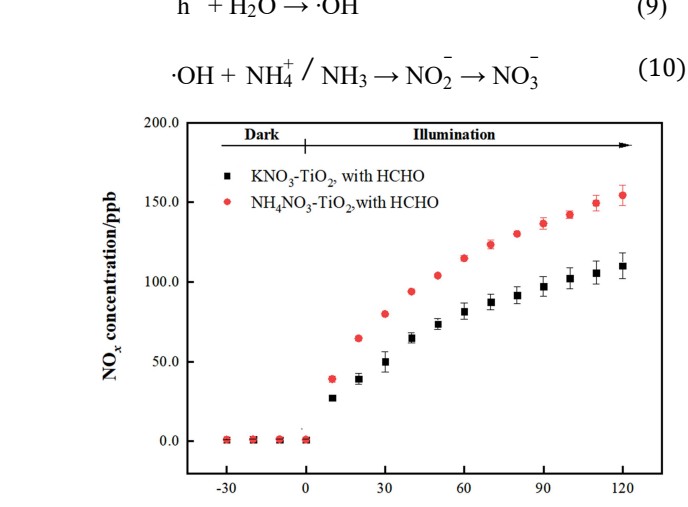


**Figure 3.** Effect of formaldehyde on the renoxification processes of 4 wt.%
$NH_4NO_3$-$TiO_2$ and 4 wt.% $KNO_3$-$TiO_2$ particles at 293 K and 0.8% of relative
humidity. 365 nm LED lamps were used during the irradiation experiment. The initial
concentration of HCHO was about 9 ppm.





### 3.3.2 The influence of nitrate content

Atmospheric particles have a wide range of nitrate content; differences in the relative

amounts of nitrate and $TiO_2$ in atmospheric particles may affect the renoxification

process. Therefore, we investigated the effects of nitrate concentration gradients on

renoxification. Changes in the $NO_x$ concentrations of $NO_3^-$-$TiO_2$ composite particles,

with or without HCHO, according to reaction time under 365 nm LED illumination

confirmed zero-order reaction kinetics. Therefore, we applied zero-order rate

constants to compare particles with different nitrate contents. For $KNO_3$-$TiO_2$, $NO_x$

was not generated in the absence of HCHO, even at high $NO_3^-$ nitrate concentrations

(Figure 4a) because no photolysis of either $NO_3^-$ or the $NO_3$ radical occurred under

nm LED illumination. For $NH_4NO_3$-$TiO_2$, the rate of $NO_x$ generation increased in

the absence of HCHO as $NH_4NO_3$ content increased, and at higher levels (80 and 95

wt.%), the $NO_x$ generation rate constant reached a plateau at ~$8.0 \times 10^{-2}$ ppb min$^{-1}$

because both $NH_4^+$ and NO are photochemically oxidized on $TiO_2$ to generate $NO_3^-$,

and part of this NO was oxidized to $NO_2$ by $O_2$.(Ma et al., 2021) Higher $NO_3^-$ content

leads to higher $NH_4^+$ concentration; thus, more $NH_4^+$ participated in the generation of

$NO_x$ through photooxidation. When $NO_3^-$ content reached 80 wt.% or higher, limited

$TiO_2$ content in the chamber led to the saturation of $NH_4^+$ photooxidation, preventing

further $NO_x$ generation. $NO_x$ release rates over $NO_3^-$-$TiO_2$ as nitrate content increased

in the presence of HCHO are shown in Figure 4b. The $NO_x$ production rate first

increased and then decreased, with a maximum of 4 wt.% nitrate content among both

$KNO_3$-$TiO_2$ and $NH_4NO_3$-$TiO_2$ particles. This increasing trend was caused by the



increased opportunities for contact between $TiO_2$ and $NO_3^-$ as nitrate content
increased, which facilitated the combination of $h^+$ with $NO_3^-$ to form $NO_3\cdot$. The trend
began to decrease when nitrate content exceeded 4 wt.%. Higher $NO_3^-$ content
hindered reactions on the surface of $TiO_2$, but rapidly decreased the Brunauer, Emmett
and Teller (BET) surface area of the composite particles (Shang et al., 2017), which
weakened HCHO uptake and particle surface reactions. The amount of $NO_x$ produced
by $NH_4NO_3$-$TiO_2$ was consistently higher than that of $KNO_3$-$TiO_2$. The possible
reasons for this difference are as follows. First, like the $K^+$ blocking effect discussed
in section 3.1, $NO_3\cdot$ generated from the reaction of $NO_3^-$ with $h^+$ was weakened; thus,
little adsorbed $HNO_3$ was available for further renoxification. Additionally, $NH_4^+$ can
undergo a photooxidation reaction to generate more $NO_x$ by $TiO_2$, as occurs in the
absence of HCHO.

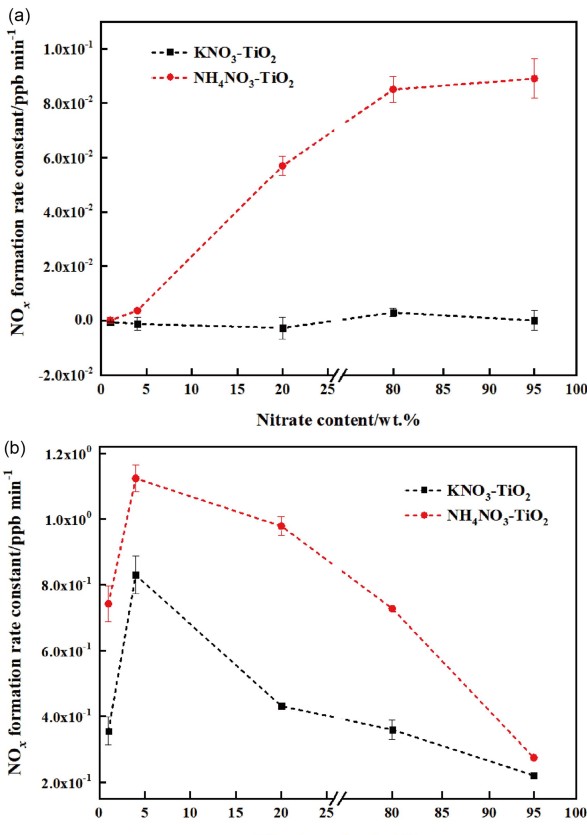


**Figure 4.** Effect of nitrate content (1 wt.%, 4 wt.%, 20 wt.%, 80 wt.% and 95 wt.%)

on the release of $NO_x$ for $NH_4NO_3$-$TiO_2$ and $KNO_3$-$TiO_2$ at 293 K and 0.8% of

relative humidity. 365 nm LED lamps were used during the illumination experiment.

(a) without HCHO; (b) the initial concentration of HCHO was about 9 ppm.

### 3.3.3 The influence of relative humidity

Water on particle surfaces can participate directly in the heterogeneous reaction

process. As shown in Eq. (9), $H_2O$ is captured by $h^+$ to generate ·OH with strong

oxidizability in photocatalytic reactions. The first-order photolysis rate constant of

$NO_3^-$ on $TiO_2$ particles decreases by an order of magnitude, from $(5.7 \pm 0.1) \times 10^{-4}$





$s^{-1}$ on dry surfaces to $(7.1 \pm 0.8) \times 10^{-5}$ $s^{-1}$ when nitrate is coadsorbed with water
above monolayer coverage (Ostaszewski et al., 2018). We explored the positive effect
of HCHO on the $NO_3^-$-$TiO_2$ particle photocatalytic renoxification at different RH
levels; the results are shown in Figure 5a. For $KNO_3$-$TiO_2$ particles, the rate of $NO_x$
production decreased as the RH of the environmental chamber increased, indicating
that increased water content in the gas phase hindered photocatalytic renoxification
for two reasons: $H_2O$ competes with $NO_3^-$ for $h^+$ on the surface of $TiO_2$ to
generate ·OH, reducing the generation of $NO_3$·, and competitive adsorption between
$H_2O$ and HCHO causes the generated ·OH to compete with $NO_3$· for HCHO,
hindering the formation of $HNO_3(ads)$ on particle surfaces. Moreover, it is also
possible that the loss of $NO_x$ on the wall increases under high humidity conditions,
resulting in a decrease in its concentration. This competitive process also occurs on
the surface of $NH_4NO_3$-$TiO_2$ particles, but at RH = 70%, the $NO_x$ generation rate
constant is slightly higher. The deliquescent humidity of $NH_4NO_3$ at 298 K is ~62%,
such that $NH_4NO_3$ had already deliquesced at RH = 70%, forming an
$NH_4^+/NH_3$-$NO_3^-$ liquid system on the particle surfaces. This quasi-liquid phase
improved the dispersion of $TiO_2$ in $NH_4NO_3$, resulting in greater $NO_x$ release. The
deliquescent humidity of $KNO_3$-$TiO_2$ was > 90%,(2009) such that no phase change
occurred at RH = 70%, and the renoxification reaction rate retained a downward trend.
In the presence of $H_2O$, in addition to the $NO_3^-$-$NO_3$·-HCHO-$HNO_3$ pathway
observed in this study, there are a variety of $HNO_3$ generation paths, such as the
hydrolysis of $N_2O_5$ via the $NO_2$-$N_2O_5$-$HNO_3$ pathway (Brown et al., 2005), the



oxidation of $NO_2$ by $\cdot OH$ (Burkholder et al., 1993), and the reaction of $NO_3\cdot$ with
$H_2O$ (Schutze and Herrmann, 2005), all of which require further consideration and
study.

414         The formation rates of NO and $NO_2$ are shown in Figure 5b and c, respectively.

$NO_2$ was the main product of surface $HNO_3$ photolysis. Under humid conditions,
generated $NO_2(ads)$ continued to react with $H_2O$ adsorbed on the surface to form
HONO(ads). HONO was desorbed from the surface and released into the gas phase
(Zhou et al., 2003; Bao et al., 2018; Pandit et al., 2021), providing gaseous HONO to
the reaction system. Because the $NO_x$ concentration remained high, the effect of
HONO on $NO_x$ analyzer results was negligible (Shi et al., 2021a). As $NO_2$ can form
$NO_2^-$ with $e^-$, a reverse reaction also occurred between $NO_2^-$ and HONO in the
presence of $H_2O$ (Ma et al., 2021; Garcia et al., 2021). Therefore, the increase in $H_2O$
increased the proportion of HONO in the nitrogen-containing products, such that the
$NO_x$ generation rate decreased as RH increased. Comparing Figure 5b and c shows
that, as RH increased, the NO production rate constant decreased more than that of
$NO_2$. HONO and $NO_2$ generated by the photolysis of $HNO_3(ads)$ decreased
accordingly, i.e., the NO source decreased. However, generated $NO_2$ and NO
underwent photocatalytic oxidation on the surface of $TiO_2$, and NO photodegradation
was more significant under the same conditions (Hot et al., 2017). Generally, a certain
amount of HONO will be generated during the reaction between HCHO and
$NO_3^-$-$TiO_2$ particles when RH is high, which affects the concentrations of
atmospheric $\cdot OH$, $NO_x$, and $O_3$. This process is more likely to occur in summer due to





high RH and light intensity affecting atmospheric oxidation. In drier winters or dusty
weather, when TiO$_2$ content is high, HCHO greatly promotes the photocatalytic
renoxification of NO$_3^-$-TiO$_2$ particles, thereby releasing more NO$_x$ into the
atmosphere, affecting the global atmospheric nitrogen budget. Thus, regardless of the
seasonal and regional changes, renoxification has significant practical importance.

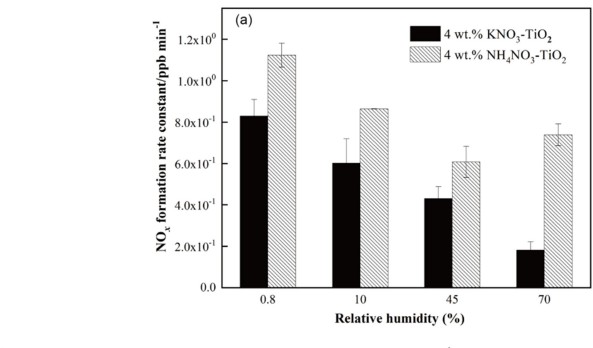

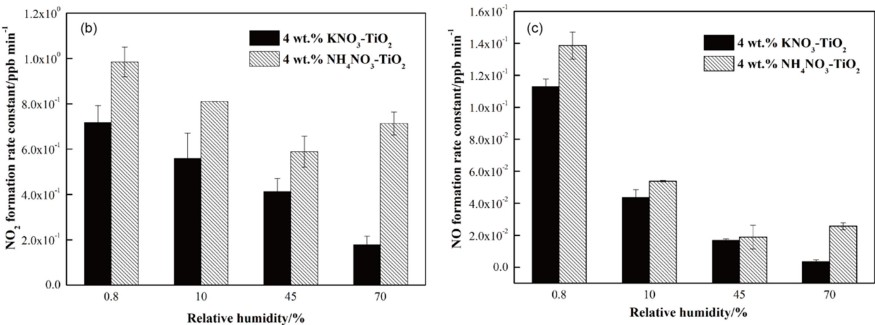


**Figure 5.** Effect of relative humidity on the release of NO$_x$ (a), NO$_2$ (b), NO (c) over 4
wt.% NH$_4$NO$_3$-TiO$_2$ and 4 wt.% KNO$_3$-TiO$_2$ particles at 293 K. 365 nm LED lamps
were used during the illumination experiment. The initial concentration of HCHO was

442                    about 9 ppm.

**3.3.4 The influence of initial HCHO concentration**
To explore whether HCHO promotes nitrate renoxification at natural concentration
levels, we reduced the initial concentration of HCHO in the environmental chamber



by a factor of 10, to ~1.0 ppm. The positive effect of HCHO on the photocatalytic
renoxification of $KNO_3$-$TiO_2$ particles was clearly weakened, with $NO_2$ concentration
first increasing and then decreasing, and NO concentration remaining stable (Figure
S10). The HCHO concentration decreased due to its consumption during the reaction,
making its positive effect decline quickly. The photocatalytic oxidation reaction
between $NO_x$ and photogenerated reactive oxygen species (ROS) on the $TiO_2$ surface
further decreased the $NO_x$ concentration. Photocatalytic oxidation of $NO_x$ by ROS on
$TiO_2$ particles occurred at an HCHO concentration of 9 ppm, but the positive effect of
HCHO remained dominant. Thus, no decrease in $NO_x$ concentration was observed
within 120 min in our experiments.

456       The concentration of HCHO in the atmosphere is relatively low, with a balance

between the photocatalytic oxidation decay of $NO_x$ and the release of $NO_x$ via
photocatalytic renoxification. The mutual transformation between particulate $NO_3^-$
and gaseous $NO_x$ is more complex. The effect of low-concentration HCHO on the
renoxification of $NO_3^-$-$TiO_2$ particles requires further investigation. However, many
types of organics provide hydrogen atoms in the atmosphere, including alkanes (e.g.,
methane and n-hexane), aldehydes (e.g., acetaldehyde), alcohols (e.g., methanol and
ethanol), and aromatic compounds (e.g., phenol) that react with $NO_3\cdot$ to produce nitric
acid (Atkinson, 1991). These organics, together with HCHO, play similar positive
roles in photocatalytic renoxification and, therefore, influence $NO_x$ concentrations.
**4 Atmospheric implications**

467       Nitric acid and nitrate are not only the final sink of $NO_x$ in the atmosphere but



are also among its important sources. $NO_x$ from nitrate through renoxification is easily
overlooked. The renoxification of nitrate on the surface of $TiO_2$ particles can be
divided into photolytic renoxification and photocatalytic renoxification. The
photocatalytic performance of $TiO_2$ promotes the renoxification process, which
explains the influence of semiconducting metal oxide components on atmospheric
mineral particles during the renoxification of nitrate. Although most previous studies
have focused on solid-phase nitrate renoxification, our exploration of the roles of
HCHO in this study will allow us to examine complex real-world pollution scenarios,
in which multiple atmospheric pollutants coexist, as well as the effects of organic
pollutants on the renoxification process. Atmospheric HCHO is taken up at the
surface of particulate matter, accounting for up to ~50% of its absorption (Li et al.,
2014), such that the heterogeneous participation of HCHO during renoxification is
important. This study is the first to report that HCHO has a positive effect on the
photocatalytic renoxification of nitrate on $TiO_2$ particles, via the
$NO_3^- \text{-} NO_3 \cdot \text{-HCHO-HNO}_3 \text{-} NO_x$ pathway (Figure 6), further increasing the release of
$NO_x$ and other nitrogen-containing active species, which in turn affects the
photochemical cycle of $HO_x$ radicals in the atmosphere and the formation of
important atmospheric oxidants such as $O_3$. Factors such as particulate matter
composition, RH, and initial HCHO concentration all influence the positive effect of
HCHO; notably, $H_2O$ competes with $NO_3^-$ for photogenerated holes. Based on these
findings, two balance systems should be explored in depth: the influence of RH on the
generation rates of HONO and $NO_x$, as water increases the proportion of HONO in


490 nitrogen-containing products; and the balance between the photocatalytic degradation

491 of generated $NO_x$ on $TiO_2$ particles and the positive effect of HCHO on $NO_x$

492 generation at low HCHO concentrations.

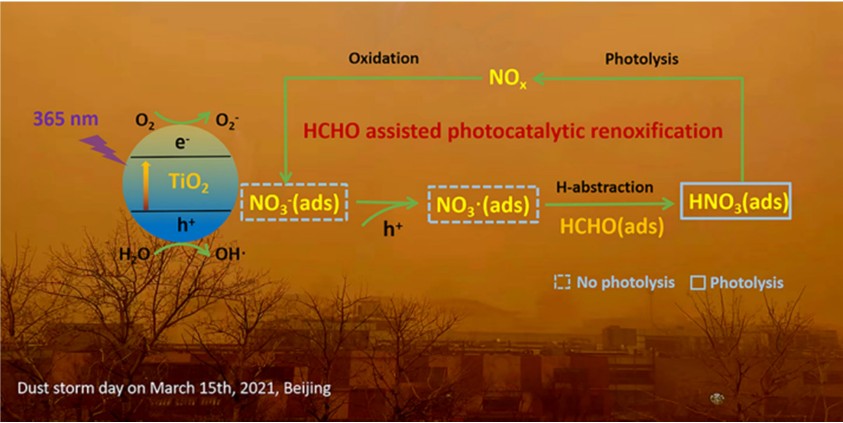

494 **Figure 6.** Positive role of HCHO on the photocatalytic renoxification of nitrate-$TiO_2$

495 composite particles via the $NO_3^-$-$NO_3\cdot$-HCHO-$HNO_3$-$NO_x$ pathway.

496 Based on our results, we conclude that in photochemical processes on the

497 surfaces of particles containing semiconductor oxides, with the participation of

498 hydrogen donor organics, a significant synergistic photocatalytic renoxification

499 enhancement effect alters the composition of surface nitrogenous species via the

500 $NO_3^-$-$NO_3\cdot$-hydrogen donor-$HNO_3$-$NO_x$ pathway, thereby affecting atmospheric

501 oxidation and nitrogen cycling. The positive effect of HCHO can be extended from

502 $TiO_2$ in this study to other components of mineral dust such as $Fe_2O_3$ and ZnO with

503 photocatalytic activity, which may have practical applications. Our proposed reaction

504 mechanism by which HCHO promotes photocatalytic renoxification will improve

505 existing atmospheric chemistry models and reduce discrepancies between model

506 simulations and field observations.



507

*Supplement.*

509 Detailed information of Figures S1-10 (which include the spectra of the lamps,

510 size distribution of TiO$_2$ particles and changes of HCHO concentration in

511 environmental chamber, changes of NO$_x$ concentration under different reaction

512 conditions, photodegradation curve of HCHO, ESR spectra of TiO$_2$ and ATD

513 particles), and Table S1 (which demonstrate ATD chemical composition) .

514

*Acknowledgments*

516 The authors are grateful to the financial support provided by National Natural

517 Science Foundation of China (Nos. 21876003, 41961134034 and 21277004), the

518 Second Tibetan Plateau Scientific Expedition and Research (No. 2019QZKK0607),

519 and the 111 Project Urban Air Pollution and Health Effects (B20009).

520

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
