# Peer review of "The Positive Effect of Formaldehyde on the Photocatalytic"

_Atmospheric Chemistry and Physics, 2022_

## Author Comment (AC1)

**Response to the reviewer 1's comments:**

**Referee #1: General Evaluation**

*This manuscript investigated the effect of HCHO on the photocatalytic renoxification of nitrate on TiO₂ particles. The investigated system is interesting. However, the experimental design has so many defects. More experiments and verification are needed to support the conclusion.*

**Response:**

Thanks for your comments, which will be all valuable and very helpful for revising and improving the manuscript, as well as the important guiding significance to our researches. We have thought deeply about the experimental design, and answer the comments point by point.

**Comments on Preprint acp-2022-6:**

*Major comments:*

*1.Methods: Line 103: 400 L chamber is usually not enough for the investigation of heterogeneous reactions. Besides, only 250 L air was injected into it, which would increase the wall effect of chamber. Line 105-106: How did the author control the chamber temperature? It is well known that the chamber temperature will increase when turn on the lights. Line 111-112: the light intensities for the tube and LED lamp were different, so how to compare their results? Why was only the results in 3.1 obtained under the irradiation of tube lamps? What was the meaning for introducing two kinds of lamps in the smog chamber?*

**Response:**

Thanks for your comments, which will be all valuable and very helpful for revising and improving the manuscript, as well as the important guiding significance to our researches. We have thought deeply about the experimental design, and answer the comments point by point.

*(1)Line 103: 400 L chamber is usually not enough for the investigation of heterogeneous reactions. Besides, only 250 L air was injected into it, which would increase the wall effect of chamber.*

**Response:**

The environmental chamber can be used to assess and predict the formation of secondary pollutants. Large volume chamber is helpful for obtaining chemical transformation laws closer to the real atmosphere, although small volume chambers are still used in laboratory dynamics simulation studies due to their economic and flexible features. For example, Shi et al (EST, 2021, 55: 854-861) studied renoxification of suspended submicron particulate sodium and ammonium nitrate through controlled laboratory photolysis experiments using an 150 L Teflon environmental chamber. The reaction system is very similar as ours, focusing on $NO_x$ release from particulate inorganic nitrate with well-characterized light conditions. Another example is Jia et al (Aerosol Sci. Tech., 2014, 48: 1-12), who studied the formation of ozone and secondary organic aerosol (SOA) from benzene–$NO_x$ and ethylbenzene–$NO_x$ irradiations in a 350 L Teflon chamber. The major substances in SOA were determined to be carboxylic acids and glyoxal hydrates. And it was found that the aqueous radical reactions and the hydration from glyoxal can be enhanced under high RH conditions, which can irreversibly enhance the formation of SOA from both benzene and ethylbenzene. In another research, a 151 L chamber was used to study the kinetic of ozone decomposition on luminescent oxide surfaces (Chen et al., J. Phys. Chem. A, 2011, 115: 11979-1198). It is also a simulation research about heterogeneous photochemistry of ozone over mineral dust aerosol, including $\alpha$-$Fe_2O_3$, $TiO_2$, and $\alpha$-$Al_2O_3$. The rate and extent of ozone decomposition on these oxide surfaces are found to be a function of the nature of the surface as well as the presence of light and relative humidity, with $TiO_2$ is active toward $O_3$ decomposition upon irradiation. Therefore, it is true that 400 L chamber is not so big but can in some extent reflect the simulation results of heterogeneous reaction.

We tried our best to decrease the effects of "wall effect" on the experiment results by two ways. One was to conduct the experiments when the particle size distribution got stable, that is, 60 min after the injection of the particles. As can be seen in Fig S2 of the original manuscript, the size distribution gets stable after 30 min and can sustain for several hours. As stated in text line 166-168 of the original

manuscript: "*After the concentrations of both HCHO and aerosol became stable, the lamps were turned on and the concentrations of NOx were monitored.*" Another way was that we strictly carried out blank experiments and all data in the study were subtracted from the corresponding blank data to ensure the reliability. For example, to determine the background value of $NO_x$ in the reaction system, four blank experiments were carried out under illumination without nitrate: "synthetic air", "synthetic air + $TiO_2$", "synthetic air + HCHO" and "synthetic air + HCHO + $TiO_2$". In the blank experiments of "synthetic air" and "synthetic air + $TiO_2$", the $NO_x$ concentration remained stable during 180 min illumination, and the concentration change was no more than 0.5 ppb (Figure S4a of the original manuscript). Therefore, the environmental chamber was thought to be relatively clean, and there was no generation and accumulation of $NO_x$ under illumination. In addition, the chamber was cleaned after each experiment and the repeatability was ensured. To make it clear, we added one sentence to the section 2.3 of the revised manuscript: "For the chamber operation, we completely evacuated the chamber after every experiment, then cleaned the chamber walls with deionized water and then dried by flushing the chamber with ultra-zero air to remove any particles or gases collected on the chamber walls."

Overall, our experiments were conducted under relatively stable conditions, thus to exclude the effects of "wall effect" and the data were reliable.

*(2)Line 105-106: How did the author control the chamber temperature? It is well known that the chamber temperature will increase when turn on the lights.*

**Response:**

As stated in line 108-111 of the original manuscript: "*One is a set of tube lamps with a main spectrum of 320-400 nm and a small amount of 480-600 nm visible light (Figure S1a). The other is a set of Light-emitting diode (LED) lamps with a narrow main spectrum of 350-390 nm (Figure S1b)*", the light sources we used emit no infrared lights, and in case 36 W and 3 W of the tube and LED lamps were used, the temperature in the chamber is not easy to be heated. Figure S1b were shown below (Figure 1 here) with the wavelength distribution. We added watt values of the lamps in the section 2.1 of the revised manuscript and above sentence is now as follows:

*"One is a set of 36 W tube lamps with a main spectrum of 320-400 nm and a small amount of 480-600 nm visible light (Figure S1a). The other is a set of 12 W Light-emitting diode (LED) lamps with a narrow main spectrum of 350-390 nm (Figure S1b)"*. In addition, we controlled the temperature of the room where the chamber is, keeping it at 20 degrees all the time. So we consider the effect of temperature to be negligible in this study.

[Figure]

[Figure]

Figure 1. Spectral energy distribution of (a) 365 nm tube lamps and (b) 365 nm LED lamps.

*(3)Line 111-112: the light intensities for the tube and LED lamp were different, so how to compare their results? Why was only the results in 3.1 obtained under the irradiation of tube lamps? What was the meaning for introducing two kinds of lamps in the smog chamber?*

**Response:**

The reaction systems and aims of the two kinds of light sources were different, which were summarized in the following Table 1. The results of tube lamp and LED lamp would not be compared, but had their own purpose, individually.

For tube lamp (contains both ultraviolet and visible light) system in section 3.1, there is no HCHO but only nitrate with or without $TiO_2$ ($KNO_3$-$SiO_2$ or $KNO_3$-$TiO_2$/$SiO_2$). The aim is to prove that $NO_3$ radical can be produced via the reaction between excited $TiO_2$ and nitrate, with its photolysis under visible light producing $NO_x$.

For LED lamp (contains only ultraviolet) system in section 3.2, HCHO was introduced and the particles are $KNO_3$-$TiO_2$ or $HNO_3$-$TiO_2$. The aim is to avoid the

photolysis of $NO_3$ radicals under visible light, but to see if the produced $NO_3$ radicals can be reacted with HCHO. As stated in line 226-229 of the original manuscript: "*Therefore, the LED lamp setup was used in subsequent experiments to exclude the direct photolysis of both $KNO_3$ and $NO_3$·, but allow the excitation of $TiO_2$. This approach allowed us to investigate the process of photocatalytic renoxification caused by HCHO in the presence of photogenerated $NO_3$·.*" In this way, the formation of $NO_x$ can be attributed to the photolysis of $HNO_3$, coming from the possible hydrogen-abstraction reaction of HCHO with $NO_3$ radicals, as discussed in the later part of section 3.2.

Therefore, the design of our experiments is: (1) Section 3.1 using tube lamp to prove that $NO_x$ is not coming from $KNO_3$ UV-light photolysis, but from the visible-light photolysis of $NO_3$ radical. By this way to prove the photocatalytic effect of $TiO_2$ on "renoxification" process, that is "photocatalytic renoxification"; (2) Section 3.2 using LED lamp to ensure no photolysis of both $KNO_3$ and $NO_3$ radical, and in this case to investigate the effect of HCHO on the release of $NO_x$ in the presence of $NO_3$ radical, i.e., the effect of HCHO on the "photocatalytic renoxification". In order to make the readers understand clearer, we rewrote the abstract according to this logic and defined the term "photocatalytic renoxification".

Table 1. A comparison of the two light sources used.

| Light source and Wavelength range | Section | System | Phenomenon | Contents/Implications |
|---|---|---|---|---|
| TUBE LAMP, 320-400 nm with small amount of 480-600 nm | 3.1 | 4 wt.% $KNO_3$-$SiO_2$ (NO HCHO) | No NOx was released | 320-400 nm irradiation cannot make nitrate photolyze, so to exclude the photolysis source of NOx from nitrate. [Fig 1] |
| | | 4 wt.% $KNO_3$-$TiO_2$ (1 wt.%)/$SiO_2$ (NO HCHO) | NOx release was observed | $TiO_2$ was composited to $SiO_2$, to see the effect of $TiO_2$ along with nitrate. 320-400 nm irradiation can excite $TiO_2$ to generate electron-hole pairs. Nitrate can react with photogenerated holes to produce $NO_3$ radicals 480-600 nm irradiation can make $NO_3$ radical photolyze, generating NOx Above explanation can be seen in section of 3.1 and the equations 1-4 in the original manuscript. |

| | | | NOx release was observed | ☐350-390 nm irradiation cannot make nitrate photolyze, so to exclude the photolysis source of NOx from nitrate. [Fig S5] |
| LED LAMP, 350-390 nm | 3.2 | (WITH HCHO) 4 wt.% nitrate-TiO₂ | | ☐350-390 nm irradiation can excite TiO₂ to generate electron-hole pairs, so as to generate NO₃ radical. |
| | | | | ☐350-390 nm irradiation cannot make NO₃ radical photolyze. |
| | | | | ☐NO₃ radical can react with HCHO to produce HNO₃(ads) |
| | | | | ☐The observed NOx comes from the photolysis of HNO₃(ads). |

☐Section 3.1 proves that NO₃ radical can be produced in the condition of "nitrate+ TiO₂ (excited by the UV light of the tube lamp)", then NO₃ radical undergo photolysis (under visible light of the tube lamp) to produce NOx.

☐In order to avoid the photolysis of NO₃ under visible light, no tube light containing visible light was used any more in section 3.2.

☐The aim of 3.2 is to investigate whether the produced NO₃ radical can be captured by HCHO.

This explanation can be seen in the first paragraph in section 3.2, that is Line 219-229 of the original manuscript.

☐This explanation can be seen in section 3.2, including Fig 2 and equations 5-8.

*2.Important defect of this article is the composition of the mixture in the part of "2.2 nitrate-TiO₂ composite samples": "TiO₂ was simply mixed in nitrate solutions at the desired mass mixing ratio to obtain a mash. The mash was dried at 90℃ and then ground carefully to ensure a uniform composite of particles." How did the author ensure that the particles are uniform composite of nitrate and TiO₂? Did the author do some experiments to confirm these? For example, in the reference of Ma et al (EST, 8604-8612, 2021), the nitrate and TiO₂ mixture was dripped onto a quartz tube inner all, then images and Raman spectra of single composition and mixture were analyzed, and mixture were confirmed to form. However, in this work, the generation method of mixture particles is different from that of Ma's work, and these mixture particles are sprayed by synthetic air into PVF bag. No experiments have been given to confirm the composition of the mixture particles in the chamber. In my opinion, this method can't generate a uniform composite of nitrate and TiO₂!!! The composition and the nitrate content are the most important quantitative method factors of all the experiments in the article. If the composition and nitrate content can't be control, how*

*to compare the NOx concentration in different experiments? Then, all the results are not convincing!!!*

**Response:**

Thanks for your comments, which will be all valuable and very helpful for revising and improving the manuscript, as well as the important guiding significance to our researches. We prepared the composite particles carefully and ensured its homogeneity. During the preparation of nitrate and $TiO_2$ composite samples, we used a very small amount of nitrate solvent, which is of 2 mL. With a relatively large specific surface area (~54.28 $m^2/g$) and a large amount (250 mg) of $TiO_2$, the mixture was viscous and then quickly dried at 90°C, followed by a thorough grind for 30 min. So, nearly no loss of $TiO_2$ due to no use of filtration, and nearly no loss of nitrate pyrolysis due to low temperature of drying. Concerning the work of Ma et al (EST, 2021, 8604-8612) mentioned by the reviewer, the nitrate and oxides were mixed by dispersing a total mass of 1.0 g of oxide powder and 0.02 g of nitrate in 50.0 mL of ultrapure water. The measured nitrate loading percent (1.9-2.0 wt%) was very closed to theoretical value (2.0 wt%). This preparation method was very similar to ours in that both $TiO_2$ powders were dissolved in a nitrate solution. So the composites of nitrate and $TiO_2$ we obtained by our preparation method are thought to be uniform.

In addition, we used diffuse reflectance fourier transform infrared spectroscopy (DRIFTS) to characterized the structure of the particles, with the results approving the homogeneity. DRIFTS spectra of $KNO_3$-$TiO_2$ particles with different contents of nitrate as well as the $KNO_3$ particle were shown in the following Figure 2. It can be seen that IR spectra of the composited particles with $KNO_3$ contents higher than 1 wt% were very close to that of $KNO_3$. According to Aghazadeh's (J Ultrafine Grained Nanostruct Mater, 2016, 49(2): 80-86) and Maeda's (Applied Catalysis B: Environmental, 2011, 103(1-2), 154) work, 1760 $cm^{-1}$ are the vibrating peaks of nitrate. The ratios of the peak area from 1730-1790 $cm^{-1}$ for 1, 4, 32, 80 wt.% composited samples is 1: 4.1: 29.8: 81.6, which is very close to that of theoretical value, proving that the samples were uniformly mixed. This DRIFTS figure has been

added as Figure S2 in *Supplement* of the revised manuscript, with its description added in section 2.2 of the revised manuscript.

[Figure]

Figure 2. DRIFTS spectra of $TiO_2$ particles compounded with different mass fractions of $KNO_3$.

*3. Another important defect of this article is the quantitative method of NOx concentration. As shown in Ma's work, they used the normalized concentration (ppb/mg) to quantify NOx. However, this work just used the NOx concentration (ppb) to compare different experiments, which meant that if more reactants were added in the chamber, the generated NOx concentration would be higher. The initial mass concentration of particles was 300 mg/m³ (75mg/250L), and the concentration of HCHO was 10 ppm, which were much higher than that in the real environment and resulted in that the obtained results could not be directly used for an analogy to real environment. The results with ppb as unit are meaningless to reflect their influence in the real atmosphere. Were the particles kept the same in different experiments during the reaction? The author mentioned that the wall loss of particle in the smog chamber was very high at the beginning. And the wall loss for different kinds of particles and for the same kind of particles in different experiment (maybe affected by the conditions of the smog chamber wall) should be different. How did the author ensure*

*that the particle distributions were the same in different experiments when turned on the light? Besides, the surface area, as an important factor in heterogeneous reactions, has not been detected in the experiments. Different surface areas directly affect the irradiation surface of $TiO_2$, the uptake of HCHO and the release of NOx. The missing information of surface area would result in the large uncertainties in the experiments. At least, the authors should give a normalized NOx concentration, then different experiments can compare with each other and give the reasonable results and reflect the influence in the real environment.*

**Response:**

Thanks for your comments, which will be all valuable and very helpful for revising and improving the manuscript, as well as the important guiding significance to our researches. We have thought deeply about the experimental design, and answer the comments point by point.

*(1)Another important defect of this article is the quantitative method of NOx concentration. As shown in Ma's work, they used the normalized concentration (ppb/mg) to quantify NOx. However, this work just used the NOx concentration (ppb) to compare different experiments, which meant that if more reactants were added in the chamber, the generated NOx concentration would be higher.*

**Response:**

For flow tube experiments, the flow tube was usually weighted before and after its loading with samples and the normalized concentration (ppb/mg) was used for a better comparison between different samples, as what has been done as Ma's work. However, in our experiments, the amount of the different samples sprayed into the chamber is same, so the mass normalization is not necessary. That is, no matter whether it is expressed as ppb/mg or ppb, the same trend will be obtained, which will not affect the conclusions of our study.

*(2)The initial mass concentration of particles was 300 mg/m³ (75mg/250L), and the concentration of HCHO was 10 ppm, which were much higher than that in the real environment and resulted in that the obtained results could not be directly used for an*

*analogy to real environment. The results with ppb as unit are meaningless to reflect their influence in the real atmosphere.*

**Response:**

The value of 75mg/250L is the amount we injected into the chamber, but according to the size distribution measurement, the real suspended particle concentration was not that high. As shown in Figure S2 of the original manuscript, the number concentration of $TiO_2$ particles is about 8500 particle/$cm^{-3}$ after reaching stability. This level of number concentration was observed by Wang et al (Environ. Chem., 2015, 34(9): 1619-1626) who measured the particle size distribution of atmospheric particulate matter number concentrations in Nanjing in August 2013, with the particle number concentration of about 8000 particle/$cm^{-3}$. As stated in line 156-158 of the original manuscript: " *The size distribution of $TiO_2$ reached stable after about 60 min with the peak particle size was about 120 nm, similar to that of atmospheric particles in some urban areas in China (Wang et al., 2015; Li et al., 2019).*" We checked our size distribution data of different samples, and found that the number concentration is not that high and usually around 4000 particle/$cm^{-3}$ when reaching stability, with the figures shown below (Figures 3-5 in this file). So, the Figure S2 in the original manuscript was deleted and replaced by the following Figure 3 (Figure S3 in the revised supplement).

[Figure]

Figure 3. Changes of particle size distribution of 4 wt.% $KNO_3$-$TiO_2$ particles in environmental chamber with time.

We admit that 10 ppm of HCHO is too high, so we also performed experiments with low concentrations of HCHO (1 ppm), as described in section 3.3.4 of the manuscript. The positive effect of HCHO on the photocatalytic renoxification of $KNO_3$-$TiO_2$ particles was still observed, with $NO_2$ concentration first increasing and then decreasing (Figure S10 of the original manuscript). Atmospheric formaldehyde concentrations are generally very low. However, in cities with high traffic density, because combustion produces emissions, formaldehyde concentrations will be much higher than normal. In the indoor environment, formaldehyde levels can increase due to smoking, emissions from gas stoves and furniture, and can reach up to around 0.4 ppm (Formaldehyde. In: Wood dust and formaldehyde. Lyon, International Agency for Research on Cancer, 1995, 217-362). So, we assume that the positive effect of HCHO on the renoxification may still exist at some specific situation with its high concentration, which requires further investigation. To make the presentation of our results more accurate, we have added this sentence in the section 4 of the revised manuscript: "*Although in the case of high concentrations of HCHO in our experiment, the response to the real situation will be biased, the results of this study illustrate a possible way of HCHO in influencing nitrate renoxification in the atmosphere.*"

*(3)Were the particles kept the same in different experiments during the reaction? The author mentioned that the wall loss of particle in the smog chamber was very high at the beginning. And the wall loss for different kinds of particles and for the same kind of particles in different experiment (maybe affected by the conditions of the smog chamber wall) should be different. How did the author ensure that the particle distributions were the same in different experiments when turned on the light?*
**Response:**

Thanks for the reviewer's thoughtful question. As mentioned before, we deflated and cleaned the chamber for each experiment, and the operation of each experiment was almost identical. So the particle number size distribution of the same kind of particles would be quite same, which can be proved by the following Figures 4 and 5.

The operation sequence of the experiment is as follows: HCHO was introduced firstly, and wait 60 min for its stability; then the particle was introduced instantly, and need 30 min for its stable, and another 30 min for HCHO's second stability; then the lights were opened. The experiment operation has been rewritten in the revised manuscript.

The left picture in the following Figure 4 shows the first 60 min of the size distribution in the dark, and the right picture of the size distribution during the irradiation time. Figure 5 is another batch experiment of $TiO_2$ in the chamber with the same operation, and it shows very similar size distribution from 0-60 mins and 120-180 mins.

[Figure]

Figure 4. Changes of particle size distribution of $TiO_2$ particles in environmental chamber with time. Left: Before irradiation; Right: After irradiation.

[Figure]

Figure 5. Changes of particle size distribution of $TiO_2$ particles in environmental chamber with time (Another batch experiment).

As for different kinds of particles, as comparison of Figure 3 with Figures 4 and 5, both main particle size (about 120 nm) and particle number concentration (about 4000 particle/cm$^{-3}$) are similar. Therefore, the particles can be kept the same in different experiments during the reaction, not only for the same particles but also for different kinds of particles. This is because of the same operation and the similar surface area of different kinds of particles (same loading of nitrate) as will be mentioned below.

*(4)Besides, the surface area, as an important factor in heterogeneous reactions, has not been detected in the experiments. Different surface areas directly affect the irradiation surface of TiO$_2$, the uptake of HCHO and the release of NOx. The missing information of surface area would result in the large uncertainties in the experiments. At least, the authors should give a normalized NOx concentration, then different experiments can compare with each other and give the reasonable results and reflect the influence in the real environment.*

**Response:**

We agree with the reviewer that surface area will affect the reaction process. We once measured BET of the KNO$_3$-TiO$_2$ particles with the results shown in Table 2. It can be seen that the BET values gradually decrease as the nitrate loading increases. As suggested by the reviewer, the surface area normalized reaction rate should be used to compare the particles. It is a regret that we did not measure the BET of NH$_4$NO$_3$-TiO$_2$ particles, so we cannot get the normalized parameter for NH$_4$NO$_3$-TiO$_2$ particles. Therefore, for sake of the reliability of the results, we deleted the 3.3.2 section "the influence of nitrate content" of different kinds of nitrate. Except for section 3.3.2, the loadings were all 4 wt.% of nitrate. According to Table 2, the BET surface areas of the particles did not change much at 4 wt.% loading, which were mainly dependent on the specific surface area of the main body of TiO$_2$. So the estimated difference in BET surface area of TiO$_2$ loaded with different nitrates at 4 wt.% loading is not significant and has little effect on the reaction results.

Table 2. BET surface area of composite particles with different $KNO_3$ content.

| $KNO_3$ content in composite particles wt.% | BET $m^2/g$ |
|---|---|
| 0 | 54.28 |
| 1 | 50.7 |
| 4 | 48.04 |
| 12 | 41.77 |
| 20 | 36.86 |
| 32 | 26.67 |
| 50 | 18.45 |
| 80 | 5.61 |

*4. Gas HCHO and mixture particles of $TiO_2$ and nitrate were contained in the system. Although some controlled experiments were conducted, the role of $TiO_2$ and HCHO still could not be isolated. A series of important experiments such as HCHO and single nitrate particles under irradiations are needed.*

**Response:**

Thanks for your comments, which will be all valuable and very helpful for revising and improving the manuscript, as well as the important guiding significance to our researches. We agree with the reviewer that controlled experiments are needed not only for "$TiO_2$ + HCHO" system, but also for "HCHO+nitrate" system, which have both been conducted in our study. Figure S6 in the original manuscript presented the HCHO decay with irradiated $TiO_2$, indicating the photocatalytic role of $TiO_2$. For "HCHO+nitrate" system, because the nitrates in our study were all loaded on particles, we composited nitrate with inert $SiO_2$. As shown in Figure 6 below, when only $KNO_3$-$SiO_2$ was present in the chamber (without HCHO), $NO_x$ concentrations fluctuated within the instrumental measurement error (0.5 ppb) both in the dark and under 365 nm LED illumination. When HCHO occurred in the chamber with $KNO_3$-$SiO_2$ particles, no $NO_x$ production was observed in the dark and under 365 nm LED illumination, indicating that HCHO could not be oxidized on the surface of

non-photocatalytically active particles. We have added this figure as Figure S9 in the revised supplement with its explanation in section 3.2.

[Figure]

Figure 6. Effect of formaldehyde on the renoxification processes of 4 wt.% KNO$_3$-SiO$_2$ particles at 293 K and 0.8% of relative humidity. 365 nm LED lamps were used during the irradiation experiment. The initial concentration of HCHO was about 9 ppm.

*5.All the proposed mechanisms couldn't be well supported only by the changes of NOx concentration. This work and Ma's work indicate HONO, HNO$_3$, NO$_3$ radical, NOx could form in these reaction systems. However, HONO, HNO$_3$, NO$_3$ radical could lead the overestimation of NO$_2$ concentrations by chemiluminescence method. How did the authors exclude the effect of these species? Besides, most important products such as NO$_3$, HNO$_3$, HONO were not detected in the experiments except OH radical. How did the authors make sure that the reaction pathway followed the proposed mechanisms? It is well known that TiO$_2$ can photocatalysis HCHO, can this reaction affect the formation of NOx?*

**Response:**

Thanks for your comments, which will be all valuable and very helpful for revising and improving the manuscript, as well as the important guiding significance to our researches. We have thought deeply about the experimental design, and answer the comments point by point.

*(1)All the proposed mechanisms couldn't be well supported only by the changes of NOx concentration. This work and Ma's work indicate HONO, HNO₃, NO₃ radical, NOx could form in these reaction systems. However, HONO, HNO₃, NO₃ radical could lead the overestimation of NO₂ concentrations by chemiluminescence method. How did the authors exclude the effect of these species? Besides, most important products such as NO₃, HNO₃, HONO were not detected in the experiments except OH radical. How did the authors make sure that the reaction pathway followed the proposed mechanisms?*

**Response:**

It is a regret that we did not detect the formation of HONO, $HNO_3$ and $NO_3$ radical due to technique limitation. The effects of these compounds on NOx measurement has been discussed in section 3.2 of the original manuscript. Most of our experiments were conducted in dry condition (0.8% RH), and according to Zhou et al (Geophys. Res. Lett., 2003, 30, 10.1029/2003gl018620), the rate of $NO_x$ generation from $HNO_3$ photolysis was greater than 97% of the total product at RH=0%. So the formation of HONO in our study was estimated to be very low. For larger RH conditions as discussed in section 3.3.3 of the original manuscript, HONO(ads) can be generated due to the reaction of $NO_2$ with adsorbed $H_2O$, which can be desorbed from the surface and released into the gas phase. While according to Shi et al (Environ. Sci. Technol., 55, 854-861,2021), the effect of HONO on $NO_x$ analyzer measurements can be neglected in case of high $NO_x$ concentration in the system. The $NO_x$ concentration in our study in most cases is about 100 ppb, so we think the effect of other products on product distribution and $NO_x$ measurements was negligible

In response to the speculation of $HNO_3$ production, we measured the pH of water extracts in $NO_3^-$-$TiO_2$ systems with and without HCHO, and found that pH was greatly reduced in the presence of HCHO (Figure 7 below). The pH decreases by 1.7% and 2.1% for $KNO_3$-$TiO_2$ and $NH_4NO_3$-$TiO_2$ particles, respectively, suggesting the formation of acidic species such as $HNO_3$(ads) in this study. We have added this results in the revised manuscript as it appears in the section 3.2: "*Next, we measured the pH of water extracts in $NO_3^-$-$TiO_2$ systems with and without HCHO. It was found*

*that for KNO₃-TiO₂ and NH₄NO₃-TiO₂ particles, the pH decreased by 1.7% and 2.1%, respectively, suggesting the formation of acidic species such as HNO₃(ads) in this study.*"

[Figure]

Figure 7. pH values of water extract of $KNO_3$-$TiO_2$ and $NH_4NO_3$-$TiO_2$ particles in the chamber with or without HCHO under 365 nm LED lamps illumination at 293 K and 0.8% of relative humidity.

The generation of $NO_3$ radicals can be indirectly proved by the results in section 3.1 and Figure 1 of the original manuscript, as we have responded to comment # 1 (3) and displayed in above Table 1. Another similar example is the published work of our group (Sci. Rep., 2017, 7, 1161). By using the same chamber, the photoreaction rate constants of HCHO on $TiO_2$ and $KNO_3$-$TiO_2$ aerosols under "365 nm lamp" or "365 nm lamp + yellow fluorescence lamp (450–750 nm)" illumination were compared (Figure 8 below). The oxidation rate constants of HCHO over $TiO_2$ were comparable under these two illumination conditions, due to that $TiO_2$ is not sensitive to visible light. However, the rate constant on $KNO_3$-$TiO_2$ aerosol under illumination of both lamps was lower than that under only the "365 nm lamp", indicating a reduced oxidation rate due to $NO_3$ radical photolysis by visible light. This provides experimental evidence for the existence of $NO_3$ radical.

As we have mentioned in the section 3.3.3 of the original manuscript, in the presence of $H_2O$, in addition to the suggested $NO_3^-$-$NO_3\cdot$-HCHO-$HNO_3$ pathway,

there are a variety of $HNO_3$ generation paths, such as the hydrolysis of $N_2O_5$ via the $NO_2$-$N_2O_5$-$HNO_3$ pathway, the oxidation of $NO_2$ by $\cdot OH$, and the reaction of $NO_3\cdot$ with $H_2O$, all of which require further consideration and study. It is a regret that due to the technique limitations, we did not detect these species directly. We will dedicate to detect these species by some instruments in the future.

[Figure]

Figure 8. Photoreaction rate constants with light illumination. HCHO photoreaction rate constants on $TiO_2$ or 4 wt.% $KNO_3$-$TiO_2$ aerosol in the condition of 8% RH under light illumination of "365 nm" or "365 nm + yellow fluorescence", respectively. (Sci. Rep., 2017, 7, 1161)

*(2)It is well known that TiO₂ can photocatalysis HCHO, can this reaction affect the formation of NOx?*

**Response:**

Yes, HCHO can be degraded in the presence of irradiated $TiO_2$ and will affect the release of $NO_x$, which had been discussed in the original manuscript. We observed the decrease of HCHO concentration both in "$TiO_2$+HCHO" and in "$TiO_2$/$NO_3^-$ +HCHO" systems. The results were shown in Figure S6 of the original manuscript (Figure 9 here). For better understanding, we revised the last sentence "Future studies should explore whether HCHO affects the photocatalytic renoxification of $NO_3^-$-$TiO_2$." in the second paragraph of section 3.2 as "In the following study, the effect of HCHO on the photocatalytic renoxification of $NO_3^-$-$TiO_2$ was explored".

[Figure]

Figure 9. Photocatalytic degradation curve of HCHO on TiO$_2$ and 4 wt.% KNO$_3$-TiO$_2$ particles under 365 nm LED lamps at 293 K and 0.8% of relative humidity.

The decreased concentration of HCHO during this process can affect the formation of NO$_x$, which can be reflected by the flattening trend of NOx production after 60 min irradiation. This effect had been discussed in the original manuscript, line 247-249: *"The slow stage is due to the photodegradation of HCHO on KNO$_3$-TiO$_2$ aerosols, which led to a decrease in its concentration, gradually weakening the positive effect"*. In addition, the amount of NO$_x$ was also significantly reduced under the experiment of low concentration of HCHO (section 3.3.4 of the original manuscript), proving again the important role of HCHO in NO$_x$ release.

*6.The mixture of HNO$_3$ and TiO$_2$ was used to support that HNO$_3$ was an important intermediate to form NOx. However, this logic is not right. If it is right, then any N-contained components mixed with TiO$_2$ that enhanced the generation of NOx could be thought as the intermediates of NOx formation. The direct way to identify the intermediates is to measure them such as FTIR/DRIFTS to measure the adsorption products.*

**Response:**

Thanks for your comments, which will be valuable and helpful for revising and improving the manuscript. What we want to emphasize here is that a hydrogen abstraction reaction was occurred between HCHO and $NO_3$ radical to produce $HNO_3$, with the description shown in equation 5 and Line 252-254 in the original manuscript (*This burst-like generation of $NO_x$ can be ascribed to the reaction between generated $NO_3·$ and HCHO via hydrogen abstraction to form adsorbed nitric acid ($HNO_3(ads)$) on $TiO_2$ particles*). That meant the formed $HNO_3$ came from the original nitrate, and was responsible for the fast $NO_x$ release. The $HNO_3$-$TiO_2$ system is used as a comparison test to demonstrate the proposed mechanism and the photolysis contribution of $HNO_3$ to $NO_x$. FTIR/DRIFTS can be used for detecting species formation, but what we used in our study were nitrates, so no significant change in peak intensity would be observed.

*Minor comments:*

*1.Abstract: many sentences are confusing me! I can't understand what the main meaning of the work. What's the main results. The languages need to be improved.*

**Response:**

Thanks for your comments, which will be all valuable and very helpful for revising and improving the manuscript. In the revised manuscript, we rewrote the abstract for better understanding. The terms of "renoxification" and "photocatalytic renoxification" were stated firstly and then the reaction system was introduced. After that, the experimental results were shown, and the reaction pathway was suggested. The revised abstract is as follows: "Renoxification is the process of recycling of $NO_3^-$/$HNO_3$ into $NO_x$ under illumination, which is mostly ascribed to the photolysis of nitrate. $TiO_2$, a typical mineral dust component, can play its photocatalytic role in "renoxification" process due to $NO_3$ radical formed, and we define this process as "photocatalytic renoxification". Formaldehyde (HCHO), the most abundant carbonyl compound in the atmosphere, may participate in the renoxification of nitrate-doped $TiO_2$ particles. In this study, we established an environmental chamber reaction

system with the presence of HCHO and nitrate-doped $TiO_2$. The direct photolyses of both nitrate and $NO_3$ radical were excluded by adjusting the illumination wavelength, so as to explore the effect of HCHO on the "photocatalytic renoxification". It is found that $NO_x$ concentration can reach up to 110 ppb for 4 wt% $KNO_3$-$TiO_2$ particles, and was 2 orders of magnitude higher than in the absence of HCHO. Nitrate type, relative humidity and HCHO concentration were found to influence $NO_x$ release. Adsorbed HCHO may react with nitrate radicals through hydrogen abstraction to form adsorbed $HNO_3$ on the surface. The mass generation of $NO_x$ was suggested to via the $NO_3^-$-$NO_3 \cdot$-HCHO-$HNO_3$-$NO_x$ pathway, with HCHO and $TiO_2$ exhibiting a significant synergistic effect. Our proposed reaction mechanism by which HCHO promotes photocatalytic renoxification is helpful for deeply understanding the atmospheric photochemical processes and nitrogen cycling.

*2."photocatalysis", "photolysis", "photocatalytic", "photochemical" appeared in the manuscript everywhere, the author should make sure the exact meaning of these words and give the right usage of these words.*

**Response:**

Thanks for your comments. Table 3 illustrates the use of the four words. "Photocatalysis" and "photocatalytic" are used to refer to the chemical reactions that occur when photocatalysts such as $TiO_2$ are irradiated. The word "photolysis" refers to the breaking of chemical bonds of the substance itself under light, especially ultraviolet light, when there is no photocatalyst. In the manuscript, "photolysis" refers to the reaction of N-containing species themselves under irradiation. These three can be grouped together as "photochemical reactions". In the manuscript, we use "photochemical" in order to illustrate the broad meaning of such kind of reactions occurred in the atmosphere. For example, "photochemical processes" in the paper refers to atmospheric oxidation and nitrogen cycling. The different words being used to better distinguish the reactions that occur in different situations. We have corrected the inappropriate wording in the revised manuscript, shown also in Table 3 below.

Table 3. Distinction between "photocatalysis", "photolysis", "photocatalytic" and "photochemical".

| Word | Meaning | Usage in our study |
|------|---------|--------------------|
| Photocatalysis (noun) | The photocatalytic properties and photocatalytic activity of the compounds. | This word was used only once in the abstract section. In order to simplify the use of words in the paper, we have modified the abstract so that the word is no longer used. |
| Photolysis (noun) | The chemical bonds of the substance itself are broken under light, especially ultraviolet light. | "Photolysis" refers to the reaction of N-containing species themselves under irradiation |
| Photocatalytic (adjective) | Photocatalytic effect of photocatalyst. | The word is used wherever $TiO_2$ is mentioned. |
| Photochemical (adjective) | Series of chemical reactions that occurred under irradiation. | Some sentence such as "..photochemical cycle of HOx radicals.." are still use the word. There is one revision in the revised manuscript. "$NH_4^+$ and NO are photochemically oxidized on $TiO_2$" is Modified to "$NH_4^+$ and NO are photocatalytically oxidized on $TiO_2$". |

*3.Line 232-233, the photodegradation of HCHO on $TiO_2$ is not zero-order reaction kinetics, the curve is not a line as shown in Figure S6, which decreased slowly and then fast. The reason for it should be the large amount of adsorption of HCHO on the particle during the long-time injection of HCHO. Besides, the continuous wall loss of particle would result in the change of kinetic coefficient. The concentration of particles and HCHO were too high, and the injection time was too long to give clear kinetic parameters. Generally, the photocatalytic process is supposed to be a first order reaction.*

**Response:**

Thanks for your comments, which will be all valuable and very helpful for revising and improving the manuscript, as well as the important guiding significance to our researches.

(1)Regarding the question of reaction kinetics

We fitted the photocatalytic degradation curves of HCHO on $TiO_2$ and $KNO_3$-$TiO_2$ using the data of Figure S6 in the original manuscript for zero-order reaction and first-order kinetics, respectively. As shown in Figure 10 below, for both

TiO$_2$ and KNO$_3$-TiO$_2$ systems, the correlation coefficients ($R^2$) of the zero-order fitting is larger than that of the first-order fitting, so the photocatalytic degradation of HCHO on TiO$_2$ and KNO$_3$-TiO$_2$ fit zero-order kinetics. In order to show the curves more clearly, we marked the KNO$_3$-TiO$_2$ line in red color with the new figure shown as Figure S7 in the revised manuscript (Figure 11 below).

[Figure]

Figure 10. The comparison of correlation coefficients of zero- and first-order reaction curves. The reaction systems are "HCHO + TiO$_2$" (a) and "HCHO + 4 wt.% KNO$_3$-TiO$_2$" (b), both under 365 nm LED lamps at 293 K and 0.8% of relative humidity.

[Figure]

Figure 11. Photocatalytic degradation curve of HCHO on TiO$_2$ and 4 wt.% KNO$_3$-TiO$_2$ particles under 365 nm LED lamps at 293 K and 0.8% of relative humidity.

(2)Regarding the question of possible change of kinetic coefficient

As shown in Figures 3-5 above, the particle size distribution in the chamber can be maintained for several hours, with the number concentration in the chamber

decreased by no more than 10% per hour. In addition, the good correlation coefficient of 0.9779 and 0.9745 for $TiO_2$ and $KNO_3$-$TiO_2$, respectively, shown in Figure 11 also reflected that the kinetics fitting and rate constants are believable.

(3)Regarding the injection time and the adsorption of HCHO

We are sorry to make the reviewer misunderstand the operation sequence of the experiment. Figure S3 in the original manuscript is the conditional experiment results to show the HCHO adsorption in the dark before and after particles injection. We think it is Figure S3 that make the reviewer misunderstand. So we modified Figure S3, shown below as Figure 12. The operation sequence of the experiment is as follows. HCHO gases was flowed 10 min into the clean chamber firstly under dark conditions. As can be seen in Figure 12, HCHO can get equilibrium around 90 min. After that, no obvious decrease of HCHO was observed meaning that no further HCHO was adsorbed by the chamber. Then the particles were introduced into the chamber instantly. The concentration of HCHO began to decrease upon particles injection and need 60 min to reach its second adsorption equilibrium. After that, HCHO concentration can be maintained in the dark for several hours, indicating no further adsorption of HCHO by the chamber and the particles. In irradiation experiments, we waited for both HCHO and particles to reach stable before turning on the lights. The related description of the operation has been revised in the manuscript.

[Figure]

Figure 12. The conditional experiments of HCHO concentration in the environmental chamber in the dark before and after the introduction of particles over time.

*4.I can't understand why the authors used KNO₃ and HNO₃ to mixture with TiO₂. In Ma's work, they indicated the NOx concentration formed from KNO₃ was the lowest. KNO₃ only accounts for small proportion in the atmospheric particles. HNO₃ is acid species and can react with TiO₂, which would result in the component changes in this mixture particles. I think that the components in this mixture particles were different from the discussion in the article.*

**Response:**

Thanks for your comments, which will be all valuable and very helpful for revising and improving the manuscript, as well as the important guiding significance to our researches. We have thought deeply about the experimental design, and answer the comments point by point.

*(1)I can't understand why the authors used KNO₃ and HNO₃ to mixture with TiO₂. In Ma's work, they indicated the NOx concentration formed from KNO₃ was the lowest. KNO₃ only accounts for small proportion in the atmospheric particles.*

**Response:**

Although $KNO_3$ accounts for small proportion in the atmospheric particles, it is also important for atmospheric chemistry studies. Wang et al (Sci. Total Environ., 2019, 660: 47-56) found that in winter haze episodes, the formation of $KNO_3$ particles in the droplet-mode plays an important role in the increase of $PM_{2.5}$ concentration. So the $KNO_3$-related chemical reactions are important for the study of high pollution weather. In addition, in laboratory studies of nitrate photolysis, $KNO_3$ is still used as a model particle. For example, Yang et al (EP, 2018, 243: 679-686) used $KNO_3$ to study the effect of nitrate photolysis on HONO formation in the presence of humic acid; Xu et al (JES, 2021, 102: 198-206) used $KNO_3$ to study the effect of $TiO_2$ crystal structure on $NO_2$ and HONO emission from the nitrates photolysis.

In our study, $NH_4NO_3$ was also used for the study and the results were compared with those of $KNO_3$ to investigate the effect of cations on the photocatalytic renoxification process (as discussed in section 3.3.1). Similar to Ma's findings, lower $NO_x$ release was observed from $KNO_3$ composite compared to $NH_4NO_3$ composite.

We think this result may be caused by the blocking effect of $K^+$ on $NO_3^-$, which has been explained in the original manuscript text line 211-215.

*(2)HNO₃ is acid species and can react with TiO₂, which would result in the component changes in this mixture particles. I think that the components in this mixture particles were different from the discussion in the article.*

**Response:**

Some researches characterized the structure of $TiO_2$ after acid treatment and found no changes. For example, Wang et al. (J Mater Sci: Mater Electron, 2021, 32: 21083) treated $TiO_2$ with 98% concentrated sulfuric acid for 12 h, and demonstrated by XRD and XPS that acid treatment does not change the structure, elemental composition and chemical state of $TiO_2$ (Figure 13 below). In our experiment, a low content of $HNO_3$ (0.002 mol) was used to avoid the possible changes in composition of $TiO_2$. So it is estimated that the components of the particles would not change. We will make structure characterization to ensure this in our future study.

[Figure]

Figure 13. XRD images of TiO₂ nanotubes, g-C₃N₄, A1, A2 and A3. (A1: TiO₂ nanotubes after 12 h of H₂SO₄ treatment; A2: acid-treated TiO₂ compounded with g-C₃N₄; A3: TiO₂ without acid treatment compounded with g-C₃N₄) (J Mater Sci: Mater Electron, 2021, 32: 21083)

*5.OH radical was measured by ESR in this study. However, the role of OH radical has not been discussed. And the OH radical generated in different particles and under different conditions have not been compared and analyzed. Besides, NO₃ radical was proposed to be the important intermediates in the reaction. Why did not the authors measure NO₃ radical?*

**Response:**

Thanks for your comments, which will be all valuable and very helpful for revising and improving the manuscript, as well as the important guiding significance to our researches. The detection of OH radicals is for $TiO_2$ and Arizona dust, which is intended to demonstrate that photocatalysis process in these two particles do exist. In particular, the presence of OH radicals in the Arizona dust upon irradiation provides evidence that the findings of our study have practical implications. The emergence of $NO_x$ observed in the chamber demonstrated that HCHO promoted the renoxification of ATD particles (Figure S9 in the original manuscript). This result suggests that mineral dust containing photocatalytic semiconductor oxides such as $TiO_2$, $Fe_2O_3$, and ZnO could promote the conversion of granular nitrate to $NO_x$ in the presence of HCHO. The above discussions have been given in the original manuscript in lines 298-310.

$TiO_2$ produces OH radicals under UV illumination, which is well established in the field of photocatalysis (Xu et al., Appl. Catal., B, 2018, 230, 194-202). We provide this data to demonstrate that $TiO_2$ can be excited under our irradiation conditions and will exert its photocatalytic effect. In this case, other samples with $TiO_2$ as the main component would also generate OH radicals, although the amount may vary, but is not the main focus of our study. As what has been suggested, $NO_3$ radicals (coming from $h^+$ with $NO_3^-$) is the key species responsible for the formation of large amounts of $NO_x$. Unfortunately, $NO_3$ radical was not detected currently due to instrument limitations. Such measurement will be considered in our future studies. However, as discussed in the manuscript, the presence of $NO_3$ radicals was indirectly illustrated.

*6.Weight percentage was used to quantify nitrate in the mixed particle. However, different nitrate has different molecule weight, which would result in that the molar concentrations of different nitrates with the same weight percentage were different. For example, the molar concentration of N in 4 wt % HNO₃-TiO₂ is higher than that of N in 4 wt % KNO₃-TiO₂. This effect should be considered in the formation of NOx.*

**Response:**

    Thank you for your comments, which will be all valuable and very helpful for revising and improving the manuscript. We agree with the reviewers that different nitrates with the same weight percentage owns the difference in mole of N, which will cause the difference in $NO_2$ formation. To exclude this effect, we replotted Figure 2 of the original manuscript (here Figure 14 shown below) with mole normalized ppb as the $NO_2$ formation unit rather than ppb. As can be seen from Figure 14, the main conclusion is the same, with $HNO_3$–$TiO_2$ presented much higher activity than $KNO_3$–$TiO_2$ in the presence of HCHO. The discussion related to this Figure has been modified in the revised manuscript.

[Figure]

Figure 14. Effect of formaldehyde on the renoxification processes of different nitrate-doped particles at 293 K and 0.8% of relative humidity. 365 nm LED lamps were used during the illumination experiment. The initial concentration of HCHO was about 9 ppm.

**Special thanks to you for your good comments.**

---

## Author Comment (AC2)

**Response to the reviewer's comments:**

**Referee #2: General Evaluation**

*Liu et al. investigated the possible renoxification processes occurring on TiO$_2$ particles and mineral dust particles in presence of adsorbed nitrate, or HNO$_3$, in presence of HCHO. They suggest that HCHO and TiO$_2$ have a significant synergistic effect on the photocatalytic renoxification via a NO$_3$-NO$_3$-HCHO-HNO$_3$-NOx pathway, in which adsorbed HCHO may react with nitrate radicals through hydrogen abstraction to form HNO$_3$ on the surface, resulting an enhanced generation of NOx.*

*Overall this is an important topic, which certainly falls within the scope of journal Atmospheric Physics and Chemistry.*

**Response:**

Thanks for your comments, which will be all valuable and very helpful for revising and improving the manuscript, as well as the important guiding significance to our researches. We have thought deeply about the experimental design, and answer the comments point by point.

**Comments on Preprint acp-2022-6:**

*1. The experiments presented here were performed in a simulation chamber consisting of a 400 L polyvinyl fluoride (PVF) bag filled with synthetic air. In such a small baga, the life time of particles is expected to be very short, as shown also in figure S2. Surprisingly, the authors do observe, after some induction time, a stable size distribution over hours. How can this happen? Is it an indication of some dynamic interactions with the chamber's walls, during which particle adsorb and desorb constantly? Such process may be induced to some air turbulences around the bag, or through its deflation during the experiments (by the way, was the bag closed and its volume shrinking or was it flushed by pure air all the time during the experiments?). Anyhow, this is a strong indication that wall effects may play a significant role in the reported experiments. Therefore, a thorough discussion of these effects has to be included in the manuscript.*

**Response:**

Thanks for your comments. We are sorry that the description of operating procedures is not so clear. So we modified this in the revised manuscript. Here, we would like to describe the operation briefly. The sequence of the experimental operation of the chamber is as follows: cleaned by deionized water, dried totally, then inflated by synthetic air to a certain volume, then HCHO introduction, and then particles introduction instantly by a high pressure air flow. After the concentrations of both HCHO and particles became stable, the lamps were turned on and the concentrations of $NO_x$ were monitored. The chamber was closed during the entire experiment. Once the particles entered into the chamber, the number concentration began to decrease due to the wall effect. After 30 min, the size distribution of both $KNO_3$-$TiO_2$ and $TiO_2$ particles got stable and can sustained for several hours (with Figures 1-3 shown below). The left picture in the following Figure 2 shows the first 60 min of the size distribution in the dark, and the right picture of the size distribution during the irradiation. Figure 3 is another batch experiment of $TiO_2$ in the chamber with the same operation, and it shows very similar size distribution from 0-60 mins and 120-180 mins. During the experiment, we strictly controlled the same experimental condition before the start of each experiment and turned on the lamps only after the particles and HCHO concentration reached stability.

The corresponding revisions are as follows: (1) The injection of the particles has been emphasized in section 2.3 of the revised manuscript: "Particles were instantly sprayed into the chamber by a transient high-pressure airflow"; (2) The sentences "For the chamber operation, we completely evacuated the chamber after every experiment, then cleaned the chamber walls with deionized water and then dried by flushing the chamber with ultra-zero air to remove any particles or gases collected on the chamber walls" was added to the section 2.3 of the revised manuscript.

In addition, we checked our size distribution data of different samples, and found that the number concentration is not that high and usually around 4000 particle/cm$^{-3}$ when reaching stability. So Figure S2 in the original manuscript was deleted and replaced by the following Figure 1 (Figure S3 in the revised supplement).

[Figure]

Figure 1. Changes of particle size distribution of 4 wt.% KNO$_3$-TiO$_2$ particles in environmental chamber with time.

[Figure]

Figure 2. Changes of particle size distribution of TiO$_2$ particles in environmental chamber with time. Left: Before irradiation; Right: After irradiation.

[Figure]

Figure 3. Changes of particle size distribution of TiO$_2$ particles in environmental chamber with time (Another batch experiment).

*2. The main conclusion of this work is that adsorbed HCHO reacts with adsorbed nitrate radicals, promoting NOx formation. This assumes that this reaction is faster than the one of HCHO with the photochemically generated holes on the surface of the mineral. Is this justified by any means? HCHO being efficiently degraded on illuminated TiO₂, one would expect that this VOCs may compete with the nitrate anions to react with the holes, with the synergy between nitrate anions and HCHO vanishing at low surface coverage (where both compounds would react with the holes with no interactions with co-adsorbed species). Is this observed here?*

**Response:**

Thanks for your comments. As the reviewer mentioned, HCHO can react with the photochemically generated holes on the surface of the mineral. The photodegradations of HCHO on $TiO_2$ and $KNO_3$-$TiO_2$ particles were observed in this study. As shown in text line 230-239 of the original manuscript: "*Atmospheric trace gases can undergo photocatalytic reactions on the surface of $TiO_2$ (Chen et al., 2012). As the illumination time increased, the concentration of HCHO showed a linear downward trend, which was consistent with zero-order reaction kinetics (Figure S6). The zero-order reaction rate constants of HCHO on $TiO_2$ and 4 wt.% $KNO_3$-$TiO_2$ particles were 9.1 × 10⁻³ and 1.4 × 10⁻² ppm min⁻¹, respectively, which were much higher than that for gaseous HCHO photolysis (Shang et al., 2017). We suggested that the produced $NO_3$· contributed to the enhanced uptake of HCHO. Therefore, we suggest that $NO_3$· production contributed to enhanced HCHO uptake. Future studies should explore whether HCHO affects the photocatalytic renoxification of $NO_3^-$-$TiO_2$*". Besides the photodegradation of HCHO on excited $TiO_2$ particles, higher photodegradation rate of HCHO was observed on $KNO_3$-$TiO_2$ particles. As for if there is a simultaneously decrease of nitrate, we once compared the absorption spectra of the extracts of $KNO_3$-$TiO_2$ particles before and after reactions, with results shown below as Figure 4. It can be seen that nitrate content was decreased after reaction, meaning the vanishing of the nitrate.

By now, we do not know which of the reactions is faster ("HCHO+hole" or "HCHO+$NO_3$ radical"), but due to the high amount of both HCHO and $TiO_2$ used in

this study, on one side, there are enough holes to react with HCHO and nitrate at the same time, and on the other side enough remaining HCHO to react with $NO_3$ radical.

[Figure]

Figure 4. Absorption spectra of the extracts of $KNO_3$-$TiO_2$ particles before and after reactions.

*3. Spraying mixture of $SiO_2$ and $TiO_2$, would result in an externally mixed aerosol, isn't it? Then it should represent an experiment with the $TiO_2$ particles simply being diluted as compared to the pure $TiO_2$ experiment.*

**Response:**

Thanks for your comments. There are two reaction systems in our experiments. The first system, in which HCHO was not introduced, was intended to investigate the positive effect of $TiO_2$ on the renoxification process. In this system, we used 4 wt.% $KNO_3$-$SiO_2$ and 4 wt.% $KNO_3$-$TiO_2$ (1 wt.%)/$SiO_2$. Here $TiO_2$ (1 wt.%)/$SiO_2$ was prepared first and then 4 wt.% $KNO_3$ was composited. In the second system, HCHO was introduced to investigate the synergistic positive effect of $TiO_2$ and HCHO on the renoxification process. The samples used in this system were 4 wt.% nitrate-$TiO_2$. Here 4 wt.% nitrate was composited to the pure $TiO_2$ particles. Note that, in the first system, the main particle is $SiO_2$ and the content of $TiO_2$ is only 1 wt.% relative to $SiO_2$. While in the second system, the main particle is $TiO_2$. So there is a large difference in the $TiO_2$ mass in these two particles. It is not a simple dilution of $TiO_2$ with $SiO_2$.

*4. An effect of acidity is observed and explained by the enhanced photolysis of $HNO_3$. Could an alternative explanation arise for the chemistry of $O_2^-$? This superoxide would react with $H^+$ inducing $HO_2$ chemistry that may change a series of surface reactions. Could the authors comment on that?*

**Response:**

Thanks for your comments. Our experiments were performed under dry conditions, so there is little $H^+$ to react with $O_2^-$ to produce $HO_2^-$, and the mass release of $NO_2$ was ascribed to the photolysis of $HNO_3$. However, in our study of relative humidity as an influencing factor, there are some possible effects arising from $O_2^-$. Under high humidity, adsorbed $H_2O$ can behave as scavenger of photogenerated holes ($h^+$), so as to make photogenerated electron ($e^-$) reacted with oxygen, resulting in the generation of $O_2^-\cdot$ (Eq. 1). $O_2^-\cdot$ can combine with $H^+$ to generate $HO_2\cdot$ (Eq. 2). Subsequently, NO undergoes a two-step photocatalytic degradation on $TiO_2$: oxidation of NO by $HO_2\cdot$ to $NO_2$ (Eq. 3) and oxidation of $NO_2$ by $OH\cdot$ to $HNO_3$ (Eq. 4) (Dalton et al., EP, 2002, 120: 415-422; Devahasdin et al., J Photoch. Photobio. A, 2003, 156: 161-170). Therefore, higher relative humidity can affect NO and $NO_2$ production due to $HO_2$ chemistry, which has been discussed in section 3.3.3 of the original manuscript.

$$O_2 + e^- \rightarrow O_2^- \cdot \qquad (1)$$
$$O_2^- \cdot + H^+ \rightarrow HO_2 \cdot \qquad (2)$$
$$NO + HO_2 \cdot \rightarrow NO_2 + OH \cdot \qquad (3)$$
$$NO_2 + OH \cdot \rightarrow HNO_3 \qquad (4)$$

**Special thanks to you for your good comments.**

---

## Author Comment (AC3)

**Response to the reviewer's comments:**

**Referee #3: General Evaluation**

*The author reported formaldehyde may have synergistic effect in photocatalytic renoxification of nitrate with $TiO_2$, in order to explain the difference between field data and modeling result. The article focuses on the significant synergistic effect, i.e., HCHO and $TiO_2$ have on photocatalytic reactions and providing one possible reaction pathway- $NO_3$-$NO_3$-HCHO-$HNO_3$-NOx. These findings improve the understanding of the role of reactions between organic components and nitrate in the chemical and physical properties of aerosol particles in low relative humidity region. It has significant implication in the research of atmosphere and air pollution, but some issues in the article must be improved. I recommend accept this article after resolve those issues. There are the comments I have for this work:*

**Response:**

Thanks for your comments, which will be all valuable and very helpful for revising and improving the manuscript, as well as the important guiding significance to our researches. We have thought deeply about the experimental design, and answer the comments point by point.

**Comments on Preprint acp-2022-6:**

*1. There are some misdescriptions in the manuscript. Like:*

*We suggested that the produced $NO_3\cdot$ contributed to the enhanced uptake of HCHO. Therefore, we suggest that $NO_3\cdot$ production contributed to enhanced HCHO uptake. (Line 236, Page 12)*

*photochemical cycle of HOx radicals in the atmosphere and the formation of (Line 448, Page 25)*

**Response:**

Sorry for our carelessness about these sentences. In the revised manuscript, we have deleted the second half of the first sentence mentioned by the reviewer. The revised sentence is: "*We suggested that the produced $NO_3\cdot$ contributed to the enhanced uptake of HCHO*". For the second sentence, HOx means some reactive species such as $HO_2$ etc. Due to there is not many discussion about $HO_2$ in the

manuscript, we revised the sentence as: *"photochemical cycle of reactive radicals in the atmosphere and the formation of…"*

*2. Please explain why 4 wt.% KNO₃-TiO₂(1 wt.%)/SiO₂ underwent reaction to release NOx, while 4 wt.% KNO₃-SiO₂ and 4 wt.% KNO₃-TiO₂ not in same condition.*

**Response:**

Thanks for your comments. In our experiment, two reaction systems existed. The reaction systems and aims of the two kinds of light sources were different, which were summarized in the following Table 1. The first system, in which HCHO was not introduced, was intended to investigate the positive effect of $TiO_2$ on the renoxification process. In this system, we used 4 wt.% $KNO_3$-$SiO_2$ and 4 wt.% $KNO_3$-$TiO_2$ (1 wt.%)/$SiO_2$. The light source for this system is tube lamps. In the second system, HCHO was introduced to investigate the synergistic positive effect of $TiO_2$ and HCHO on the renoxification process. Our samples used in this system study were 4 wt.% nitrate-$TiO_2$. The light source for this system is LED lamps. LED lamps do not contain visible light component, so the effect of $NO_x$ release from the photolysis of $NO_3$ radicals under visible light could be excluded. So 4 wt.% $KNO_3$-$TiO_2$(1 wt.%)/$SiO_2$ (or 4wt.% $KNO_3$-$SiO_2$) and 4wt.% $KNO_3$-$TiO_2$ are not used in the same reaction system. The 4 wt.%$KNO_3$-$SiO_2$ did not release $NO_x$ because $SiO_2$ has no photocatalytic activity, as we discussed in section 3.1 of the manuscript.

**Table 1.** A comparison of the different particles used.

| Light source and Wavelength range | Section | System | Phenomenon | Contents/Implications |
|---|---|---|---|---|
| TUBE LAMP, 320-400 nm with small amount of 480-600 nm | 3.1 | 4 wt.% KNO₃-SiO₂ (NO HCHO) | No NOx was released | ⬜320-400 nm irradiation cannot make nitrate photolyze, so to exclude the photolysis source of NOx from nitrate. [Fig 1] |
| | | 4 wt.% KNO₃-TiO₂ (1 wt.%)/SiO₂ (NO HCHO) | NOx release was observed | ⬜TiO₂ was composited to SiO₂, to see the effect of TiO₂ along with nitrate. ⬜320-400 nm irradiation can excite TiO₂ to generate electron-hole pairs. ⬜Nitrate can react with photogenerated holes to produce NO₃ radicals ⬜480-600 nm irradiation can make NO₃ radical photolyze, generating NOx ⬜Above explanation can be seen in section of 3.1 and the equations 1-4 in the original manuscript. |

| | | | | |
|---|---|---|---|---|
| ◻Section 3.1 proves that $NO_3$ radical can be produced in the condition of "nitrate+ $TiO_2$ (excited by the UV light of the tube lamp)", then $NO_3$ radical undergo photolysis (under visible light of the tube lamp) to produce NOx.
◻In order to avoid the photolysis of $NO_3$ under visible light, no tube light containing visible light was used any more in section 3.2.
◻The aim of 3.2 is to investigate whether the produced $NO_3$ radical can be captured by HCHO.
This explanation can be seen in the first paragraph in section 3.2, that is Line 219-229 of the original manuscript. | | | | |
| LED LAMP,
350-390 nm | 3.2 | (WITH HCHO)
4 wt.% nitrate-$TiO_2$ | NOx release was observed | ◻350-390 nm irradiation cannot make nitrate photolyze, so to exclude the photolysis source of NOx from nitrate. [Fig S5]
◻350-390 nm irradiation can excite $TiO_2$ to generate electron-hole pairs, so as to generate $NO_3$ radical.
◻350-390 nm irradiation cannot make $NO_3$ radical photolyze.
◻$NO_3$ radical can react with HCHO to produce $HNO_3$(ads)
◻The observed NOx comes from the photolysis of $HNO_3$(ads). |
| ◻This explanation can be seen in section 3.2, including Fig 2 and equations 5-8. | | | | |

*3. In Figure S3, the concentration of HCHO and $TiO_2$ particles reached stable in 60 min after introduced into experimental chamber, but other experiments almost stared in -30min, did HCHO and $TiO_2$ have been stable?*

**Response:**

Thanks for your comments. We are sorry to make the reviewer misunderstand the operation sequence of the experiment. Figure S3 in the original manuscript is the conditional experiment results to show the HCHO adsorption in the dark before and after particles injection. We have modified Figure S3, shown below as Figure 1. The operation sequence of the experiment is as follows. HCHO gases was flowed 10 min into the clean chamber firstly under dark conditions. As can be seen in Figure 1, HCHO can get equilibrium around 90 min. After that, no obvious decrease of HCHO was observed meaning that no further HCHO was adsorbed by the chamber. Then the particles were introduced into the chamber instantly. The concentration of HCHO began to decrease upon particles injection and need 60 min to reach its second adsorption equilibrium. After that, HCHO concentration can be maintained in the dark

for several hours, indicating no further adsorption of HCHO by the chamber and the particles. In irradiation experiments, we waited for both HCHO and particles to reach stable before turning on the lights. The related description of the operation has been revised in the manuscript. The -30 min in others figures refers to the concentration measurements of $NO_2$ in the dark, and the adsorption of HCHO or particles was begun long before that.

[Figure]

Figure 1. The conditional experiments of HCHO concentration in the environmental chamber in the dark before and after the introduction of particles over time.

*4. The BET of TiO₂ nanoparticles is huge, the uptake of HCHO in TiO₂ nano-particles can't be ignored. The photodegradation of HCHO on TiO₂ and 4 wt.% KNO₃-TiO₂ particles should start after adsorption and desorption balance.*

**Response:**

Thanks for your comments. In our study, the uptake of HCHO on $TiO_2$ nano-particles was considered. As shown in Figure 1 above of conditional experiments of HCHO adsorption in the dark, it needs 60 min (from 180-240 min in the Figure) for HCHO to get stable after the particles' injection. For each experiments, we waited at least 60 min to ensure that the adsorption equilibrium has been reached.

*5. In section 3.3.1, 4 wt.% KNO₃-TiO₂ particles release less NOx than Equal amounts of 4 wt.% NH₄NO₃-TiO₂ particles at 293K and 0.8% relative humidity, which may be the result of the Relative molecular mass difference between KNO₃ and NH₄NO₃.*

**Response:**

Thanks for your comments. We agree with the reviewers that different nitrates with the same weight percentage owns the difference in mole of N, which will cause the difference in $NO_2$ formation. To exclude this effect, we replotted Figure 3 of the original manuscript (here Figure 2 shown below) with mole normalized ppb as the $NO_2$ formation unit rather than ppb. As can be seen from Figure 2, the main conclusion is the same, with $NH_4NO_3$-$TiO_2$ presented higher activity than $KNO_3$-$TiO_2$ in the presence of HCHO. The discussion related to this Figure has been modified in the revised manuscript.

[Figure]

Figure 2. Effect of formaldehyde on the renoxification processes of 4 wt.% $NH_4NO_3$-$TiO_2$ and 4 wt.% $KNO_3$-$TiO_2$ particles at 293 K and 0.8% of relative humidity. 365 nm LED lamps were used during the irradiation experiment. The initial concentration of HCHO was about 9 ppm.

*6. In section 3.3.4. more different initial concentration of HCHO should be test to find out from which content the positive effect become weakening.*

**Response:**

Thanks for your comments. There is an equilibrium of $NO_2$ release in our reaction system, one is the photocatalytic oxidation reaction between $NO_x$ and ROS (generated from excited $TiO_2$), and the other is the renoxification of $NO_3^--TiO_2$ particles. As discussed in the section 3.3.4 in original manuscript, these two competitive reactions will determine the up or down of $NO_2$.

We measured the variations of HCHO concentration both in its high and low system, with results shown in Figures S6 and Figure 3 below. The rate constant of HCHO decay are $1.4 \times 10^{-2}$ ppm min$^{-1}$ and $1.3 \times 10^{-3}$ ppm min$^{-1}$, respectively. The $NO_2$ generation rate in Figure S2 and Figure S10 are 1.2 ppm min$^{-1}$ and 0.1 ppm min$^{-1}$. So the HCHO concentration, HCHO decay rate and $NO_2$ generation rate all decreased a factor of 10. This coincidence gives us a clue that there may be some connections among these parameters. Another clue is that from Figure S10, it can be seen that the concentration of $NO_2$ begin to decrease at the time of 50 min, and corresponds to about 0.95 ppm of HCHO (as shown in Figure 3). So it is indicated that below 0.95 ppm of HCHO, the reaction between $NO_x$ and ROS is dominant. More experimental evidences regarding the point of HCHO concentration making positive effect weaken need further investigation. Atmospheric HCHO concentrations are generally very low. However, in cities with high traffic density, because combustion produces emissions, HCHO concentrations will be much higher than normal. In the indoor environment, HCHO levels can increase due to smoking, emissions from gas stoves and furniture, and can reach up to around 0.4 ppm (Formaldehyde. In: Wood dust and formaldehyde. Lyon, International Agency for Research on Cancer, 1995, 217-362). So, we assume that the positive effect of HCHO on the renoxification may still exist at some specific situation with its high concentration.

[Figure]

Figure 3. Photodegradation curves of low concentration formaldehyde on 4 wt.% KNO$_3$-TiO$_2$ particles under 365 nm LED illumination.

It is worthy of noting that although there usually not so much high concentration of HCHO, there are many other organics in the atmosphere which can provide hydrogen atoms to behave similar role as HCHO. As discussed in line 459-465 of the manuscript: "*The effect of low-concentration HCHO on the renoxification of NO$_3^-$-TiO$_2$ particles requires further investigation. However, many types of organics provide hydrogen atoms in the atmosphere, including alkanes (e.g., methane and n-hexane), aldehydes (e.g., acetaldehyde), alcohols (e.g., methanol and ethanol), and aromatic compounds (e.g., phenol) that react with NO$_3$· to produce nitric acid (Atkinson, 1991). These organics, together with HCHO, play similar positive roles in photocatalytic renoxification and, therefore, influence NO$_x$ concentrations*". We also believe that the effect of more different concentrations of HCHO on the renoxification of NO$_3^-$-TiO$_2$ particles deserves to be studied.

**Special thanks to you for your good comments.**

---

## Author Response (AR3)

August 4, 2022

Editor-in-Chief, Atmospheric Chemistry and Physics

Dear Prof. Ryan Sullivan:

Thank you very much for your kind consideration on our paper "The Positive Effect of Formaldehyde on the Photocatalytic Renoxification of Nitrate on $TiO_2$ Particles" (Manuscript ID: Preprint acp-2022-6). The comments are very valuable and helpful for revising and improving the manuscript. In the new revised manuscript, we have considered the comments and made revisions accordingly.

We have added the relevant experimental parameters used and stated the limitation and implication of our manuscript in Abstract. Moreover, the pathway has been modified from "$NO_3^-$-$NO_3\cdot$-HCHO-$HNO_3$-$NO_x$" to "$NO_3^-$-$NO_3\cdot$-$HNO_3$-$NO_x$" together with some explanation for better understanding, and has also been modified in the full text. In the section of Introduction, a specific scenario of low $NO_3^-/(NO_3^-+TiO_2)$ ratio was presented.

Please see the response and the revised manuscript files for details. Thanks a lot for your time and help.

Yours sincerely,

Jing Shang

College of Environmental Sciences and Engineering, Peking University

Beijing, 100871, P. R. China

E-mail: shangjing@pku.edu.cn

Comments

The central issue of concern here is how relevant are these experimental particle systems to atmospheric conditions. It is hard to imagine a situation in the atmosphere where you would have mostly just very small $TiO_2$ particles with lots of $HNO_3$/nitrate adsorbed, and 100s of ppb of HCHO available to also absorb and participate in the photochemistry.

The abstract should specify the relevant experimental parameters used, especially the high HCHO concentrations used. And also state that these experimental conditions are only proxies or simplified mimics for atmospheric mineral dust that do not necessarily reflect the chemistry that atmospheric mineral dust particles will facilitate. Furthermore, the statement in the abstract "The mass generation of $NO_x$ was suggested to via the $NO_3^-$-$NO_3\cdot$-HCHO-$HNO_3$-$NO_x$ pathway" is really hard to understand. Please rephrase this to express this statement more clearly.

Please also correct the figure numbering as was noted by the editorial staff: "For the next revision, please renumber the figures in the manuscript. Now you have two figures #5, missing the figure #4."

**Response:**

(1) We agree with the editor that it is not a normal atmospheric situation of the mimic reaction system. In fact, this phenomenon may exist in dusty weather. Some background had been mentioned in the section of Introduction of the manuscript: "*Atmospheric $TiO_2$ is mainly derived from windblown mineral dust, with mass mixing ratios ranging from 0.1 to 10% (Chen et al., Chem. Rev., 2012, 112, 5919-5948). $TiO_2$ is widely used in industrial processes and building exteriors for its favorable physical and chemical properties, and therefore the atmospheric $TiO_2$ content accumulates gradually. Titanium and nitrate ions have been found to coexist in atmospheric particulates in different regions worldwide (Sun et al., J. Geophys. Res. Atmos., 2005, 110, 1-11; Liu et al., AE, 2005, 39, 4453-4470; Yang et al., ACP, 2011, 11, 5207-5219; Kim et al., B. Korean Chem. Soc., 2012, 33, 3651-3656).*" Moreover, Sun et al. (J. Geophys. Res. Atmos., 2005, 110, 1-11) observed that the $NO_3^-$/($NO_3^-$+$TiO_2$) mass

percentage of total suspended particulate matter (TSP) during dust storms can be lower than 20%. We have added this result in the revised manuscript, see in Line 77-79. In this case, the $NO_3^-$-$TiO_2$ composite particles with $TiO_2$ as the main body can in some extent represent the real situation under sandstorm weather and have some practical significance. Moreover, in cities with high traffic density, HCHO concentrations will be much higher than normal because of combustion emissions. In the indoor environment, HCHO levels can increase due to smoking, emissions from gas stoves and furniture, and can reach up to around 0.4 ppm (Formaldehyde. In: Wood dust and formaldehyde. Lyon, International Agency for Research on Cancer, 1995, 217-362; T. Salthammer, Build. Environ., 2019, 150, 219-232). This description of the presence of high HCHO concentration has been added in the Line 96-98 of revised manuscript.

(2) With regard to the Abstract, we rewrote it for better understanding by listing the experimental parameters, stating the limitation and clarifying the reaction pathway in the revised manuscript. The revised abstract is as follows with added or modified text marked in red: "Renoxification is the process of recycling of $NO_3^-$/$HNO_3$ into $NO_x$ under illumination, which is mostly ascribed to the photolysis of nitrate. $TiO_2$, a typical mineral dust component, can play its photocatalytic role in "renoxification" process due to $NO_3$ radical formed, and we define this process as "photocatalytic renoxification". Formaldehyde (HCHO), the most abundant carbonyl compound in the atmosphere, may participate in the renoxification of nitrate-doped $TiO_2$ particles. In this study, we established a 400 L environmental chamber reaction system capable of controlling 0.8-70% relative humidity at 293K, with the presence of 1 or 9 ppm HCHO and 4 wt.% nitrate-doped $TiO_2$. The direct photolyses of both nitrate and $NO_3$ radical were excluded by adjusting the illumination wavelength, so as to explore the effect of HCHO on the "photocatalytic renoxification". It was found that $NO_x$ concentration can reach up to more than 100 ppb for nitrate-doped $TiO_2$ particles, while almost no $NO_x$ was generated in the absence of HCHO. Nitrate type, relative humidity and HCHO concentration were found to influence $NO_x$ release. It was suggested that substantial amounts of $NO_x$ were produced via the

$NO_3^-$-$NO_3\cdot$-$HNO_3$-$NO_x$ pathway, where $TiO_2$ worked for converting "$NO_3^-$" to "$NO_3\cdot$", HCHO participated in transformation of "$NO_3\cdot$" to "$HNO_3$" through hydrogen abstraction, and "$HNO_3$" photolysis answered for mass NOx release. So, HCHO played a significant role in this "photocatalytic renoxification" process. These results were found based on simplified mimics for atmospheric mineral dust under specific experimental conditions, which might deviate from the real situation, but illustrated a possible way of HCHO in influencing nitrate renoxification in the atmosphere. Our proposed reaction mechanism by which HCHO promotes photocatalytic renoxification is helpful for deeply understanding the atmospheric photochemical processes and nitrogen cycling, and could be considered for better fitting of atmospheric model simulations with field observations in some specific scenarios."

(3) In the revised manuscript, we have renumbered the fourth figure in the manuscript as "Figure 4".

---

## Author Response (AR4)

**List of changes made in the revised manuscript**

1. In the full text, the pathway has been modified from "$NO_3^-$-$NO_3\cdot$-HCHO-$HNO_3$-$NO_x$" to "$NO_3^-$-$NO_3\cdot$-$HNO_3$-$NO_x$" for better understanding.

2. The abstract was rewritten. The terms of "renoxification" and "photocatalytic renoxification" were stated firstly and then the reaction system was introduced. After that, the experimental results were shown, and the reaction pathway "$NO_3^-$-$NO_3\cdot$-$HNO_3$-$NO_x$" was suggested. Moreover, we have added the relevant experimental parameters used and stated the limitation and implication.

3. In the section of Introduction, a specific scenario of low $NO_3^-/(NO_3^-+TiO_2)$ ratio was presented.

4. To prove that the samples were uniformly mixed and confirm the composition of the mixture particles, in section 2.2 of the revised manuscript, we added the results of diffuse reflectance fourier transform infrared spectroscopy (DRIFTS) characterization of $TiO_2$ particles compounded with different mass fractions of $KNO_3$. This DRIFTS figure has been added as Figure S2 in revised Supplement.

5. In section 2.3 of the revised manuscript, we described the environmental chamber experiment in more detail. (1) The cleaning of chamber was added: "*For the chamber operation, we completely evacuated the chamber after every experiment, then cleaned the chamber walls with deionized water and then dried by flushing the chamber with ultra-zero air to remove any particles or gases collected on the chamber walls*". (2) The injection of the particles was emphasized: "*Particles were instantly sprayed into the chamber by a transient high-pressure airflow*". (3) Figure S3 in the original manuscript (Figure S4 in the revised manuscript) was modified to emphasize that this is a conditional experiment results to show the HCHO adsorption in the dark before and after particles injection. Based on this conditional experiment, the time points were decided for particles' introduction and for lights turning on. (4) The picture of particle size distribution of $TiO_2$ particles in environmental chamber (Figure S2 in the original Supplement) was

replaced by picture of particle size distribution of 4 wt.% $KNO_3$-$TiO_2$ and $TiO_2$ particles in environmental chamber, and plotted them together with suitable symbols size (Figure S3 in the revised Supplement).

6. Since different nitrates with the same weight percentage owns the difference in mole of N, which will cause the difference in $NO_x$ formation. To exclude this effect, we replotted Figures 2, 3 and 5 of the original manuscript with millimole normalized ppb as the $NO_x$ formation unit rather than ppb (shown as Figures 2, 3 and 4, respectively). The main conclusions remain unchanged.

7. In section 3.2 of the revised manuscript, we have replenished some additional experimental results. (1) The effect of HCHO on the renoxification processes of 4 wt.% $KNO_3$-$SiO_2$ particles under 365 nm LED lamps irradiation were added, showed as Figure S9 in the revised Supplement. (2) The pH results of water extracts in $KNO_3$-$TiO_2$ systems with and without HCHO were added, suggesting the formation of acidic species such as $HNO_3$(ads) in this study.

8. Given that different kinds of nitrate owning different surface area will affect the reaction process, for sake of the reliability of the results, 3.3.2 section ("the influence of nitrate content") and the involved Figure 4 of original manuscript were deleted. Meanwhile, Figure S12 with data once embedded in Figure 4 was added in Supplement, with demonstrating that $KNO_3$-$TiO_2$ and $NH_4NO_3$-$TiO_2$ could not generate $NO_x$ in the absence of HCHO under LED irradiation.

9. To more clearly indicate that the photocatalytic degradation of HCHO on $TiO_2$ and $KNO_3$-$TiO_2$ fit zero-order kinetics, we modified the original Figure S6 by marking the $KNO_3$-$TiO_2$ line in red color with the new figure shown as Figure S7 in the revised manuscript.

**Response to the reviewer 1's comments:**

**Referee #1: General Evaluation**

*This manuscript investigated the effect of HCHO on the photocatalytic renoxification of nitrate on $TiO_2$ particles. The investigated system is interesting. However, the experimental design has so many defects. More experiments and verification are needed to support the conclusion.*

**Response:**

Thanks for your comments, which will be all valuable and very helpful for revising and improving the manuscript, as well as the important guiding significance to our researches. We have thought deeply about the experimental design, and answer the comments point by point.

**Comments on Preprint acp-2022-6:**

*Major comments:*

*1.Methods: Line 103: 400 L chamber is usually not enough for the investigation of heterogeneous reactions. Besides, only 250 L air was injected into it, which would increase the wall effect of chamber. Line 105-106: How did the author control the chamber temperature? It is well known that the chamber temperature will increase when turn on the lights. Line 111-112: the light intensities for the tube and LED lamp were different, so how to compare their results? Why was only the results in 3.1 obtained under the irradiation of tube lamps? What was the meaning for introducing two kinds of lamps in the smog chamber?*

**Response:**

Thanks for your comments, which will be all valuable and very helpful for revising and improving the manuscript, as well as the important guiding significance to our researches. We have thought deeply about the experimental design, and answer the comments point by point.

*(1)Line 103: 400 L chamber is usually not enough for the investigation of heterogeneous reactions. Besides, only 250 L air was injected into it, which would increase the wall effect of chamber.*

**Response:**

The environmental chamber can be used to assess and predict the formation of secondary pollutants. Large volume chamber is helpful for obtaining chemical transformation laws closer to the real atmosphere, although small volume chambers are still used in laboratory dynamics simulation studies due to their economic and flexible features. For example, Shi et al (EST, 2021, 55: 854-861) studied renoxification of suspended submicron particulate sodium and ammonium nitrate through controlled laboratory photolysis experiments using an 150 L Teflon environmental chamber. The reaction system is very similar as ours, focusing on $NO_x$ release from particulate inorganic nitrate with well-characterized light conditions. Another example is Jia et al (Aerosol Sci. Tech., 2014, 48: 1-12), who studied the formation of ozone and secondary organic aerosol (SOA) from benzene–$NO_x$ and ethylbenzene–$NO_x$ irradiations in a 350 L Teflon chamber. The major substances in SOA were determined to be carboxylic acids and glyoxal hydrates. And it was found that the aqueous radical reactions and the hydration from glyoxal can be enhanced under high RH conditions, which can irreversibly enhance the formation of SOA from both benzene and ethylbenzene. In another research, a 151 L chamber was used to study the kinetic of ozone decomposition on luminescent oxide surfaces (Chen et al., J. Phys. Chem. A, 2011, 115: 11979-1198). It is also a simulation research about heterogeneous photochemistry of ozone over mineral dust aerosol, including $\alpha\text{-}Fe_2O_3$, $TiO_2$, and $\alpha\text{-}Al_2O_3$. The rate and extent of ozone decomposition on these oxide surfaces are found to be a function of the nature of the surface as well as the presence of light and relative humidity, with $TiO_2$ is active toward $O_3$ decomposition upon irradiation. Therefore, it is true that 400 L chamber is not so big but can in some extent reflect the simulation results of heterogeneous reaction.

We tried our best to decrease the effects of "wall effect" on the experiment results by two ways. One was to conduct the experiments when the particle size distribution got stable, that is, 60 min after the injection of the particles. As can be seen in Figure S2 of the original manuscript, the size distribution gets stable after 30 min and can sustain for several hours. As stated in text line 166-168 of the original

manuscript: "*After the concentrations of both HCHO and aerosol became stable, the lamps were turned on and the concentrations of NOx were monitored.*" Another way was that we strictly carried out blank experiments and all data in the study were subtracted from the corresponding blank data to ensure the reliability. For example, to determine the background value of $NO_x$ in the reaction system, four blank experiments were carried out under illumination without nitrate: "synthetic air", "synthetic air + $TiO_2$", "synthetic air + HCHO" and "synthetic air + HCHO + $TiO_2$". In the blank experiments of "synthetic air" and "synthetic air + $TiO_2$", the $NO_x$ concentration remained stable during 180 min illumination, and the concentration change was no more than 0.5 ppb (Figure S4a of the original manuscript). Therefore, the environmental chamber was thought to be relatively clean, and there was no generation and accumulation of $NO_x$ under illumination. In addition, the chamber was cleaned after each experiment and the repeatability was ensured. To make it clear, we added one sentence to the section 2.3 Line 168-171 of the revised manuscript: "*For the chamber operation, we completely evacuated the chamber after every experiment, then cleaned the chamber walls with deionized water and then dried by flushing the chamber with ultra-zero air to remove any particles or gases collected on the chamber walls.*"

Overall, our experiments were conducted under relatively stable conditions, thus to exclude the effects of "wall effect" and the data were reliable.

*(2)Line 105-106: How did the author control the chamber temperature? It is well known that the chamber temperature will increase when turn on the lights.*

**Response:**

As stated in line 108-111 of the original manuscript: "*One is a set of tube lamps with a main spectrum of 320-400 nm and a small amount of 480-600 nm visible light (Figure S1a). The other is a set of Light-emitting diode (LED) lamps with a narrow main spectrum of 350-390 nm (Figure S1b)*", the light sources we used emit no infrared lights, and in case 36 W and 12 W of the tube and LED lamps were used, the temperature in the chamber is not easy to be heated. Figure S1b below (Figure 1 here) shows the wavelength distribution of the two kinds of lamps. We added watt values of

the lamps in the section 2.1 Line 121-124 of the revised manuscript and above sentence is now as follows: "*One is a set of 36 W tube lamps with a main spectrum of 320-400 nm and a small amount of 480-600 nm visible light (Figure S1a). The other is a set of 12 W Light-emitting diode (LED) lamps with a narrow main spectrum of 350-390 nm (Figure S1b)*". In addition, we controlled the temperature of the room where the chamber is, keeping it at 20 degrees all the time. So we consider the effect of temperature to be negligible in this study.

[Figure]

[Figure]

Figure 1. Spectral energy distribution of (a) 365 nm tube lamps and (b) 365 nm LED lamps.

*(3)Line 111-112: the light intensities for the tube and LED lamp were different, so how to compare their results? Why was only the results in 3.1 obtained under the irradiation of tube lamps? What was the meaning for introducing two kinds of lamps in the smog chamber?*

**Response:**

The reaction systems and aims of the two kinds of light sources were different, which were summarized in the following Table 1. The results of tube lamp and LED lamp would not be compared, but had their own purpose, individually.

For tube lamp (contains both ultraviolet and visible light) system in section 3.1, there is no HCHO but only nitrate with or without $TiO_2$ ($KNO_3$-$SiO_2$ or $KNO_3$-$TiO_2$/$SiO_2$). The aim is to prove that $NO_3$ radical can be produced via the reaction between excited $TiO_2$ and nitrate, with its photolysis under visible light producing $NO_x$.

For LED lamp (contains only ultraviolet) system in section 3.2, HCHO was introduced and the particles are $KNO_3$-$TiO_2$ or $HNO_3$-$TiO_2$. The aim is to avoid the

photolysis of $NO_3$ radicals under visible light, but to see if the produced $NO_3$ radicals can be reacted with HCHO. As stated in line 226-229 of the original manuscript: *"Therefore, the LED lamp setup was used in subsequent experiments to exclude the direct photolysis of both $KNO_3$ and $NO_3·$, but allow the excitation of $TiO_2$. This approach allowed us to investigate the process of photocatalytic renoxification caused by HCHO in the presence of photogenerated $NO_3·$."* In this way, the formation of $NO_x$ can be attributed to the photolysis of $HNO_3$, coming from the possible hydrogen-abstraction reaction of HCHO with $NO_3$ radicals, as discussed in the later part of section 3.2.

Therefore, the design of our experiments is: (1) Section 3.1 using tube lamp to prove that $NO_x$ is not coming from $KNO_3$ UV-light photolysis, but from the visible-light photolysis of $NO_3$ radical. By this way to prove the photocatalytic effect of $TiO_2$ on "renoxification" process, that is "photocatalytic renoxification"; (2) Section 3.2 using LED lamp to ensure no photolysis of both $KNO_3$ and $NO_3$ radical, and in this case to investigate the effect of HCHO on the release of $NO_x$ in the presence of $NO_3$ radical, i.e., the effect of HCHO on the "photocatalytic renoxification". In order to make the readers understand clearer, we rewrote the abstract according to this logic and defined the term "photocatalytic renoxification".

Table 1. A comparison of the two light sources used.

| Light source and Wavelength range | Section | System | Phenomenon | Contents/Implications |
|---|---|---|---|---|
| TUBE LAMP, 320-400 nm with small amount of 480-600 nm | 3.1 | 4 wt.% $KNO_3$-$SiO_2$ (NO HCHO) | No NOx was released | • 320-400 nm irradiation cannot make nitrate photolyze, so to exclude the photolysis source of NOx from nitrate. [Figure 1] |
| | | 4 wt.% $KNO_3$-$TiO_2$ (1 wt.%)/$SiO_2$ (NO HCHO) | NOx release was observed | • $TiO_2$ was composited to $SiO_2$, to see the effect of $TiO_2$ along with nitrate.
• 320-400 nm irradiation can excite $TiO_2$ to generate electron-hole pairs.
• Nitrate can react with photogenerated holes to produce $NO_3$ radicals.
• 480-600 nm irradiation can make $NO_3$ radical photolyze, generating NOx.
• Above explanation can be seen in section of 3.1 and the equations 1-4 in the original manuscript. |
| • Section 3.1 proves that $NO_3$ radical can be produced in the condition of "nitrate+ $TiO_2$ (excited by the UV light of the tube lamp)", then $NO_3$ radical undergo photolysis (under visible light of the tube lamp) to produce NOx. | | | | |

| | | | | |
|---|---|---|---|---|
| •In order to avoid the photolysis of NO₃ under visible light, no tube light containing visible light was used any more in section 3.2.
 •The aim of 3.2 is to investigate whether the produced NO₃ radical can be captured by HCHO.
 This explanation can be seen in the first paragraph in section 3.2, that is Line 219-229 of the original manuscript. | | | | |
| LED LAMP, 350-390 nm | 3.2 | (WITH HCHO) 4 wt.% nitrate-TiO₂ | NOx release was observed | •350-390 nm irradiation cannot make nitrate photolyze, so to exclude the photolysis source of NOx from nitrate. [Figure S5]
 •350-390 nm irradiation can excite TiO₂ to generate electron-hole pairs, so as to generate NO₃ radical.
 •350-390 nm irradiation cannot make NO₃ radical photolyze.
 •NO₃ radical can react with HCHO to produce HNO₃(ads).
 •The observed NOx comes from the photolysis of HNO₃(ads). |
| •This explanation can be seen in section 3.2, including Figure 2 and equations 5-8. | | | | |

*2. Important defect of this article is the composition of the mixture in the part of "2.2 nitrate-TiO₂ composite samples": "TiO₂ was simply mixed in nitrate solutions at the desired mass mixing ratio to obtain a mash. The mash was dried at 90℃ and then ground carefully to ensure a uniform composite of particles." How did the author ensure that the particles are uniform composite of nitrate and TiO₂? Did the author do some experiments to confirm these? For example, in the reference of Ma et al (EST, 8604-8612, 2021), the nitrate and TiO₂ mixture was dripped onto a quartz tube inner all, then images and Raman spectra of single composition and mixture were analyzed, and mixture were confirmed to form. However, in this work, the generation method of mixture particles is different from that of Ma's work, and these mixture particles are sprayed by synthetic air into PVF bag. No experiments have been given to confirm the composition of the mixture particles in the chamber. In my opinion, this method can't generate a uniform composite of nitrate and TiO₂!!! The composition and the nitrate content are the most important quantitative method factors of all the experiments in the article. If the composition and nitrate content can't be control, how*

*to compare the NOx concentration in different experiments? Then, all the results are not convincing!!!*

**Response:**

Thanks for your comments, which will be all valuable and very helpful for revising and improving the manuscript, as well as the important guiding significance to our researches. We prepared the composite particles carefully and ensured its homogeneity. During the preparation of nitrate and $TiO_2$ composite samples, we used a very small amount of nitrate solvent, which is of 2 mL. With a relatively large specific surface area (~54.28 $m^2/g$) and a large amount (250 mg) of $TiO_2$, the mixture was viscous and then quickly dried at 90°C, followed by a thorough grind for 30 min. So, nearly no loss of $TiO_2$ due to no use of filtration, and nearly no loss of nitrate pyrolysis due to low temperature of drying. Concerning the work of Ma et al (EST, 2021, 8604-8612) mentioned by the reviewer, the nitrate and oxides were mixed by dispersing a total mass of 1.0 g of oxide powder and 0.02 g of nitrate in 50.0 mL of ultrapure water. The measured nitrate loading percent (1.9-2.0 wt%) was very closed to theoretical value (2.0 wt%). This preparation method was very similar to ours in that both $TiO_2$ powders were dissolved in a nitrate solution. So the composites of nitrate and $TiO_2$ we obtained by our preparation method are thought to be uniform.

In addition, we used diffuse reflectance fourier transform infrared spectroscopy (DRIFTS) to characterized the structure of the particles, with the results approving the homogeneity. DRIFTS spectra of $KNO_3$-$TiO_2$ particles with different contents of nitrate as well as the $KNO_3$ particle were shown in the following Figure 2. It can be seen that DRIFTS spectra of the composited particles with $KNO_3$ contents higher than 1 wt% were very close to that of $KNO_3$. According to Aghazadeh's (J Ultrafine Grained Nanostruct Mater, 2016, 49(2): 80-86) and Maeda's (Applied Catalysis B: Environmental, 2011, 103(1-2), 154) work, 1760 $cm^{-1}$ are the vibrating peaks of nitrate. The ratios of the peak area from 1730-1790 $cm^{-1}$ for 1, 4, 32, 80 wt.% composited samples is 1: 4.1: 29.8: 81.6, which is very close to that of theoretical value, proving that the samples were uniformly mixed. This DRIFTS figure has been added as Figure S2 in Supplement of the revised manuscript, with its description

added in section 2.2 Line 147-155 of the revised manuscript: "*The mash was dried at 90 °C and then ground carefully for 30 min. A series of samples with different amount of nitrate were prepared and diffuse reflectance fourier transform infrared spectroscopy (DRIFTS) measurements were made to test their homogeneity. Figure S2 shows DRIFTS spectra of these $KNO_3$-$TiO_2$ composites, of which 1760 $cm^{-1}$ peak is one of the typical vibrating peaks of nitrate (Aghazadeh, 2016; Maeda et al., 2011). Ratio value of peak area from 1730-1790 cm-1 for 1, 4, 32, 80 wt.% composited samples is 1: 4.1: 29.8: 81.6, which is very close to that of theoretical value, proving that the samples were uniformly mixed.*"

[Figure]

Figure 2. DRIFTS spectra of $TiO_2$ particles compounded with different mass fractions of $KNO_3$.

*3.Another important defect of this article is the quantitative method of NOx concentration. As shown in Ma's work, they used the normalized concentration (ppb/mg) to quantify NOx. However, this work just used the NOx concentration (ppb) to compare different experiments, which meant that if more reactants were added in the chamber, the generated NOx concentration would be higher. The initial mass concentration of particles was 300 $mg/m^3$ (75mg/250L), and the concentration of HCHO was 10 ppm, which were much higher than that in the real environment and resulted in that the obtained results could not be directly used for an analogy to real environment. The results with ppb as unit are meaningless to reflect their influence in the real atmosphere. Were the particles kept the same in different experiments during the reaction? The author mentioned that the wall loss of particle in the smog chamber*

was very high at the beginning. And the wall loss for different kinds of particles and for the same kind of particles in different experiment (maybe affected by the conditions of the smog chamber wall) should be different. How did the author ensure that the particle distributions were the same in different experiments when turned on the light? Besides, the surface area, as an important factor in heterogeneous reactions, has not been detected in the experiments. Different surface areas directly affect the irradiation surface of $TiO_2$, the uptake of HCHO and the release of NOx. The missing information of surface area would result in the large uncertainties in the experiments. At least, the authors should give a normalized NOx concentration, then different experiments can compare with each other and give the reasonable results and reflect the influence in the real environment.

**Response:**

Thanks for your comments, which will be all valuable and very helpful for revising and improving the manuscript, as well as the important guiding significance to our researches. We have thought deeply about the experimental design, and answer the comments point by point.

(1)Another important defect of this article is the quantitative method of NOx concentration. As shown in Ma's work, they used the normalized concentration (ppb/mg) to quantify NOx. However, this work just used the NOx concentration (ppb) to compare different experiments, which meant that if more reactants were added in the chamber, the generated NOx concentration would be higher.

**Response:**

For flow tube experiments, the flow tube was usually weighted before and after its loading with samples and the normalized concentration (ppb/mg) was used for a better comparison between different samples, as what has been done as Ma's work. However, in our experiments, the amount of the different samples sprayed into the chamber is same, so the mass normalization is not necessary. That is, no matter whether it is expressed as ppb/mg or ppb, the same trend will be obtained, which will not affect the conclusions of our study.

*(2)The initial mass concentration of particles was 300 mg/m³ (75mg/250L), and the concentration of HCHO was 10 ppm, which were much higher than that in the real environment and resulted in that the obtained results could not be directly used for an analogy to real environment. The results with ppb as unit are meaningless to reflect their influence in the real atmosphere.*

**Response:**

The value of 75mg/250L is the amount we injected into the chamber, but according to the size distribution measurement, the real suspended particle concentration was not that high. As shown in Figure S2 of the original manuscript, the number concentration of $TiO_2$ particles is about 8500 particle/cm$^{-3}$ after reaching stability. This level of number concentration was observed by Wang et al (Environ. Chem., 2015, 34(9): 1619-1626) who measured the particle size distribution of atmospheric particulate matter number concentrations in Nanjing in August 2013, with the particle number concentration of about 8000 particle/cm$^{-3}$. As stated in line 156-158 of the original manuscript: "*The size distribution of $TiO_2$ reached stable after about 60 min with the peak particle size was about 120 nm, similar to that of atmospheric particles in some urban areas in China (Wang et al., 2015; Li et al., 2019).*" We checked our size distribution data of different samples, and found that the number concentration is not that high and usually around 4000 particle/cm$^{-3}$ when reaching stability, with the figures shown below (Figures 3-5 in this file). So, the Figure S2 in the original manuscript was deleted and replaced by the following Figure 3 (Figure S3 in the revised supplement).

[Figure]

Figure 3. Changes of particle size distribution of 4 wt.% $KNO_3$-$TiO_2$ particles in environmental chamber with time.

We admit that 10 ppm of HCHO is too high, so we also performed experiments with low concentrations of HCHO (1 ppm), as described in section 3.3.4 of the original manuscript (now 3.3.3 in the revised manuscript). The positive effect of HCHO on the photocatalytic renoxification of $KNO_3$-$TiO_2$ particles was still observed, with $NO_2$ concentration first increasing and then decreasing (Figure S10 of the original manuscript). Atmospheric HCHO concentrations are generally very low. However, in cities with high traffic density, because combustion produces emissions, HCHO concentrations will be much higher than normal. In the indoor environment, HCHO levels can increase due to smoking, emissions from gas stoves and furniture, and can reach up to around 0.4 ppm (Formaldehyde. In: Wood dust and formaldehyde. Lyon, International Agency for Research on Cancer, 1995, 217-362). So, we assume that the positive effect of HCHO on the renoxification may still exist at some specific situation with its high concentration, which requires further investigation. To make the presentation of our results more accurate, we have added this sentence in the section 4 Line 480-483 of the revised manuscript: "*Although in the case of high concentrations of HCHO in our experiment, the response to the real situation will be biased, the results of this study illustrate a possible way of HCHO in influencing nitrate renoxification in the atmosphere.*"

*(3)Were the particles kept the same in different experiments during the reaction? The author mentioned that the wall loss of particle in the smog chamber was very high at*

*the beginning. And the wall loss for different kinds of particles and for the same kind of particles in different experiment (maybe affected by the conditions of the smog chamber wall) should be different. How did the author ensure that the particle distributions were the same in different experiments when turned on the light?*

**Response:**

Thanks for the reviewer's thoughtful question. As mentioned before, we deflated and cleaned the chamber for each experiment, and the operation of each experiment was almost identical. So the particle number size distribution of the same kind of particles would be quite same, which can be proved by the following Figures 4 and 5.

The operation sequence of the experiment is as follows: HCHO was introduced firstly, and wait 60 min for its stability; then the particle was introduced instantly, and need 30 min for its stable, and another 30 min for HCHO's second stability; then the lights were opened. The experiment operation has been rewritten in the revised manuscript.

The left picture in the following Figure 4 shows the first 60 min of the size distribution in the dark, and the right picture of the size distribution during the irradiation time. Figure 5 is another batch experiment of $TiO_2$ in the chamber with the same operation, and it shows very similar size distribution from 0-60 mins and 120-180 mins.

[Figure]

Figure 4. Changes of particle size distribution of $TiO_2$ particles in environmental chamber with time. Left: Before irradiation; Right: After irradiation.

[Figure]

Figure 5. Changes of particle size distribution of TiO$_2$ particles in environmental chamber with time (Another batch experiment).

As for different kinds of particles, as comparison of Figure 3 with Figures 4 and 5, both main particle size (about 120 nm) and particle number concentration (about 4000 particle/cm$^{-3}$) are similar. Therefore, the particles can be kept the same in different experiments during the reaction, not only for the same particles but also for different kinds of particles. This is because of the same operation and the similar surface area of different kinds of particles (same loading of nitrate) as will be mentioned below.

*(4)Besides, the surface area, as an important factor in heterogeneous reactions, has not been detected in the experiments. Different surface areas directly affect the irradiation surface of TiO$_2$, the uptake of HCHO and the release of NOx. The missing information of surface area would result in the large uncertainties in the experiments. At least, the authors should give a normalized NOx concentration, then different experiments can compare with each other and give the reasonable results and reflect the influence in the real environment.*

**Response:**

We agree with the reviewer that surface area will affect the reaction process. We once measured BET of the KNO$_3$-TiO$_2$ particles with the results shown in Table 2. It can be seen that the BET values gradually decrease as the nitrate loading increases. As suggested by the reviewer, the surface area normalized reaction rate should be used to compare the particles. It is a regret that we did not measure the BET of NH$_4$NO$_3$-TiO$_2$ particles, so we cannot get the normalized parameter for NH$_4$NO$_3$-TiO$_2$ particles.

Therefore, for sake of the reliability of the results, we deleted the 3.3.2 section "the influence of nitrate content" of different kinds of nitrate. Except for section 3.3.2, the loadings were all 4 wt.% of nitrate. According to Table 2, the BET surface areas of the particles did not change much at 4 wt.% loading, which were mainly dependent on the specific surface area of the main body of $TiO_2$. So the estimated difference in BET surface area of $TiO_2$ loaded with different nitrates at 4 wt.% loading is not significant and has little effect on the reaction results. In addition, for 4 wt% of samples of different kinds of nitrate, we performed millimole normalized ppb for $NO_x$ concentrations, so that the effect of molecular weight of different species can be excluded.

Table 2. BET surface area of composite particles with different $KNO_3$ content.

| $KNO_3$ content in composite particles wt.% | BET $m^2/g$ |
| --- | --- |
| 0 | 54.28 |
| 1 | 50.7 |
| 4 | 48.04 |
| 12 | 41.77 |
| 20 | 36.86 |
| 32 | 26.67 |
| 50 | 18.45 |
| 80 | 5.61 |

*4.Gas HCHO and mixture particles of $TiO_2$ and nitrate were contained in the system. Although some controlled experiments were conducted, the role of $TiO_2$ and HCHO still could not be isolated. A series of important experiments such as HCHO and single nitrate particles under irradiations are needed.*

**Response:**

Thanks for your comments, which will be all valuable and very helpful for revising and improving the manuscript, as well as the important guiding significance to our researches. We agree with the reviewer that controlled experiments are needed not only for "$TiO_2$ + HCHO" system, but also for "HCHO+nitrate" system, which

have both been conducted in our study. Figure S6 in the original manuscript presented the HCHO decay with irradiated $TiO_2$, indicating the photocatalytic role of $TiO_2$. For "HCHO+nitrate" system, because the nitrates in our study were all loaded on particles, we composited nitrate with inert $SiO_2$. As shown in Figure 6 below, it can be seen that no $NO_2$ formation was observed whether HCHO was present or not, indicating that photocatalytically active particle $TiO_2$ is critical to the photocatalytic renoxification process. We have added this figure as Figure S9 in the revised supplement with its explanation in Line 326-328 in section 3.2.

[Figure]

Figure 6. Effect of formaldehyde on the renoxification processes of 4 wt.% $KNO_3$-$SiO_2$ particles at 293 K and 0.8% of relative humidity. 365 nm LED lamps were used during the irradiation experiment. The initial concentration of HCHO was about 9 ppm.

*5.All the proposed mechanisms couldn't be well supported only by the changes of NOx concentration. This work and Ma's work indicate HONO, HNO₃, NO₃ radical, NOx could form in these reaction systems. However, HONO, HNO₃, NO₃ radical could lead the overestimation of NO₂ concentrations by chemiluminescence method. How did the authors exclude the effect of these species? Besides, most important products such as NO₃, HNO₃, HONO were not detected in the experiments except OH radical. How did the authors make sure that the reaction pathway followed the*

*proposed mechanisms? It is well known that TiO₂ can photocatalysis HCHO, can this reaction affect the formation of NOx?*

**Response:**

Thanks for your comments, which will be all valuable and very helpful for revising and improving the manuscript, as well as the important guiding significance to our researches. We have thought deeply about the experimental design, and answer the comments point by point.

*(1)All the proposed mechanisms couldn't be well supported only by the changes of NOx concentration. This work and Ma's work indicate HONO, HNO₃, NO₃ radical, NOx could form in these reaction systems. However, HONO, HNO₃, NO₃ radical could lead the overestimation of NO₂ concentrations by chemiluminescence method. How did the authors exclude the effect of these species? Besides, most important products such as NO₃, HNO₃, HONO were not detected in the experiments except OH radical. How did the authors make sure that the reaction pathway followed the proposed mechanisms?*

**Response:**

It is a regret that we did not detect the formation of HONO, $HNO_3$ and $NO_3$ radical due to technique limitation. The effects of these compounds on NOx measurement has been discussed in section 3.2 of the original manuscript. Most of our experiments were conducted in dry condition (0.8% RH), and according to Zhou et al (Geophys. Res. Lett., 2003, 30, 10.1029/2003gl018620), the rate of $NO_x$ generation from $HNO_3$ photolysis was greater than 97% of the total product at RH=0%. So the formation of HONO in our study was estimated to be very low. For larger RH conditions as discussed in section 3.3.3 of the original manuscript, HONO(ads) can be generated due to the reaction of $NO_2$ with adsorbed $H_2O$, which can be desorbed from the surface and released into the gas phase. While according to Shi et al (Environ. Sci. Technol., 55, 854-861,2021), the effect of HONO on $NO_x$ analyzer measurements can be neglected in case of high $NO_x$ concentration in the system. The $NO_x$ concentration in our study in most cases is about 100 ppb, so we think the effect of other products on product distribution and $NO_x$ measurements was negligible

In response to the speculation of $HNO_3$ production, we measured the pH of water extracts in $NO_3^-$-$TiO_2$ systems with and without HCHO, and found that pH was greatly reduced in the presence of HCHO (Figure 7 below). The pH decreases by 1.7% and 2.1% for $KNO_3$-$TiO_2$ and $NH_4NO_3$-$TiO_2$ particles, respectively, suggesting the formation of acidic species such as $HNO_3$(ads) in this study. We have added this results in the revised manuscript as it appears in the section 3.2 Line 282-285: "*We measured the pH of water extracts in $NO_3^-$-$TiO_2$ systems with and without HCHO. It was found that the pH decreased by 1.7% for $KNO_3$-$TiO_2$, suggesting the formation of acidic species such as $HNO_3$(ads) in this study*"

[Figure]

Figure 7. pH values of water extract of $KNO_3$-$TiO_2$ and $NH_4NO_3$-$TiO_2$ particles in the chamber with or without HCHO under 365 nm LED lamps illumination at 293 K and 0.8% of relative humidity.

The generation of $NO_3$ radicals can be indirectly proved by the results in section 3.1 and Figure 1 of the original manuscript, as we have responded and displayed in above Table 1. Another similar example is the published work of our group (Sci. Rep., 2017, 7, 1161). By using the same chamber, the photoreaction rate constants of HCHO on $TiO_2$ and $KNO_3$-$TiO_2$ aerosols under "365 nm lamp" or "365 nm lamp + yellow fluorescence lamp (450–750 nm)" illumination were compared (Figure 8 below). The oxidation rate constants of HCHO over $TiO_2$ were comparable under these two illumination conditions, due to that $TiO_2$ is not sensitive to visible light. However, the rate constant on $KNO_3$-$TiO_2$ aerosol under illumination of both lamps

was lower than that under only the "365 nm lamp", indicating a reduced oxidation rate due to $NO_3$ radical photolysis by visible light. This provides experimental evidence for the existence of $NO_3$ radical.

As we have mentioned in the section 3.3.3 of the original manuscript, in the presence of $H_2O$, in addition to the suggested $NO_3^- - NO_3 \cdot - HCHO - HNO_3$ pathway, there are a variety of $HNO_3$ generation paths, such as the hydrolysis of $N_2O_5$ via the $NO_2 - N_2O_5 - HNO_3$ pathway, the oxidation of $NO_2$ by $\cdot OH$, and the reaction of $NO_3 \cdot$ with $H_2O$, all of which require further consideration and study. It is a regret that due to the technique limitations, we did not detect these species directly. We will dedicate to detect these species by some instruments in the future.

[Figure]

Figure 8. Photoreaction rate constants with light illumination. HCHO photoreaction rate constants on $TiO_2$ or 4 wt.% $KNO_3$-$TiO_2$ aerosol in the condition of 8% RH under light illumination of "365 nm" or "365 nm + yellow fluorescence", respectively. (Sci. Rep., 2017, 7, 1161)

*(2)It is well known that $TiO_2$ can photocatalysis HCHO, can this reaction affect the formation of NOx?*

**Response:**

Yes, HCHO can be degraded in the presence of irradiated $TiO_2$ and will affect the release of $NO_x$, which had been discussed in the original manuscript. We observed the decrease of HCHO concentration both in "$TiO_2$+HCHO" and in "$TiO_2$/$NO_3^-$ +HCHO" systems. The results were shown in Figure S6 of the original manuscript (Figure 9 here). For better understanding, we revised the last sentence "Future studies should

explore whether HCHO affects the photocatalytic renoxification of $NO_3^-$-$TiO_2$." in the second paragraph of section 3.2 Line 265-266 as "*In the following study, the effect of HCHO on the photocatalytic renoxification of $NO_3^-$-$TiO_2$ was explored*".

[Figure]

Figure 9. Photocatalytic degradation curve of HCHO on $TiO_2$ and 4 wt.% $KNO_3$-$TiO_2$ particles under 365 nm LED lamps at 293 K and 0.8% of relative humidity.

The decreased concentration of HCHO during this process can affect the formation of $NO_x$, which can be reflected by the flattening trend of NOx production after 60 min irradiation. This effect had been discussed in the original manuscript, line 247-249: "*The slow stage is due to the photodegradation of HCHO on $KNO_3$-$TiO_2$ aerosols, which led to a decrease in its concentration, gradually weakening the positive effect*". In addition, the amount of $NO_x$ was also significantly reduced under the experiment of low concentration of HCHO (section 3.3.4 of the original manuscript), proving again the important role of HCHO in $NO_x$ release.

*6. The mixture of $HNO_3$ and $TiO_2$ was used to support that $HNO_3$ was an important intermediate to form NOx. However, this logic is not right. If it is right, then any N-contained components mixed with $TiO_2$ that enhanced the generation of NOx could be thought as the intermediates of NOx formation. The direct way to identify the intermediates is to measure them such as FTIR/DRIFTS to measure the adsorption products.*

**Response:**

Thanks for your comments, which will be valuable and helpful for revising and improving the manuscript. What we want to emphasize here is that a hydrogen abstraction reaction was occurred between HCHO and $NO_3$ radical to produce $HNO_3$, with the description shown in equation 5 and Line 252-254 in the original manuscript (*This burst-like generation of $NO_x$ can be ascribed to the reaction between generated $NO_3\cdot$ and HCHO via hydrogen abstraction to form adsorbed nitric acid ($HNO_3(ads)$) on $TiO_2$ particles*). That meant the formed $HNO_3$ came from the original nitrate, and was responsible for the fast $NO_x$ release. The $HNO_3$-$TiO_2$ system is used as a comparison test to demonstrate the proposed mechanism and the photolysis contribution of $HNO_3$ to $NO_x$. FTIR/DRIFTS can be used for detecting species formation, but what we used in our study were nitrates, so no significant change in peak intensity would be observed.

*Minor comments:*

*1.Abstract: many sentences are confusing me! I can't understand what the main meaning of the work. What's the main results. The languages need to be improved.*

**Response:**

Thanks for your comments, which will be all valuable and very helpful for revising and improving the manuscript. In the revised manuscript, we rewrote the abstract for better understanding. The terms of "renoxification" and "photocatalytic renoxification" were stated firstly and then the reaction system was introduced. After that, the experimental results were shown, and the reaction pathway was suggested. The revised abstract is as follows: "*Renoxification is the process of recycling of $NO_3{}^-$/$HNO_3$ into $NO_x$ under illumination, which is mostly ascribed to the photolysis of nitrate. $TiO_2$, a typical mineral dust component, can play its photocatalytic role in "renoxification" process due to $NO_3$ radical formed, and we define this process as "photocatalytic renoxification". Formaldehyde (HCHO), the most abundant carbonyl compound in the atmosphere, may participate in the renoxification of nitrate-doped $TiO_2$ particles. In this study, we established a 400 L environmental chamber reaction*

*system capable of controlling 0.8-70% relative humidity at 293K, with the presence of 1 or 9 ppm HCHO and 4 wt.% nitrate-doped TiO₂. The direct photolyses of both nitrate and NO₃ radical were excluded by adjusting the illumination wavelength, so as to explore the effect of HCHO on the "photocatalytic renoxification". It is found that NO$_x$ concentration can reach up to more than 100 ppb for nitrate-doped TiO₂ particles, while almost no NO$_x$ was generated in the absence of HCHO. Nitrate type, relative humidity and HCHO concentration were found to influence NO$_x$ release. It was suggested that substantial amounts of NO$_x$ were produced via the NO₃⁻-NO₃·-HNO₃-NO$_x$ pathway, where TiO₂ worked for converting "NO₃⁻" to "NO₃·", HCHO participated in transformation of "NO₃·" to "HNO₃" through hydrogen abstraction, and "HNO₃" photolysis answered for mass NOx release. So, HCHO played a significant role in this "photocatalytic renoxification" process. These results were found based on simplified mimics for atmospheric mineral dust under specific experimental conditions, which might deviate from the real situation, but illustrated a possible way of HCHO in influencing nitrate renoxification in the atmosphere. Our proposed reaction mechanism by which HCHO promotes photocatalytic renoxification is helpful for deeply understanding the atmospheric photochemical processes and nitrogen cycling, and could be considered for better fitting of atmospheric model simulations with field observations in some specific scenarios."*

*2. "photocatalysis", "photolysis", "photocatalytic", "photochemical" appeared in the manuscript everywhere, the author should make sure the exact meaning of these words and give the right usage of these words.*

**Response:**

Thanks for your comments. Table 3 illustrates the use of the four words. "Photocatalysis" and "photocatalytic" are used to refer to the chemical reactions that occur when photocatalysts such as TiO₂ are irradiated. The word "photolysis" refers to the breaking of chemical bonds of the substance itself under light, especially ultraviolet light, when there is no photocatalyst. In the manuscript, "photolysis" refers

to the reaction of N-containing species themselves under irradiation. These three can be grouped together as "photochemical reactions". In the manuscript, we use "photochemical" in order to illustrate the broad meaning of such kind of reactions occurred in the atmosphere. For example, "photochemical processes" in the paper refers to atmospheric oxidation and nitrogen cycling. The different words being used to better distinguish the reactions that occur in different situations. We have corrected the inappropriate wording in the revised manuscript, shown also in Table 3 below.

Table 3. Distinction between "photocatalysis", "photolysis", "photocatalytic" and "photochemical".

| Word | Meaning | Usage in our study |
|---|---|---|
| Photocatalysis (noun) | The photocatalytic properties and photocatalytic activity of the compounds. | This word was used only once in the abstract section. In order to simplify the use of words in the paper, we have modified the abstract so that the word is no longer used. |
| Photolysis (noun) | The chemical bonds of the substance itself are broken under light, especially ultraviolet light. | "Photolysis" refers to the reaction of N-containing species themselves under irradiation |
| Photocatalytic (adjective) | Photocatalytic effect of photocatalyst. | The word is used wherever $TiO_2$ is mentioned. |
| Photochemical (adjective) | Series of chemical reactions that occurred under irradiation. | Some sentence such as "..photochemical cycle of HOx radicals.." are still use the word. There is one revision in the revised manuscript. "$NH_4^+$ and NO are photochemically oxidized on $TiO_2$" is Modified to "$NH_4^+$ and NO are photocatalytically oxidized on $TiO_2$". |

3.Line 232-233, the photodegradation of HCHO on $TiO_2$ is not zero-order reaction kinetics, the curve is not a line as shown in Figure S6, which decreased slowly and then fast. The reason for it should be the large amount of adsorption of HCHO on the particle during the long-time injection of HCHO. Besides, the continuous wall loss of particle would result in the change of kinetic coefficient. The concentration of particles and HCHO were too high, and the injection time was too long to give clear kinetic parameters. Generally, the photocatalytic process is supposed to be a first order reaction.

**Response:**

Thanks for your comments, which will be all valuable and very helpful for revising and improving the manuscript, as well as the important guiding significance to our researches.

(1) Regarding the question of reaction kinetics

We fitted the photocatalytic degradation curves of HCHO on $TiO_2$ and $KNO_3$-$TiO_2$ using the data of Figure S6 in the original manuscript for zero-order reaction and first-order kinetics, respectively. As shown in Figure 10 below, for both $TiO_2$ and $KNO_3$-$TiO_2$ systems, the correlation coefficients ($R^2$) of the zero-order fitting is larger than that of the first-order fitting, so the photocatalytic degradation of HCHO on $TiO_2$ and $KNO_3$-$TiO_2$ fit zero-order kinetics. In order to show the curves more clearly, we marked the $KNO_3$-$TiO_2$ line in red color with the new figure shown as Figure S7 in the revised manuscript (Figure 11 below).

[Figure]

Figure 10. The comparison of correlation coefficients of zero- and first-order reaction curves. The reaction systems are "HCHO + $TiO_2$" (a) and "HCHO + 4 wt.% $KNO_3$-$TiO_2$" (b), both under 365 nm LED lamps at 293 K and 0.8% of relative humidity.

[Figure]

Figure 11. Photocatalytic degradation curve of HCHO on $TiO_2$ and 4 wt.% $KNO_3$-$TiO_2$ particles under 365 nm LED lamps at 293 K and 0.8% of relative humidity.

(2) Regarding the question of possible change of kinetic coefficient

As shown in Figures 3-5 above, the particle size distribution in the chamber can be maintained for several hours, with the number concentration in the chamber decreased by no more than 10% per hour. In addition, the good correlation coefficient of 0.9779 and 0.9745 for $TiO_2$ and $KNO_3$-$TiO_2$, respectively, shown in Figure 11 also reflected that the kinetics fitting and rate constants are believable.

(3) Regarding the injection time and the adsorption of HCHO

We are sorry to make the reviewer misunderstand the operation sequence of the experiment. Figure S3 in the original manuscript is the conditional experiment results to show the HCHO adsorption in the dark before and after particles injection. We think it is Figure S3 that make the reviewer misunderstand. So we modified Figure S3, shown below as Figure 12 (Figure S4 in the revised manuscript). The operation sequence of the experiment is as follows. HCHO gases was flowed 10 min into the clean chamber firstly under dark conditions. As can be seen in Figure 12, HCHO can get equilibrium around 90 min. After that, no obvious decrease of HCHO was observed meaning that no further HCHO was adsorbed by the chamber. Then the particles were introduced into the chamber instantly. The concentration of HCHO began to decrease upon particles injection and need 60 min to reach its second adsorption equilibrium. After that, HCHO concentration can be maintained in the dark

for several hours, indicating no further adsorption of HCHO by the chamber and the particles. In irradiation experiments, we waited for both HCHO and particles to reach stable before turning on the lights. The related description of the operation has been revised in the manuscript, Line 185-194.

[Figure]

Figure 12. The conditional experiments of HCHO concentration in the environmental chamber in the dark before and after the introduction of particles over time.

*4.I can't understand why the authors used KNO₃ and HNO₃ to mixture with TiO₂. In Ma's work, they indicated the NOx concentration formed from KNO₃ was the lowest. KNO₃ only accounts for small proportion in the atmospheric particles. HNO₃ is acid species and can react with TiO₂, which would result in the component changes in this mixture particles. I think that the components in this mixture particles were different from the discussion in the article.*

**Response:**

Thanks for your comments, which will be all valuable and very helpful for revising and improving the manuscript, as well as the important guiding significance to our researches. We have thought deeply about the experimental design, and answer the comments point by point.

*(1)I can't understand why the authors used KNO₃ and HNO₃ to mixture with TiO₂. In Ma's work, they indicated the NOx concentration formed from KNO₃ was the lowest. KNO₃ only accounts for small proportion in the atmospheric particles.*

**Response:**

Although $KNO_3$ accounts for small proportion in the atmospheric particles, it is also important for atmospheric chemistry studies. Wang et al (Sci. Total Environ., 2019, 660: 47-56) found that in winter haze episodes, the formation of $KNO_3$ particles in the droplet-mode plays an important role in the increase of $PM_{2.5}$ concentration. So the $KNO_3$-related chemical reactions are important for the study of high pollution weather. In addition, in laboratory studies of nitrate photolysis, $KNO_3$ is still used as a model particle. For example, Yang et al (EP, 2018, 243: 679-686) used $KNO_3$ to study the effect of nitrate photolysis on HONO formation in the presence of humic acid; Xu et al (JES, 2021, 102: 198-206) used $KNO_3$ to study the effect of $TiO_2$ crystal structure on $NO_2$ and HONO emission from the nitrates photolysis.

In our study, $NH_4NO_3$ was also used for the study and the results were compared with those of $KNO_3$ to investigate the effect of cations on the photocatalytic renoxification process (as discussed in section 3.3.1). Similar to Ma's findings, lower $NO_x$ release was observed from $KNO_3$ composite compared to $NH_4NO_3$ composite. We think this result may be caused by the blocking effect of $K^+$ on $NO_3^-$, which has been explained in the original manuscript text line 211-215.

*(2)$HNO_3$ is acid species and can react with $TiO_2$, which would result in the component changes in this mixture particles. I think that the components in this mixture particles were different from the discussion in the article.*

**Response:**

Some researches characterized the structure of $TiO_2$ after acid treatment and found no changes. For example, Wang et al. (J Mater Sci: Mater Electron, 2021, 32: 21083) treated $TiO_2$ with 98% concentrated sulfuric acid for 12 h, and demonstrated by XRD and XPS that acid treatment does not change the structure, elemental composition and chemical state of $TiO_2$ (Figure 13 below). In our experiment, a low content of $HNO_3$ (0.002 mol) was used to avoid the possible changes in composition of $TiO_2$. So it is estimated that the components of the particles would not change. We will make structure characterization to ensure this in our future study.

[Figure]

Figure 13. XRD images of TiO$_2$ nanotubes, g-C$_3$N$_4$, A1, A2 and A3. (A1: TiO$_2$ nanotubes after 12 h of H$_2$SO$_4$ treatment; A2: acid-treated TiO$_2$ compounded with g-C$_3$N$_4$; A3: TiO$_2$ without acid treatment compounded with g-C$_3$N$_4$) (J Mater Sci: Mater Electron, 2021, 32: 21083)

*5.OH radical was measured by ESR in this study. However, the role of OH radical has not been discussed. And the OH radical generated in different particles and under different conditions have not been compared and analyzed. Besides, NO$_3$ radical was proposed to be the important intermediates in the reaction. Why did not the authors measure NO$_3$ radical?*

**Response:**

  Thanks for your comments, which will be all valuable and very helpful for revising and improving the manuscript, as well as the important guiding significance to our researches. The detection of OH radicals is for TiO$_2$ and Arizona dust, which is intended to demonstrate that photocatalysis process in these two particles do exist. In particular, the presence of OH radicals in the Arizona dust upon irradiation provides evidence that the findings of our study have practical implications. The emergence of NO$_x$ observed in the chamber demonstrated that HCHO promoted the renoxification of ATD particles (Figure S9 in the original manuscript). This result suggests that mineral dust containing photocatalytic semiconductor oxides such as TiO$_2$, Fe$_2$O$_3$, and ZnO could promote the conversion of granular nitrate to NO$_x$ in the presence of

HCHO. The above discussions have been given in the original manuscript in lines 298-310.

TiO$_2$ produces OH radicals under UV illumination, which is well established in the field of photocatalysis (Xu et al., Appl. Catal., B, 2018, 230, 194-202). We provide this data to demonstrate that TiO$_2$ can be excited under our irradiation conditions and will exert its photocatalytic effect. In this case, other samples with TiO$_2$ as the main component would also generate OH radicals, although the amount may vary, but is not the main focus of our study. As what has been suggested, NO$_3$ radicals (coming from h$^+$ with NO$_3^-$) is the key species responsible for the formation of large amounts of NO$_x$. Unfortunately, NO$_3$ radical was not detected currently due to instrument limitations. Such measurement will be considered in our future studies. However, as discussed in the manuscript, the presence of NO$_3$ radicals was indirectly illustrated.

*6.Weight percentage was used to quantify nitrate in the mixed particle. However, different nitrate has different molecule weight, which would result in that the molar concentrations of different nitrates with the same weight percentage were different. For example, the molar concentration of N in 4 wt % HNO$_3$-TiO$_2$ is higher than that of N in 4 wt % KNO$_3$-TiO$_2$. This effect should be considered in the formation of NOx.*

**Response:**

Thank you for your comments, which will be all valuable and very helpful for revising and improving the manuscript. We agree with the reviewers that different nitrates with the same weight percentage owns the difference in mole of N, which will cause the difference in NO$_2$ formation. To exclude this effect, we replotted Figure 2 of the original manuscript (here Figure 14 shown below) with mole normalized ppb as the NO$_2$ formation unit rather than ppb. As can be seen from Figure 14, the main conclusion is the same, with HNO$_3$–TiO$_2$ presented much higher activity than KNO$_3$–TiO$_2$ in the presence of HCHO. The discussion related to this Figure has been modified in section 3.2 of the revised manuscript.

[Figure]

Figure 14. Effect of formaldehyde on the renoxification processes of different nitrate-doped particles at 293 K and 0.8% of relative humidity. 365 nm LED lamps were used during the illumination experiment. The initial concentration of HCHO was about 9 ppm.

**Response to the reviewer 2's comments:**

**Referee #2: General Evaluation**

*Liu et al. investigated the possible renoxification processes occurring on TiO₂ particles and mineral dust particles in presence of adsorbed nitrate, or HNO₃, in presence of HCHO. They suggest that HCHO and TiO₂ have a significant synergistic effect on the photocatalytic renoxification via a NO₃-NO₃-HCHO-HNO₃-NOx pathway, in which adsorbed HCHO may react with nitrate radicals through hydrogen abstraction to form HNO₃ on the surface, resulting an enhanced generation of NOx.*

*Overall this is an important topic, which certainly falls within the scope of journal Atmospheric Physics and Chemistry.*

**Response:**

Thanks for your comments, which will be all valuable and very helpful for revising and improving the manuscript, as well as the important guiding significance to our researches. We have thought deeply about the experimental design, and answer the comments point by point.

**Comments on Preprint acp-2022-6:**

*1. The experiments presented here were performed in a simulation chamber consisting of a 400 L polyvinyl fluoride (PVF) bag filled with synthetic air. In such a small baga, the life time of particles is expected to be very short, as shown also in figure S2. Surprisingly, the authors do observe, after some induction time, a stable size distribution over hours. How can this happen? Is it an indication of some dynamic interactions with the chamber's walls, during which particle adsorb and desorb constantly? Such process may be induced to some air turbulences around the bag, or through its deflation during the experiments (by the way, was the bag closed and its volume shrinking or was it flushed by pure air all the time during the experiments?). Anyhow, this is a strong indication that wall effects may play a significant role in the reported experiments. Therefore, a thorough discussion of these effects has to be included in the manuscript.*

**Response:**

Thanks for your comments. We are sorry that the description of operating procedures is not so clear. So we modified this in the revised manuscript. Here, we would like to describe the operation briefly. The sequence of the experimental operation of the chamber is as follows: cleaned by deionized water, dried totally, then inflated by synthetic air to a certain volume, then HCHO introduction, and then particles introduction instantly by a high pressure air flow. After the concentrations of both HCHO and particles became stable, the lamps were turned on and the concentrations of $NO_x$ were monitored. The chamber was closed during the entire experiment. Once the particles entered into the chamber, the number concentration began to decrease due to the wall effect. After 30 min, the size distribution of both $KNO_3$-$TiO_2$ and $TiO_2$ particles got stable and can sustained for several hours (with Figures 3-5 shown above, see response to Referee #1 Major comments 3(3)). The left picture in the following Figure 4 shows the first 60 min of the size distribution in the dark, and the right picture of the size distribution during the irradiation. Figure 5 is another batch experiment of $TiO_2$ in the chamber with the same operation, and it shows very similar size distribution from 0-60 mins and 120-180 mins. During the experiment, we strictly controlled the same experimental condition before the start of

each experiment and turned on the lamps only after the particles and HCHO concentration reached stability.

The corresponding revisions are as follows: (1) The injection of the particles has been emphasized in section 2.3 Line 174-175 of the revised manuscript: "*then 75 mg particles were instantly sprayed into the chamber by a transient high-pressure airflow*"; (2) The sentences "*For the chamber operation, we completely evacuated the chamber after every experiment, then cleaned the chamber walls with deionized water and then dried by flushing the chamber with ultra-zero air to remove any particles or gases collected on the chamber walls*" was added to the section 2.3 Line 168-171 of the revised manuscript.

In addition, we checked our size distribution data of different samples, and found that the number concentration is not that high and usually around 4000 particle/cm$^{-3}$ when reaching stability. So Figure S2 in the original manuscript was deleted and replaced by the following Figure 3 (Figure S3 in the revised supplement).

*2. The main conclusion of this work is that adsorbed HCHO reacts with adsorbed nitrate radicals, promoting NOx formation. This assumes that this reaction is faster than the one of HCHO with the photochemically generated holes on the surface of the mineral. Is this justified by any means? HCHO being efficiently degraded on illuminated TiO₂, one would expect that this VOCs may compete with the nitrate anions to react with the holes, with the synergy between nitrate anions and HCHO vanishing at low surface coverage (where both compounds would react with the holes with no interactions with co-adsorbed species). Is this observed here?*

**Response:**

Thanks for your comments. As the reviewer mentioned, HCHO can react with the photochemically generated holes on the surface of the mineral. The photodegradations of HCHO on $TiO_2$ and $KNO_3$-$TiO_2$ particles were observed in this study. As shown in text line 230-239 of the original manuscript: "*Atmospheric trace gases can undergo photocatalytic reactions on the surface of TiO₂ (Chen et al., 2012). As the illumination time increased, the concentration of HCHO showed a linear downward trend, which was consistent with zero-order reaction kinetics (Figure S6).*

*The zero-order reaction rate constants of HCHO on TiO₂ and 4 wt.% KNO₃-TiO₂ particles were $9.1 \times 10^{-3}$ and $1.4 \times 10^{-2}$ ppm min⁻¹, respectively, which were much higher than that for gaseous HCHO photolysis (Shang et al., 2017). We suggested that the produced NO₃· contributed to the enhanced uptake of HCHO. Therefore, we suggest that NO₃· production contributed to enhanced HCHO uptake. Future studies should explore whether HCHO affects the photocatalytic renoxification of NO₃⁻-TiO₂*". Besides the photodegradation of HCHO on excited TiO₂ particles, higher photodegradation rate of HCHO was observed on KNO₃-TiO₂ particles. As for if there is a simultaneously decrease of nitrate, we once compared the absorption spectra of the extracts of KNO₃-TiO₂ particles before and after reactions, with results shown below as Figure 15. It can be seen that nitrate content was decreased after reaction, meaning the vanishing of the nitrate.

By now, we do not know which of the reactions is faster ("HCHO+hole" or "HCHO+NO₃ radical"), but due to the high amount of both HCHO and TiO₂ used in this study, on one side, there are enough holes to react with HCHO and nitrate at the same time, and on the other side enough remaining HCHO to react with NO₃ radical.

[Figure]

Figure 15. Absorption spectra of the extracts of KNO₃-TiO₂ particles before and after reactions.

*3. Spraying mixture of SiO₂ and TiO₂, would result in an externally mixed aerosol, isn't it? Then it should represent an experiment with the TiO₂ particles simply being diluted as compared to the pure TiO₂ experiment.*

**Response:**

Thanks for your comments. There are two reaction systems in our experiments. The first system, in which HCHO was not introduced, was intended to investigate the positive effect of $TiO_2$ on the renoxification process. In this system, we used 4 wt.% $KNO_3$-$SiO_2$ and 4 wt.% $KNO_3$-$TiO_2$ (1 wt.%)/$SiO_2$. Here $TiO_2$ (1 wt.%)/$SiO_2$ was prepared first and then 4 wt.% $KNO_3$ was composited. In the second system, HCHO was introduced to investigate the synergistic positive effect of $TiO_2$ and HCHO on the renoxification process. The samples used in this system were 4 wt.% nitrate-$TiO_2$. Here 4 wt.% nitrate was composited to the pure $TiO_2$ particles. Note that, in the first system, the main particle is $SiO_2$ and the content of $TiO_2$ is only 1 wt.% relative to $SiO_2$. While in the second system, the main particle is $TiO_2$. So there is a large difference in the $TiO_2$ mass in these two particles. It is not a simple dilution of $TiO_2$ with $SiO_2$.

*4. An effect of acidity is observed and explained by the enhanced photolysis of $HNO_3$. Could an alternative explanation arise for the chemistry of $O_2^-$? This superoxide would react with $H^+$ inducing $HO_2$ chemistry that may change a series of surface reactions. Could the authors comment on that?*

**Response:**

Thanks for your comments. Our experiments were performed under dry conditions, so there is little $H^+$ to react with $O_2^-$ to produce $HO_2^-$, and the mass release of $NO_2$ was ascribed to the photolysis of $HNO_3$. However, in our study of relative humidity as an influencing factor, there are some possible effects arising from $O_2^-$. Under high humidity, adsorbed $H_2O$ can behave as scavenger of photogenerated holes ($h^+$), so as to make photogenerated electron ($e^-$) reacted with oxygen, resulting in the generation of $O_2^-\cdot$ (Eq. 1). $O_2^-\cdot$ can combine with $H^+$ to generate $HO_2\cdot$ (Eq. 2). Subsequently, NO undergoes a two-step photocatalytic degradation on $TiO_2$: oxidation of NO by $HO_2\cdot$ to $NO_2$ (Eq. 3) and oxidation of $NO_2$ by $OH\cdot$ to $HNO_3$ (Eq. 4) (Dalton et al., EP, 2002, 120: 415-422; Devahasdin et al., J Photoch. Photobio. A, 2003, 156: 161-170). Therefore, higher relative humidity can affect NO and $NO_2$

production due to $HO_2$ chemistry, which has been discussed in section 3.3.3 of the original manuscript.

$$O_2 + e^- \rightarrow O_2^- \cdot \qquad (1)$$
$$O_2^- \cdot + H^+ \rightarrow HO_2 \cdot \qquad (2)$$
$$NO + HO_2 \cdot \rightarrow NO_2 + OH \cdot \qquad (3)$$
$$NO_2 + OH \cdot \rightarrow HNO_3 \qquad (4)$$

**Response to the reviewer 3's comments:**

**Referee #3: General Evaluation**

*The author reported formaldehyde may have synergistic effect in photocatalytic renoxification of nitrate with $TiO_2$, in order to explain the difference between field data and modeling result. The article focuses on the significant synergistic effect, i.e., HCHO and $TiO_2$ have on photocatalytic reactions and providing one possible reaction pathway- $NO_3$-$NO_3$-HCHO-$HNO_3$-NOx. These findings improve the understanding of the role of reactions between organic components and nitrate in the chemical and physical properties of aerosol particles in low relative humidity region. It has significant implication in the research of atmosphere and air pollution, but some issues in the article must be improved. I recommend accept this article after resolve those issues. There are the comments I have for this work:*

**Response:**

Thanks for your comments, which will be all valuable and very helpful for revising and improving the manuscript, as well as the important guiding significance to our researches. We have thought deeply about the experimental design, and answer the comments point by point.

**Comments on Preprint acp-2022-6:**

*1. There are some misdescriptions in the manuscript. Like:*

*We suggested that the produced $NO_3 \cdot$ contributed to the enhanced uptake of HCHO.*

*Therefore, we suggest that $NO_3 \cdot$ production contributed to enhanced HCHO uptake. (Line 236, Page 12)*

*photochemical cycle of HOx radicals in the atmosphere and the formation of (Line 448, Page 25)*

**Response:**

Sorry for our carelessness about these sentences. In the revised manuscript, we have deleted the second half of the first sentence mentioned by the reviewer. The revised sentence is: "*We suggested that the produced $NO_3\cdot$ contributed to the enhanced uptake of HCHO*". For the second sentence, HOx means some reactive species such as $HO_2$ etc. Due to there is not many discussion about $HO_2$ in the manuscript, we revised the sentence as: "*photochemical cycle of reactive radicals in the atmosphere and the formation of...*"

*2. Please explain why 4 wt.% $KNO_3$-$TiO_2$(1 wt.%)/$SiO_2$ underwent reaction to release NOx, while 4 wt.% $KNO_3$-$SiO_2$ and 4 wt.% $KNO_3$-$TiO_2$ not in same condition.*

**Response:**

Thanks for your comments. In our experiment, two reaction systems existed. The reaction systems and aims of the two kinds of light sources were different, which were summarized in the Table 1 (see Response to Referee #1 Major comments 1(3)). The first system, in which HCHO was not introduced, was intended to investigate the positive effect of $TiO_2$ on the renoxification process. In this system, we used 4 wt.% $KNO_3$-$SiO_2$ and 4 wt.% $KNO_3$-$TiO_2$ (1 wt.%)/$SiO_2$. The light source for this system is tube lamps. In the second system, HCHO was introduced to investigate the synergistic positive effect of $TiO_2$ and HCHO on the renoxification process. Our samples used in this system study were 4 wt.% nitrate-$TiO_2$. The light source for this system is LED lamps. LED lamps do not contain visible light component, so the effect of $NO_x$ release from the photolysis of $NO_3$ radicals under visible light could be excluded. So 4 wt.% $KNO_3$-$TiO_2$(1 wt.%)/$SiO_2$ (or 4wt.% $KNO_3$-$SiO_2$) and 4wt.% $KNO_3$-$TiO_2$ are not used in the same reaction system. The 4 wt.% $KNO_3$-$SiO_2$ did not release $NO_x$ because $SiO_2$ has no photocatalytic activity, as we discussed in section 3.1 of the original manuscript.

*3. In Figure S3, the concentration of HCHO and TiO$_2$ particles reached stable in 60 min after introduced into experimental chamber, but other experiments almost stared in -30min, did HCHO and TiO$_2$ have been stable?*

**Response:**

Thanks for your comments. We are sorry to make the reviewer misunderstand the operation sequence of the experiment. As we stated in response to Referee #1 Minor comments 3(3), Figure S3 in the original manuscript is the conditional experiment results to show the HCHO adsorption in the dark before and after particles injection. We have modified Figure S3 (Figure S4 in the revised manuscript), shown above as Figure 12. The related description of the operation has been revised, see Line 185-194 of the revised manuscript: "*In order to know the HCHO adsorption before and after the particles' introduction, we conducted a conditional experiment in the dark. It can be seen from Figure S4 that it took about 90 min for the concentration of HCHO to reach stable, and can be sustained. Then, 75 mg TiO$_2$ or NO$_3^-$/TiO$_2$ powders were introduced instantly and the concentration of HCHO decreased upon the introduction. It took about 60 min for HCHO to reach its second adsorption equilibrium, and the concentration of HCHO can be stable for several hours in the dark. Therefore, for the irradiation experiments, the particles were injected at 90 min after HCHO's introduction, and the lamps were turned on at 60 min after the particle's introduction.*" The -30 min in others figures refers to the concentration measurements of NO$_2$ in the dark, and the adsorption of HCHO or particles was begun long before that.

*4. The BET of TiO$_2$ nanoparticles is huge, the uptake of HCHO in TiO$_2$ nano-particles can't be ignored. The photodegradation of HCHO on TiO$_2$ and 4 wt.% KNO$_3$-TiO$_2$ particles should start after adsorption and desorption balance.*

**Response:**

Thanks for your comments. In our study, the uptake of HCHO on TiO$_2$ nano-particles was considered. As shown in Figure 12 above of conditional experiments of HCHO adsorption in the dark, it needs 60 min (from 180-240 min in

the Figure) for HCHO to get stable after the particles' injection. For each experiments, we waited at least 60 min to ensure that the adsorption equilibrium has been reached.

*5. In section 3.3.1, 4 wt.% KNO₃-TiO₂ particles release less NOx than Equal amounts of 4 wt.% NH₄NO₃-TiO₂ particles at 293K and 0.8% relative humidity, which may be the result of the Relative molecular mass difference between KNO₃ and NH₄NO₃.*

**Response:**

Thanks for your comments. We agree with the reviewers that different nitrates with the same weight percentage owns the difference in mole of N, which will cause the difference in $NO_2$ formation. To exclude this effect, we replotted Figure 3 of the original manuscript (here Figure 16 shown below) with mole normalized ppb as the $NO_2$ formation unit rather than ppb. As can be seen from Figure 16, the main conclusion is the same, with $NH_4NO_3$-$TiO_2$ presented higher activity than $KNO_3$-$TiO_2$ in the presence of HCHO. The discussion related to this Figure has been modified in section 3.3.1 Line 352-354 of the revised manuscript: "*Similar as Figure 2, millimole normalized ppb was used in order to compare the amount of $NO_x$ release for different kinds of nitrate with same percentage weight*".

[Figure]

Figure 16. Effect of formaldehyde on the renoxification processes of 4 wt.% $NH_4NO_3$-$TiO_2$ and 4 wt.% $KNO_3$-$TiO_2$ particles at 293 K and 0.8% of relative humidity. 365 nm LED lamps were used during the irradiation experiment. The initial concentration of HCHO was about 9 ppm.

*6. In section 3.3.4. more different initial concentration of HCHO should be test to find out from which content the positive effect become weakening.*

**Response:**

Thanks for your comments. There is an equilibrium of $NO_2$ release in our reaction system, one is the photocatalytic oxidation reaction between $NO_x$ and ROS (generated from excited $TiO_2$), and the other is the renoxification of $NO_3^--TiO_2$ particles. As discussed in the section 3.3.4 in original manuscript, these two competitive reactions will determine the up or down of $NO_2$.

We measured the variations of HCHO concentration both in its high and low system, with results shown in Figures S6 (Figure S7 in the revised manuscript) and Figure 17 below. The rate constant of HCHO decay are $1.4\times10^{-2}$ ppm min$^{-1}$ and $1.3\times10^{-3}$ ppm min$^{-1}$, respectively. The $NO_2$ generation rate in Figure 2 and Figure S10 (Figure S13 in the revised manuscript) are 1.2 ppb min$^{-1}$ and 0.1 ppb min$^{-1}$. So the HCHO concentration, HCHO decay rate and $NO_2$ generation rate all decreased a factor of 10. This coincidence gives us a clue that there may be some connections among these parameters. Another clue is that from Figure S10 (Figure S13 in the revised manuscript), it can be seen that the concentration of $NO_2$ begin to decrease at the time of 50 min, and corresponds to about 0.95 ppm of HCHO (as shown in Figure 17). So it is indicated that below 0.95 ppm of HCHO, the reaction between $NO_x$ and ROS is dominant. More experimental evidences regarding the point of HCHO concentration making positive effect weaken need further investigation. Atmospheric HCHO concentrations are generally very low. However, in cities with high traffic density, because combustion produces emissions, HCHO concentrations will be much higher than normal. In the indoor environment, HCHO levels can increase due to smoking, emissions from gas stoves and furniture, and can reach up to around 0.4 ppm (Formaldehyde. In: Wood dust and formaldehyde. Lyon, International Agency for Research on Cancer, 1995, 217-362). So, we assume that the positive effect of HCHO on the renoxification may still exist at some specific situation with its high concentration.

[Figure]

Figure 17. Photodegradation curves of low concentration formaldehyde on 4 wt.% KNO$_3$-TiO$_2$ particles under 365 nm LED illumination.

It is worthy of noting that although there usually not so much high concentration of HCHO, there are many other organics in the atmosphere which can provide hydrogen atoms to behave similar role as HCHO. As discussed in line 459-465 of the original manuscript: "*The effect of low-concentration HCHO on the renoxification of NO$_3^-$-TiO$_2$ particles requires further investigation. However, many types of organics provide hydrogen atoms in the atmosphere, including alkanes (e.g., methane and n-hexane), aldehydes (e.g., acetaldehyde), alcohols (e.g., methanol and ethanol), and aromatic compounds (e.g., phenol) that react with NO$_3$· to produce nitric acid (Atkinson, 1991). These organics, together with HCHO, play similar positive roles in photocatalytic renoxification and, therefore, influence NO$_x$ concentrations*". We also believe that the effect of more different concentrations of HCHO on the renoxification of NO$_3^-$-TiO$_2$ particles deserves to be studied.

**Second response to the reviewer 1's comments:**

*1. The authors stated that "DRIFTS measurements were made to test their homogeneity". The particles were laid and aggregated on the holder in DRIFTS experiment, not the real particles suspended by sprayed into the chamber. I suggested the authors to collect the particles from the chamber to do SEM or TEM, which can*

*give the clear composition of the suspended particles in the chamber, to see whether the particle is physical mixing or chemical dopped.*

**Response:**

DRIFTS is a kind of instrument which can measure the composition of materials. The observed proportional increase of nitrate peak areas in the $TiO_2$-nitrate composite samples shown in Figure S2 is a good evidence that the prepared samples are homogeneously mixed. Many researches focusing on "renoxification effects" applied similar preparation method as ours to get composite particles (Ndour et al., Geophys. Res. Lett., 2009, 36, L05816; Monge et al., Phys. Chem. Chem. Phys., 2010, 12, 8991-8998), including the article (Ma et al., EST, 2021, 55, 8604) mentioned by the reviewer in his last comment letter. Spectral technique was used to examine the composition of the composite samples. For example, Ma et. al applied micro-Raman spectrometry to confirm that "the form of nitrate was kept during the sample preparation process" (Ma et al., EST, 2021, 55, 8604).

We collected the particles from the chamber to conduct TEM characterization as suggested by the reviewer. Figure 18 displays the TEM and HRTEM images of the $TiO_2$ and 4 wt.% $KNO_3$-$TiO_2$ samples. It can be seen that with small amount of nitrate doping the morphology of the two samples is similar. The lattice-fringe distance of 0.352 nm in $TiO_2$ sample, and the additional lattice-fringe distance of 0.246 nm in $KNO_3$-$TiO_2$ sample, can be ascribed to the interplanar spacing of (101) lattice plane of anatase $TiO_2$ (Zhang et al., Appl. Catal. B, 2013, 142, 249-258) and (111) lattice plane of TiN (Ma et al., Energy Storage Mater., 2022, 44, 180-189), respectively. This suggests that there is chemical bond formed in the $KNO_3$-$TiO_2$ sample. In addition, the obvious enhancement of NOx release in "$KNO_3$-$TiO_2$ + HCHO" (corresponds to 110 ppb at 120 min in Figure 2) system compared to "$TiO_2$ + HCHO" system (3 ppb at 120 min in Figure S5) also indicates that $KNO_3$ and $TiO_2$ make the effects together in the chamber.

[Figure]

Figure 18. TEM and HRTEM images of TiO₂ (a, c) and 4 wt.% KNO₃-TiO₂ (b, d) samples.

*2. The quantity and the wall loss of particles still have large uncertainty, and this defect wasn't well solved. The size of those symbols in Figure 3-5 is too large to clearly see the difference of particle size distribution at different time. Why did the authors not put all the plots in one figure? I feel that these three figures have some differences. The particle number is very important for the quantification and kinetics study, but we can see "about" in front of most particle numbers description. As I pointed before, large wall loss of particles must exist after the particles sprayed in the chamber. The value of 75mg/250L particles were injected into the chamber, only 8000 particle/cm3 (dN/dLog(dp)) or 4000 particle/cm3 were detected for the most abundant diameter particle, where were the other particles? were they still in the chamber? Heterogeneous reactions can also occur on the wall! Larger particle can*

*easily deposit on the wall, which also can be found from the difference of the particle distributions between 0 min and hundreds of minutes later in Figure 3-5. Besides, the detected particle size was only up to 250 nm, how about those larger particles? These results were not enough to predict the wall effect.*

**Response:**

We redrew the size distribution of 4 wt.% $KNO_3$-$TiO_2$ and $TiO_2$ and plotted them together with suitable symbols size, as suggested by the reviewer. It can be seen in Figure 19 that the size distributions of $KNO_3$-$TiO_2$ and $TiO_2$ samples are similar, with both reach stable after about 60 min. The peak number concentration is averaged of 3991 and 3886 particle/cm$^{-3}$ during illumination period for $KNO_3$-$TiO_2$ and $TiO_2$ sample, respectively, indicating that the repeatability of the introduction of particles into the chamber is good and the comparison among different samples in our study is reliable. This can be attributed to the strict cleaning of the chamber and the same operation of each batch experiment. In the new revised supplement, we used this Figure 19 to replace Figure S3, and the description was added in the new revised manuscript in section 2.3.

The number concentration at different illumination time of each sample is in a range, so we used the word "about" in the response file. We averaged the number concentration after illumination with the value showing above. As for the size of the particles, the SNPS-20 we used can only detect the size distribution range of 7~820 nm, so there is no data about larger particles. However, we can deduce the particle size based on the $TiO_2$ powder we used. It is a commercial product with particle size of 10-40 nm. So the size of $TiO_2$ and $NO_3^-$-$TiO_2$ samples would not become larger than micrometer in the chamber. This can also be confirmed by the TEM images shown in Figure 1 above with both $NO_3^-$-$TiO_2$ and $TiO_2$ particles collected from the chamber having size around 15 nm.

It is true that there is wall loss of particles in the chamber and the deposited particles are still in the chamber. We mentioned this in the original as well as revised manuscript in section 2.3: "*the particle number concentration of $KNO_3$-$TiO_2$ or $TiO_2$ sample decreased rapidly owing to wall effect including the possible electrostatic*

*adsorption of the particles by the environmental chamber"*. The wall effect cannot be avoided in the chamber experiments and we had conducted series of comparative experiments including with or without HCHO, with or without $TiO_2$, with or without illumination, and lamp with or without visible light etc to highlight the positive effect of formaldehyde on the release of $NO_x$. Moreover, it is a batch experiment as stressed in the manuscript and strict cleaning and operation were conducted to ensure the repeatability, which can also be reflected by the good control of our blank data (Figures S5, S6 and S9).

[Figure]

Figure 19. Particle size distribution of 4 wt.% KNO₃-TiO₂ (a), TiO₂ (b) and comparison of 4 wt.% KNO₃-TiO₂ (red line) and TiO₂ (blue line) (c) in environmental chamber with time. (60 minute is the time of turning on the lamps)

*3. The proposed mechanisms still couldn't convince me. What's the main significance of this article? The quantity of NOx has great uncertainties, and most of the mechanisms were deduced from so many hypotheses. Though some phenomena and products were observed and detected in previous studies, those proposed key intermediates and products were not directly measured in your experiment. If all those phenomena could be deduced and expected from previous studies, there is no need to do such experiments. Some supplementary experiments are needed, not in the future but in the present work!*

**Response:**

Here, we would like to stress the background, research design and significance of our work, as have been described in the manuscript. The active nitrogen species (HONO and NOx) have important impacts on the atmospheric oxidative capacity and the transformation of many atmospheric species. Nitric acid and nitrate ($HNO_3/NO_3^-$) are not only the final sinks of NOx in the atmosphere, but also one of its important sources. The "renoxification" process of NOx generation from $NO_3^-$ is easily overlooked. A fast photochemical renoxification rate of adsorbed $HNO_3/NO_3^-$ to active nitrogen species was detected on real urban $PM_{2.5}$ (EST, 2020, 54, 3121-3128). Ninneman et al (ACS Earth Space Chem., 2020, 4, 1985-1992) found that the "renoxification" of nitrate to NOx is an important source of NOx in rural New York in winter, especially at low $O_3$ production rates. The transport, transformation, and lifetime of NOx are key factors affecting regional air quality, so the study of the "renoxification" process is of great environmental importance.

In previous studies of the "renoxification" process, it was generally believed that NOx was generated from $NO_3^-$ photolysis ($\lambda \leq 350$ nm). Mineral dust mixing with nitrate is ubiquitous in the atmosphere. Our study found that there is a new NOx generation pathway in the coexistence of $TiO_2$, nitrate and HCHO, with HCHO behaving as a proton-donor. We divide the "renoxification" process of nitrate on the surface of $TiO_2$ particles into "photolysis renoxification" and "photocatalytic renoxification", which can more clearly understand the promoting effect of $TiO_2$ photocatalytic performance on the "renoxification" process. Different from the

traditional "photolysis renoxification" (caused by $NO_3^-$ photolysis), "photocatalytic renoxification" refers to the "renoxification" process involving $h^+$ and $NO_3$ radicals based on the photocatalytic properties of $TiO_2$.

In this study, the direct photolysis of $NO_3^-$ was excluded, and the "renoxification" of $NO_3^-$ due to the photocatalytic properties of $TiO_2$, i.e. "photocatalytic renoxification", was investigated separately by monitoring the irradiated wavelength. The role played by HCHO on NOx generation companied with $TiO_2$'s photocatalytic performance was investigated for the first time. The introduction of HCHO is a break from the traditional way of studying atmospheric reactive nitrogen cycle and transformation, and makes it go forward to the real atmospheric situation where multiple pollutants exist simultaneously. The large increase in the release of NOx would have some effects on the formation of important atmospheric oxidants such as OH radicals and $O_3$. The results of this study will help for assessing the importance of "photocatalytic renoxification" process on atmospheric NOx concentration with the presence of volatile organic pollutants.

The promotion of "photocatalytic renoxification" of $NO_3^-$-$TiO_2$ particles by HCHO observed in our study cannot be predicted from the previous references. We conducted systematic chamber simulation experiments and obtained the conclusions based on so many comparative experiments and references analyses. $NO_3$ radical has been proposed to generate in the nitrate-$TiO_2$ reaction system, but was not detected directly, in many researches (Ndour et al., Geophys. Res. Lett., 2009, 36, No. L05816; Rubasinghege et al., J. Phys. Chem. Lett., 2010, 1, 1729-1737; Chen et al., Chem. Rev., 2012, 112, 5919-5948; Gankanda et al., J. Phys. Chem. C, 2014, 118, 29117-29125; George et al., Chem. Rev., 2015, 115, 4218-4225; Ma et al., EST, 2021, 55, 8604) and may participate in the oxidation of volatile organic compounds (Stockwell et al., J. Geophys. Res.-Oceans, 1983, 88, 6673-6682; Wayne et al., Atmos. Environ., 1991, 25, 1-203; Atkinson et al., Atmos. Environ., 2003, 27, 197-219; Shen et al., EST, 2021, 55, 15658-15671). It is an ideal situation that the supposed intermediates or radicals can be detected in the experimental studies. By using ESR and chemical probe methods, we can accomplish the detection of some radicals such

as OH radical, as has been shown in Figure S10. With the development of more dedicated and portable instruments exploited, more radicals are being expected to be detected easily to support the mechanisms analyses and the field observation results as well.

*4. The zero-order reaction kinetics still not right. The y-axis is wrong for the first-order reaction. I feel that the correlation is better in red line than black line in Figure 10. why did the R2 shows opposite results? I still can't accept the zero-order reaction kinetics, the red dots are still not in a line but a curve line with little difference from a line. The authors misunderstand my points, I mean the "60 min for HCHO equilibrium after particles injection" are maintained too long time, that a lot of HCHO (3 ppm HCHO as shown in Figure 12) will be adsorbed on the particles. Then two aspect effects will appear, the first is that HCHO occupy the active sites of particles which inhibit the photoreactions, the other is that both the adsorbed HCHO and gas-phase HCHO can attend the photoreactions. These effects lead the results of Figure 11. The authors should carefully consider the kinetic analysis part, in case mislead the readers.*

**Response:**

The reaction indeed fits zero-order kinetics, not the first-order as the reviewer said. Table 4 shows the raw data of the HCHO concentrations as well as the logarithm (ln) of the concentration over time in $TiO_2$ and 4 wt.% $KNO_3$-$TiO_2$ reaction systems. The curves based on the data were shown in Figure 20. We are sorry that we forgot to exhibit the lnC as the right y-axis in our last response file, and make the reviewer misunderstand. It can be seen that the C-t curve owns better correlation coefficient ($R^2$) than that of lnC-t curve for both $TiO_2$ and $KNO_3$-$TiO_2$ reaction system, indicating that the reaction fits zero-order kinetics better. In response to the reviewer's feeling of red line having a better correlation than black line, this is because the values become smaller after taking ln, resulting in less difference between the data and therefore appearing more linear. However, the correlation should be judged by correlation coefficients. Therefore, the zero-order reaction kinetics is right and the kinetic analysis part of the text is correct.

Table 4. The raw data of HCHO concentrations over time for $TiO_2$ and 4 wt.% $KNO_3$-$TiO_2$ particles.

| Time (min) | HCHO + $TiO_2$ | | HCHO + 4 wt.% $KNO_3$-$TiO_2$ | |
|---|---|---|---|---|
| | HCHO concentration (ppm) | ln(HCHO concentration) | HCHO concentration (ppm) | ln(HCHO concentration) |
| 0 | 8.49398 | 2.13936 | 8.65642 | 2.15830 |
| 30 | 8.28188 | 2.11407 | 8.40190 | 2.12846 |
| 60 | 8.18289 | 2.10205 | 8.06253 | 2.08723 |
| 90 | 7.82938 | 2.05788 | 7.76558 | 2.04970 |
| 120 | 7.57486 | 2.02483 | 7.29895 | 1.98773 |
| 150 | 7.16479 | 1.96918 | 6.83232 | 1.92166 |
| 180 | 6.88199 | 1.92891 | 6.15358 | 1.81703 |

[Figure]

Figure 20. The comparison of correlation coefficients of zero- and first-order reaction curves. The reaction systems are "HCHO + $TiO_2$" (a) and "HCHO + 4 wt.% $KNO_3$-$TiO_2$" (b), both under 365 nm LED lamps at 293 K and 0.8% of relative humidity.

Concerning the HCHO adsorption as the reviewer mentioned, it is the first step of gaseous species for occurring its heterogeneous reaction on the surface of particles.

It needs 60 min for HCHO to reach adsorption saturation as shown in Figure S4. The oxidation process of HCHO has been described in detail in our previous published paper (Scientific Reports, 2017, 7, 10.1038/s41598-017-01396-x). It can be seen from Figure 21 below (Figure 1 in the published paper) that HCHO occurred its oxidation during the reaction with formic acid and $CO_2$ as the intermediate and final product, respectively. The total carbon involving HCHO, $CO_2$ and formic acid was near stable throughout the experiments (Figure 21 (d)), implying that the mass balance of carbon was almost closed. This indicates that adsorbed HCHO at this concentration level does not inhibit the reactive sites of the particles, while can be degraded gradually. The effect of HCHO concentration on the renoxification had been discussed in section 3.3.3 in the revised manuscript. Decreased HCHO concentration brought out lower NOx release, which in turn proved the positive effect of HCHO.

[Figure]

Figure 21. Kinetic curves of reactant and products. (a) Concentrations of formaldehyde. (b) Concentrations of $CO_2$. (c) Concentrations of formic acid. (d) Concentrations of total carbon (cited from Scientific Reports, 2017, 7, 10.1038/s41598-017-01396-x)

*5. The authors misunderstand my question about the mixture of $HNO_3$ and $TiO_2$. $HNO_3$ has small affect on $TiO_2$, but this treatment has great impact on $HNO_3$. The*

*mixture can't be HNO₃-TiO₂. What's the state of HNO₃ on TiO₂ surface? gas adsorbed? nitrate? liquid? definitely not HNO₃ solid!!!*

**Response:**

    The preparation process of 4 wt.% $HNO_3$-$TiO_2$ composite particles is to mix $TiO_2$ with diluted nitric acid solution, followed by stirring, air-drying, and grinding, as described in the experimental section. The surface presence state of $HNO_3$ in composite particles were thought in the form of adsorbed $HNO_3$ and particle nitrate at the same time. As we have elaborated in the last revised manuscript of Line 310-312: "*This is because that HNO₃ dissociates on particle surfaces to generate NO₃⁻, such that HNO₃ exists on TiO₂ as both HNO₃(ads) and NO₃⁻(ads).*"

    Other studies have reached similar conclusions, for example, Goodman et. al (J. Phys. Chem. A, 2001, 105, 6443-6457) and Gankanda et.al (J. Phys. Chem. C, 2014, 118, 29117-29125) have demonstrated by FTIR that the interaction of $HNO_3$ with $TiO_2$ can be described by the equation (1). In this reaction, $HNO_3$ reacts with the hydroxyl groups on the $TiO_2$ surface to form adsorbed nitrate and water molecules.

$$HNO_3 + M(OH) \rightarrow MNO_3(ads) + H_2O(ads) \text{ (1)}$$

(M stands for titanium surface atoms)

    On the other hand, Ti-OH in $TiO_2$ is not enough alkaline to eliminate $H^+$ in $HNO_3$. Furthermore, the photogenerated hole can react with adsorbed water on the surface of $TiO_2$ under illumination, with the resulting $H^+$ enhances the surface acidity. So it is partly in form of adsorbed $HNO_3$ as well on the surface.

    There has been many studies that conducted in laboratory and field observation (Environ. Sci. Technol. 2016, 50, 3530; Environ. Sci. Technol. 2017, 51, 6849; Nat. Geosci. 2011, 4, 440; Environ. Sci. Technol. 2013, 47, 815; ACS Earth Space Chem. 2019, 3, 811) found that the photolysis of surface-adsorbed $HNO_3$ or particle nitrate enhanced a lot compared to gas-phase $HNO_3$ photolysis and could be a reactive nitrogen species ($NO_x$ and HONO) source. It is thought that in the presence of protons, the adsorbed $NO_3^-$ can interact with protons by H-bonding and electrostatic force. Because $HNO_3$ has a stronger oxidation ability than $NO_3^-$, the reduction of the formed $HNO_3$ to $NOx$ or HONO is much more thermodynamically favorable (Environ. Sci.

Technol. 2020, 54, 3121). The fact observed in our study of significant NOx release when using $HNO_3$-$TiO_2$ particle is consistent with the previous studies, and the enhanced NOx release in the presence of HCHO was highlighted with its role being as a proton donor. Focusing on this, this manuscript presents a new pathway of renoxification on photoactive mineral dust and raises the issue of possible effect of atmospheric volatile organic compounds on this process.